physiology

body fat, dietary intervention, stable isotopes, water turnover

**Author for correspondence:**
Rebecca Rimbach
e-mail: rrimbach@gmail.com

# Air temperature and diet influence body composition and water turnover in zoo-living African elephants (*Loxodonta africana*)

Herman Pontzer[1,2], Rebecca Rimbach[1,3], Jenny Paltan[4], Erin L. Ivory[5] and Corinne J. Kendall[5]

[1]Evolutionary Anthropology, and [2]Duke Global Health Institute, Duke University, Durham, NC, USA
[3]School of Animal, Plant & Environmental Sciences, University of the Witwatersrand, Johannesburg, South Africa
[4]Department of Anthropology, Hunter College, New York, NY, USA
[5]North Carolina Zoo, Asheboro, NC, USA

(iD) RR, 0000-0003-3059-0382; CJK, 0000-0003-4429-4496

African elephants, the largest land animal, face particular physiological challenges in captivity and the wild. Captive elephants can become over- or under-conditioned with inadequate exercise and diet management. Few studies have quantified body composition or water turnover in elephants, and none to date have examined longitudinal responses to changes in diet or air temperature. Using the stable isotope deuterium oxide ($^2H_2O$), we investigated changes in body mass, estimated fat-free mass (FFM, including fat-free gut content) and body fat in response to a multi-year intervention that reduced dietary energy density for adult African elephants housed at the North Carolina Zoo. We also examined the relationship between air temperature and water turnover. Deuterium dilution and depletion rates were assayed via blood samples and used to calculate body composition and water turnover in two male and three female African elephants at six intervals over a 3-year period. Within the first year after the dietary intervention, there was an increase in overall body mass, a reduction in body fat percentage and an increase in FFM. However, final values of both body fat percentage and FFM were similar to initial values. Water turnover (males: $359 \pm 9 \, l \, d^{-1}$; females: $241 \pm 28 \, l \, d^{-1}$) was consistent with the allometric scaling of water use in other terrestrial mammals. Water turnover increased with outdoor air temperature. Our study highlights the

physiological water dependence of elephants and shows that individuals have to drink every 2–3 days to avoid critical water loss of approximately 10% body mass in hot conditions.

## 1. Introduction

As the world's largest living land animals, African elephants (*Loxodonta africana*) are of particular interest in comparative and evolutionary biology, providing insight into the structural, physiological and ecological demands of terrestrial life at extreme size [1–5]. In the wild, African elephants are remarkably active and social, travelling approximately 8–12 km per day over expansive home ranges [4,6] and maintaining complex, long-term social relationships with a large number of affiliates [7,8]. Much of their movement, particularly in arid climates, is oriented around water sources [9]. Much of southern Africa is predicted to get drier and hotter in the future [10]. Such climatic changes are probably going to place increased water stress on elephants inhabiting these areas. Increased water stress may result in conflict with humans over this resource and could negatively impact the population of African elephants which are currently considered as 'vulnerable' by the International Union for Conservation of Nature [11].

In addition to the daily demands of water for normal physiological function, elephants use water for evaporative cooling by wetting their skin and via transfer across their skin [12,13]. Owing to their large size, elephants use more water per day than other land animals [14,15]. A recent study reported daily water turnover of $222 \pm 60 \, l \, d^{-1}$ ($n = 20$) for zoo-living female African elephants [16], in line with earlier estimates [14,15]. Water requirements may be considerably greater in hot climates. Human managed elephants have a total evaporative water loss (i.e. expired + cutaneous) of $22 \, l \, d^{-1}$ in cool conditions (18°C day and night) and greater than $100 \, l \, d^{-1}$ in hot conditions (35°C day, 20°C night) [12]. Body water plays a central role in thermal regulation, cell volume and nutrient transport [17] and water conservation is especially important for animals living in arid areas. However, the effects of body size and air temperature on elephant water budgets have yet to be quantified.

Health of zoo elephants has become a focus of research aiming to improve husbandry practices [18,19]. Management strategies that promote daily physical activity and social interaction are associated with positive health outcomes of elephants in human care [18]. Walking 14 h per week or more, feeding on an unpredictable schedule, and low dietary diversity were all found to correlate with lower, healthier body condition score (BCS) [20]. Larger enclosures and more time spent outside and on naturalistic, soft surfaces (e.g. grass) were also correlated with better foot health, an area of particular concern for elephants [21–23].

More than 55% and 70% of European and North American zoo elephants, respectively, are considered overweight or obese [20,24]. Elephants with a high BCS also have high levels of serum triglycerides, a marker of adiposity [20]. Elephants that walk more per day also have lower levels of serum leptin [25], a hormone that is synthesized primarily by adipose tissue and reduces appetite [26]. Negative health effects associated with obesity have been documented in zoo elephants [16,18,20,27,28]. Recent research has advanced our understanding of elephant biology and health [16,18,28,29], especially with regard to their nutritional and activity requirements. For example, a greater than 50% reduction in dietary energy content resulted in a decrease in BCS of two obese Asian elephants (*Elephas maximus*) housed in a Brazilian zoo [30]. Diet changes that increased roughage and decreased the amount of concentrates successfully decreased BCS in African (*L. africana*) and Asian (*E. maximus*) elephants [19]. Changes in BCS were tracked over a long time span in Asian, but not in African elephants [19]. Longitudinal studies are currently missing to examine the effects of dietary interventions on changes in body mass and adiposity in zoo-living African elephants.

To date, most studies of elephant physiology and health have been cross-sectional in design (but see [19]). An underused, but potentially powerful, complementary approach to improving our understanding of elephant physiology and health is to measure longitudinal responses to changes in diet, activity, or other environmental variables (such as precipitation and ambient air temperature). Longitudinal studies require greater time investment, but allow comparisons within subjects and can help isolate the effects of specific variables (i.e. environmental or intervention strategies). In zoo settings, manipulations can be controlled and elephants monitored regularly to assess effects. Prospective studies could be particularly useful for resolving the effects of diet and activity on body mass and adiposity.

The aims of this study were to: (i) examine the effects of dietary changes that decreased energy content on body mass and body fat in zoo-living, adult African elephants, over a 3-year period; and (ii) to assess the effects of body size and air temperature on elephant water budgets. We measured adiposity and water turnover using deuterium dilution and depletion rates, respectively. This

minimally invasive approach provides estimates of total body water (TBW, including gut content), fat-free mass (FFM, including gut content), fat mass (FM) and daily water turnover ($l\,d^{-1}$). We predicted that changes in diet that increased browse and reduced the pellet component and energy content of the diet would reduce body fat percentage and body mass, but not affect FFM, among elephants. Furthermore, we predicted that water turnover among elephants would follow the body size scaling evident in other mammals, and that warmer air temperatures would increase daily water turnover. In order to assess some of the physiological challenges associated with their large body size, we compared daily water turnover among elephants to that of other mammals and measured the effects of seasonal changes in air temperature on daily water turnover.

# 2. Material and methods

## 2.1. Sample and measurement periods

North Carolina Zoo (Asheboro, NC, USA) houses six African elephants (four female, two male), all are adult animals (aged 32–43) except for one female who was 13 years of age at the beginning of this study. One adult female was not included in the study as it was not possible to weigh her during the study period. None of the females was pregnant or lactating during the study. Elephants were weighed monthly via a digital scale (Avery Weigh Tronix Model 640; ±2.3 kg) which is covered by a platform. Elephants were asked to step on the scale and the weight was taken by an experienced keeper who ensured that all four feet were on the platform. The scale is tested and serviced regularly. At the beginning of the study, elephants weighed 2556–5785 kg. These elephants are primarily housed on two 1.4 ha habitats and are generally given access to the habitats throughout the day and the night. On the habitat, they have access to tree shade, cool pools and mud wallows, and the habitat includes areas where trees block the wind. When air temperatures are below 4°C and occasionally overnight, elephants are housed in or given access to a 537 $m^2$ barn with four attached outdoor paddocks (237 $m^2$ each). At the barn, they can move between paddocks and stalls, unless air temperatures dropped below −1°C, in which case they were secured in the barn for warmth. Elephants were housed in mixed social groups including male and female animals. Social structure of the group varies, and animals are housed by themselves (for males), in pairs (for females) or in groups of three (two females, one male), four (all females), or five (four females, one male).

Body mass, TBW (including gut content), FFM (including gut content) and daily water turnover ($l\,d^{-1}$) were measured twice a year during a 3-year period (2015–2017). Measurements were performed in January and March 2015, August 2015, March 2016, September 2016, February–March 2017 and October 2017. Owing to logistical constraints, only two elephants were included in the initial measurement (winter 2015). Owing to problems with dosing (e.g. uningested dose), some elephants were excluded from analyses for some measurement periods (electronic supplementary material, table S1).

## 2.2. Body composition and water turnover

Body composition and water turnover were determined through deuterium ($^2H_2O$) dilution and depletion (see the electronic supplementary material, table S1 for information on deuterium depletion rates and dilution space). Elephants were administered doses of 99.8% deuterium (Sigma Aldrich) at 0.04 $g\,kg^{-1}$ of body mass. These doses (approx. 225 g for males, approx. 150 g for females) were injected into dry bread and fed to the elephants, similar to the method previously used in African elephants [16]. The dosing occurred during morning training sessions. We did not find that the time of day of dosing with deuterium occurred, or the interaction between time of day and identity had a significant influence on the measurements of FFM, TBW, body fat percentage and water turnover (ANOVAs: $p > 0.47$). Animals were on habitat overnight, with access to hay, browse and grass and would come to the barn in the morning where they may have ingested 1–2 flakes of hay (roughly 3 kg) before dosing, but animals were fed the majority of their diet after dosing on their habitat.

Blood samples (5 ml) were drawn via venipuncture once prior to deuterium administration and then two or five times over the subsequent 10 days, at as close as feasible to multiples of 24 h. The range of absolute deviation from multiples of 24 h was 1–327 min (25th percentile = 22.5 min; median = 40 min, 75th percentile = 80.75 min). Blood was collected from a vein on the back of the ear using a 21- or 19-gauge needle. Animals were not restrained during this procedure. They were asked to present their side parallel to the bollards and to stick their ear through an ear portal. The ear was cleaned with

warm water and then the area of the blood draw was disinfected prior to the blood draw. Animals are able to decide if they want to participate in the blood draw or not. Blood samples were centrifuged immediately after collection to separate the plasma and plasma samples were stored at −80°C. Plasma samples were then shipped to Hunter College, NY and analysed by J.P. and H.P. for deuterium enrichment using cavity ring-down spectrometry (Picarro 2017–2018). Initial deuterium enrichment was calculated using the slope-intercept method [31–33] and compared to baseline (pre-dose) enrichment to calculate TBW. Time of sampling and ln-isotope enrichment regression were strongly correlated (median $R^2$ = 0.997; range 0.990–1.000). The rate of deuterium depletion over the 10-day measurement period was used to calculate daily water turnover ($l\,d^{-1}$).

FFM (including gut content) was calculated from TBW as FFM = TBW/0.746, using the mean ratio of TBW : FFM for 15 mammal species reported by Wang *et al.* [34]. These ratios have not been empirically determined for elephants, but are generally conserved among mammals. It should be noted here that Wang *et al.* [34] reviewed studies that excluded gut contents when determining hydration, examined animals which were deprived of food and water, or studies that used carcasses. Using data from Wang *et al.* [34], the standard deviation of the mean ratio of TBW : FFM for 15 mammal species is 0.0175, or 2% coefficient of variation. Elephants in this study were never deprived of food or water, and thus their body masses include gut contents, which can account for 7–17% of body mass in elephants [35]. Gut contents provide an additional reservoir for isotope dilution, and are therefore included in calculations of TBW and hence FFM. We discuss the potential effects of gut content variation and uncertainty in the TBW : FFM ratio for our analyses below. FM was calculated from the difference in body mass and FFM.

## 2.3. Comparative data for water turnover

In order to test whether elephants fit the relationship between body size and water turnover in other mammals, we assembled published data on water turnover for 62 species of mammals [36,37]. We excluded non-terrestrial species (i.e. bats and aquatic mammals) because ecological pressures on water turnover in these species may differ from terrestrial species. We also excluded marsupials because their daily energy expenditure differs from that of eutherian mammals [38], and daily expenditure is related physiologically to water turnover. Likewise, for two species, we excluded water turnover measurements taken during seasons of energy stress and reduced metabolic rate (*Equus caballus* Shetland ponies: winter [36]; *Oryx leucoryx* Arabian oryx: summer [37]). The only species with multiple measures in the comparative dataset is the domestic horse, *Equus caballus*, for which we included three breeds with different body masses (Shetland ponies: 160 kg; Welsh mountain ponies: 212 kg; thoroughbred: 493 kg).

## 2.4. Daily air temperature

To assess the effect of air temperature on water turnover, we recorded mean daily air temperature for each measurement period using data from the Asheboro Municipal weather station (Asheboro, NC), 27 km from the zoo, as reported on weatherunderground.com. We used air temperature as a crude estimate of ambient temperature that elephants experience in shaded locations. Studies conducted on African elephants in southern Africa found that air temperature measurements track black globe temperatures, which provide an integrated measure of air temperature, radiant temperature and the cooling effect of wind [39,40]. We calculated average maximum, minimum and mean air temperature for each 10 day measurement period using readings of daily maximum, minimum and mean air temperature. We also calculated mean dewpoint and cumulative precipitation over each 10 day measurement period.

## 2.5. Diet management

In February 2016, North Carolina Zoo transitioned all elephants from a 'high inclusion' of pellet feed (Wild Herbivore Plus; 10–15% of total calories on a dry matter basis) to a 'low inclusion' diet with grain-free pellet feed (Mazuri Hay Enhancer™; 5% of total calories on a dry matter basis). Prior to February 2016, female elephants were provided with 4–5 kg $d^{-1}$ of standard pellets (Wild Herbivore Plus) and males with 9–11 kg $d^{-1}$; these amounts were reduced to 3 kg $d^{-1}$ and 4 kg $d^{-1}$ of the grain-free pellets (Mazuri Hay Enhancer™) beginning in February 2016, respectively. The grain-free pellet has higher crude fibre (30% versus 22%) and lower digestible energy content (2595 kcal $kg^{-1}$ versus 3030 kcal $kg^{-1}$ as fed). Distribution of hay was also modified. Prior to February 2016, each elephant was given two rectangle bales of timothy hay per day. Beginning in February 2016, each elephant was

provided two and a half to three bales of hay, with an additional two to three bales distributed in the habitat. Furthermore, in conjunction with these dietary modifications, greater effort was made to supplement the diet with natural browse, and the amount of browse (collected from vegetation found on zoo grounds) provided was increased. Elephants had access to hay, browse and grass during the night. Most food items were consumed overnight. In the morning, elephants received 1–2 flakes of hay as 'breakfast'. The pellets were used in enrichment or for training and each elephant was given their appropriate quantity of grain individually. However, data on daily dietary consumption per individual is not available. Further information on the nutritional effects of the diet change are available [41]. Behavioural observations, performed as sporadic scan sampling between 09.00 and 17.00 during January 2015–November 2016, collected before and after the diet change suggest that elephants did not increase total time spent locomoting or foraging after the modification, despite the fact that the diet change allowed for greater dispersion of the food [42].

## 2.6. Statistical analyses

We analysed all data using R v. 3.5.1 [43]. To visualize group-level trends in body mass, body composition and water turnover, we divided each subject's measurements for a given measurement period by the subject's mean value for that measurement over the course of the study. We used paired *t*-tests to examine whether body mass, FFM and body fat percentage of the study animals had changed throughout the study. We used an elephant's first (January 2015: $n = 1$; March 2015: $n = 1$ or August 2015: $n = 2$) and last (February 2017: $n = 1$ or October 2017: $n = 3$) measurement for this analysis. Data are presented as mean ± s.d.

To examine how air temperature (average, maximum and minimum) and dew point and body mass influence water turnover, we used linear mixed models (LMMs) with Gaussian error structure using the package 'lme4' [44]. We log-transformed water turnover to reduce heteroscedasticity when fitting the models, and we used individual identity as a random factor because individuals were sampled repeatedly. We did not include sex as a predictor in the models because there was collinearity between sex and mass. We included the interaction term between air temperature and body mass, and removed this interaction owing to non-significance. We standardized (*z*-transformed) all numeric predictors for more accurate model fitting and to facilitate comparisons of model estimates [45]. We verified the model by inspecting Q–Q plots and by plotting model residuals against fitted values. We checked for multi-collinearity by calculating variance inflation factors [46] for the predictor variables using the 'vif' function in the car package [47], which did not indicate collinearity (all vifs < 2).

We used ANOVAs that included time of day, individual identity and the interaction between time of day and identity to assess whether measurements of FFM, TBW, body fat percentage and water turnover were influenced by the time of the day when dosing with deuterium occurred. We used Pearson's product-moment correlations to examine overall and within-individual relationships between body mass, FFM and body fat percentage. One female was 13 years old at the onset of the study, and thus, was still growing throughout the study [48]. To assess if including an individual in its growth phase influenced our results, we conducted all analyses again excluding this female. There were almost no differences and thus, we report results including this individual and mention results that changed after excluding this individual.

## 3. Results

Individuals varied in body mass, body composition and water turnover, with males weighing considerably more and using more water (table 1 and figure 1). Elephants in this study gained body mass over the first year of diet change (average increase March 2016–February 2017: 324 ± 165 kg). This increase in body mass was largely the result of increased FFM (including gut content, average increase March 2016–February 2017: 593 ± 385 kg; figure 1). Males had a significantly larger body mass than females (estimate ± s.d. = 2021.869 ± 675.103, $t = 2.994$, $p = 0.002$), and body mass was weakly influenced by mean daily air temperature (estimate ± s.d. = −8.474 ± 4.281, $t = −1.979$, $p = 0.047$). There was no relationship between body mass and mean daily air temperature when excluding the young female (estimate ± s.d. = −8.790 ± 5.190, $t = −1.693$, $p = 0.09$). Body fat percentage generally decreased during this period (−6.0 ± 5.4%), but considerable variation was evident among subjects (figure 1). Final measurements of body mass, FFM (including gut content) and body fat percentage were similar to the initial values (paired *t*-tests: body mass: $t_3 = −0.058$, $p = 0.957$; FFM: $t_3 = −0.086$, $p = 0.936$; body fat percentage:

**Table 1.** Key characteristics for the five elephants in this study. (Means ($\pm$s.d.) are shown. Data per sampling event presented in the electronic supplementary material, table S1.)

| subject | sex | age | body mass (kg) | total body water (l) | fat-free mass (kg)[a] | body fat % | water turnover (l d$^{-1}$) |
|---|---|---|---|---|---|---|---|
| 1770 | M | 32–34 | 5815 ± 190 | 4020 ± 349 | 5389 ± 467 | 7 ± 6 | 353 ± 43 |
| 27 | M | 41–43 | 5553 ± 107 | 3698 ± 177 | 4957 ± 238 | 11 ± 4 | 365 ± 97 |
| 1772 | F | 13–15 | 2654 ± 96 | 1731 ± 79 | 2320 ± 106 | 13 ± 2 | 227 ± 40 |
| 1611 | F | 33–35 | 4401 ± 140 | 2817 ± 131 | 3776 ± 175 | 14 ± 2 | 223 ± 42 |
| 1771 | F | 37–39 | 3905 ± 150 | 2587 ± 142 | 3467 ± 190 | 11 ± 2 | 273 ± 47 |

[a]Estimates of FFM include gut content.

$t_3 = 0.182$, $p = 0.867$). Body fat percentage was significantly lower at the end of the study compared to initial values when excluding the young female ($t_2 = 6.608$, $p = 0.022$). Time of dosing with deuterium and the interaction between time of day and identity did not influence estimates of FFM, TBW, body fat percentage and water turnover (ANOVAs: all $p > 0.47$).

There was a strong positive relationship between FFM and body mass (Pearson's product-moment correlation: $t_{22} = 26.10$, $p < 0.0001$, $R^2 = 0.96$; figure 2a), and a weaker and negative relationship body fat percentage and both FFM ($t_{22} = -3.17$, $p = 0.004$, $R^2 = 0.31$) and body mass ($t_{22} = -2.09$, $p = 0.048$, $R^2 = 0.16$). There was no relationship between body mass and FFM within individuals M27 and F1611 (all $p > 0.39$; figure 2a), although there was a positive trend for M1770, F1771 and F1772 ($p = 0.042$–0.089; figure 2a). There was no relationship between body mass and body fat percentage within individuals (all $p > 0.21$). There was a negative relationship between FFM and body fat percentage in males (all $p < 0.01$; figure 2b), but not in females (all $p > 0.15$; figure 2b).

Daily water turnover was related to both body mass (estimate ± s.d. = 0.186 ± 0.055, $t = 3.345$, $p = 0.0008$; figure 3a) and mean daily air temperature (estimate ± s.d. = 0.141 ± 0.028, $t = 5.045$, $p < 0.0001$). Mean water turnover increased with mean body mass at a rate of approximately 4.5 l d$^{-1}$ for every 100 kg increase in body mass. In one individual (F1611), water turnover was negatively correlated with body mass ($t_1 = -26.07$, $p = 0.02$, $R^2 = 0.98$), but this relationship was not found in the other individuals (all $p > 0.05$).

Water turnover fluctuated seasonally, with lower turnover in the cooler months (figure 1). When mean daily air temperature ranged from 6 to 14°C, water turnover was 325 ± 30 l d$^{-1}$ for males and 218 ± 38 l d$^{-1}$ for females. During warmer measurement periods (mean daily air temperature 23–24°C) water turnover was 427 ± 89 l d$^{-1}$ for males and 278 ± 18 l d$^{-1}$ for females. Measured as a percentage of TBW, water turnover increased at a rate of approximately 1% TBW for every 4°C increase in mean daily air temperature (figure 3b). Daily water turnover was also related to maximum (estimate ± s.d. = 0.527 ± 0.229, $t = 0.229$, $p = 0.032$) and minimum air temperature (estimate ± s.d. = 0.715 ± 0.201, $t = 3.548$, $p = 0.002$), and dew point (estimate ± s.d. = 0.641 ± 0.212, $t = 3.014$, $p = 0.007$).

Water turnover among adult African elephants was consistent with that of other mammals. Mean water turnover for males (body mass: 5684 ± 200 kg), averaged across all periods (359 l d$^{-1}$), was 17% above the regression for other mammals (figure 4). Mean water turnover for female elephants (body mass: 3508 ± 787 kg), averaged across all periods (241 l d$^{-1}$), was 18% above the regression for other mammals. These deviations from the regression were modest relative to other species in the comparative dataset (figure 4). Indeed, the regression equation for mammals excluding elephants ($y = 0.920x - 0.974$, d.f. = 38, adj. $r^2 = 0.97$, $p < 0.0001$) was nearly identical to the regression equation including elephants ($y = 0.924x - 0.971$, d.f. = 40, adj. $r^2 = 0.97$, $p < 0.0001$; figure 4).

## 4. Discussion

Male and female African elephants in this study showed increased FFM (including gut content) and reduced body fat percentage in the first year after a dietary intervention. However, there was no difference between initial and final measurements of body mass and body composition. The difficulty in maintaining beneficial changes in body composition among these elephants mirrors the challenges experienced in other species, including humans [49–52]. Nonetheless, benefits of an improved diet, in

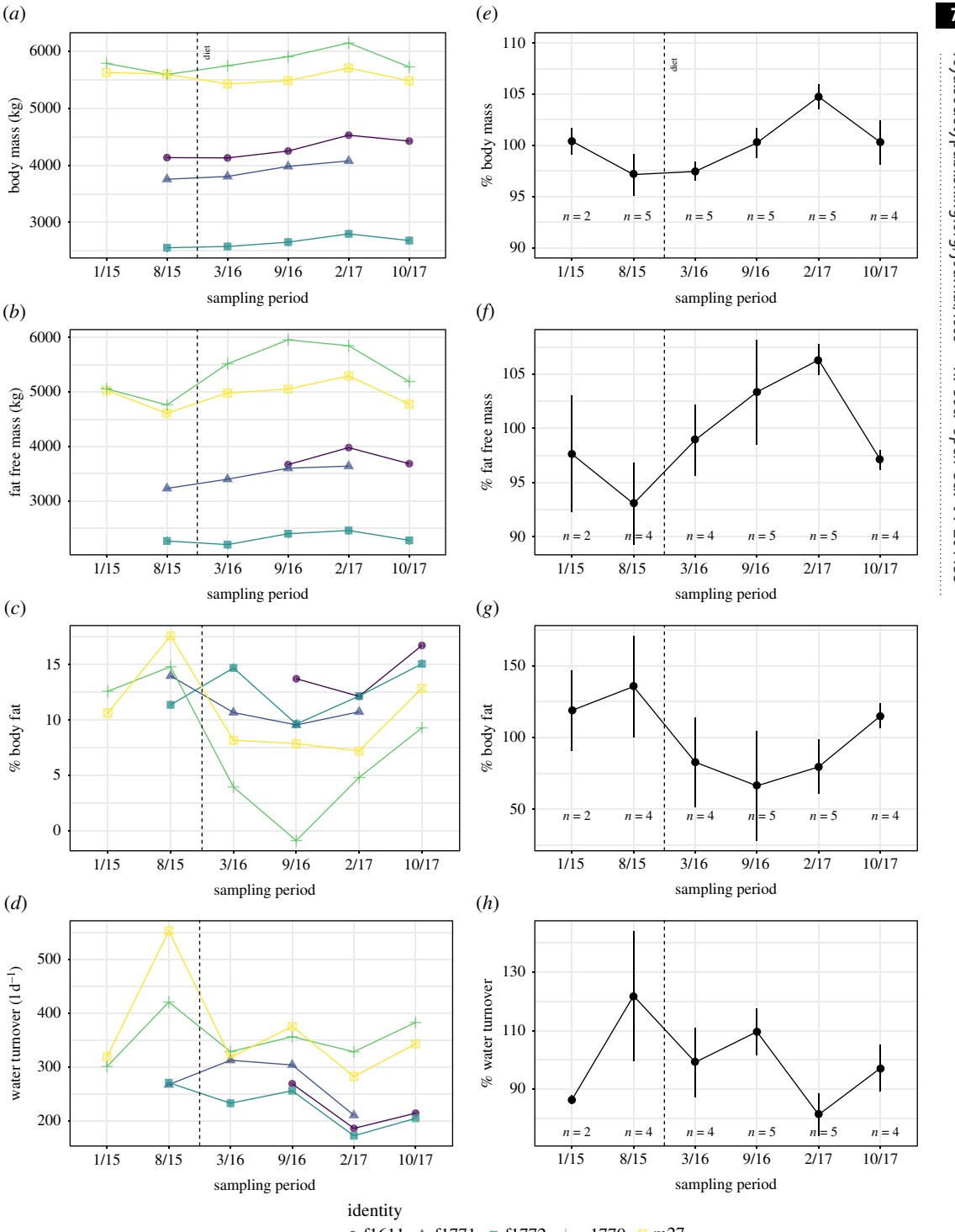

**Figure 1.** Changes in key characteristics over the course of the study. Absolute values for each subject (*a–d*) and pooled values (*e–h*) with each subject's values normalized to their mean for the study. Month/year of measurement is indicated. Diet was modified in February 2016 (vertical dashed line). Sample sizes per sampling period are presented for pooled values (*e–h*). (*b*) FFM includes gut contents. Body fat percentages presented in (*c*) and (*g*) represent body fat as a percentage of body mass. Data in the electronic supplementary material, table S1.

this case, a reduced amount of grain and caloric intake [41], can include a wide range of physiological measures, including improved cardiovascular health, even in the absence of permanent changes in body composition [50]. Future intervention studies should investigate factors that improve the duration of body composition changes and assay additional health impacts and parameters.

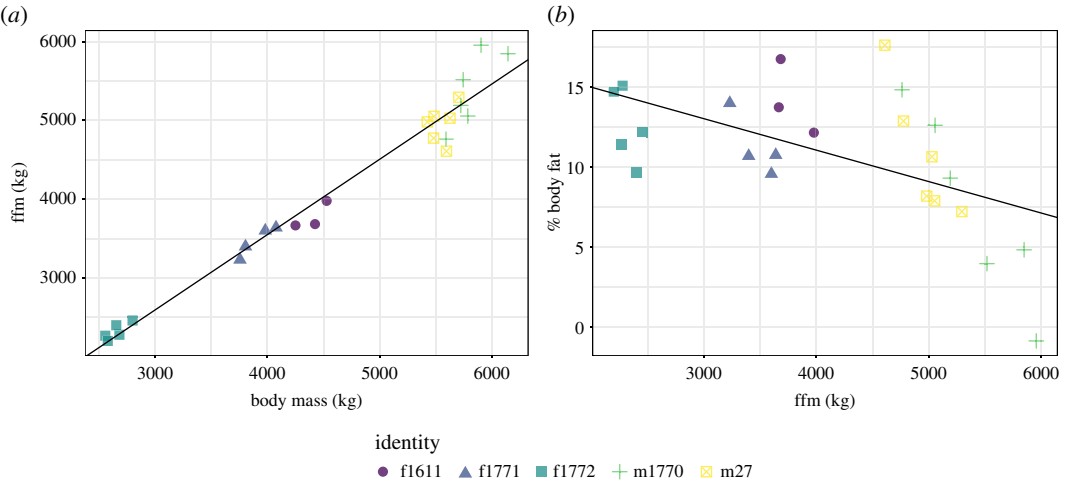

**Figure 2.** Relationship between (*a*) body mass and FFM (including gut contents) and (*b*) FFM (including gut contents) and body fat as a percentage of body mass (lines indicate linear regression lines).

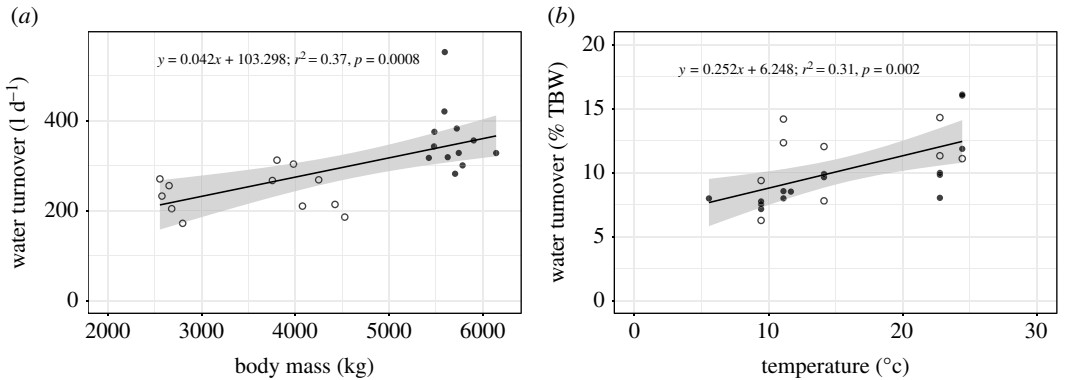

**Figure 3.** Water turnover in adult African elephants (females = open circles, males = filled circles). (*a*) Water turnover (l d$^{-1}$) increased with body mass. (*b*) Water turnover as a percentage of total body water (% TBW) increased with mean daily air temperature. Line indicates linear regression line and shaded area shows 95% confidence interval.

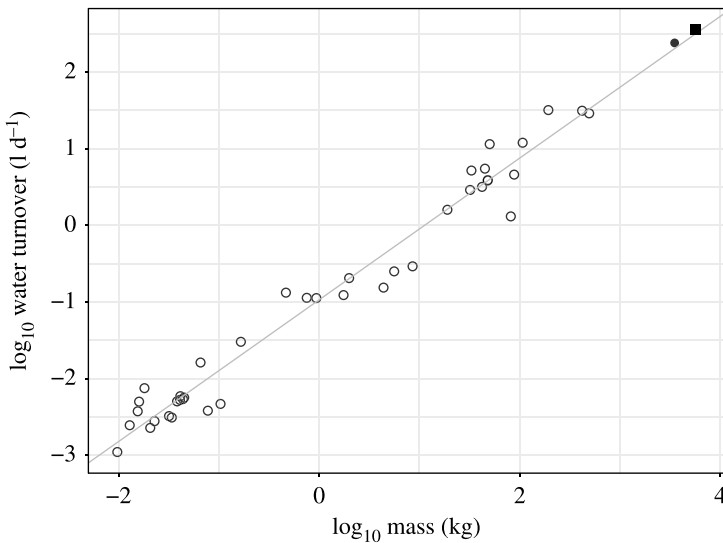

**Figure 4.** Water turnover in adult female (filled circle) and male (filled square) elephants and other terrestrial mammals (open circles). Line indicates least-squares regression for all data, including elephants: $\log_{10}$ water turnover = $0.924 \pm 0.023 \log_{10}$ mass − $0.971 \pm 0.039$ (d.f. = 40, adj. $r^2 = 0.97$, $p < 0.0001$).

Body fat percentages for elephants in this study were similar to those reported for a large adult female sample (9 ± 4%, range: 5–17%, $n = 20$) [16]. Adult males and females appear to have similar body fat percentages, but we are limited in this assessment to the small number of individuals in this study. A lack of sexual dimorphism in body fat percentage would suggest that the energetic burdens of pregnancy and lactation are not sufficiently large as to select for additional energy buffering adaptations in females. However, a much larger sample size is needed to verify this finding as a general characteristic of African elephants. Moreover, it will be pertinent to assess variation of relative body fat with age and life stage.

It should be noted that we used a TBW : FFM ratio of 0.746 in this study, based on a comparative mammalian dataset [34], whereas Chusyd *et al.* [16] used a value of 0.730. If we were to use the 0.730 value, body fat percentages for elephants in this study would be reduced approximately 2% (i.e. 11body fat percentagewould be 9%). These differences are not substantial, but the proper TBW : FFM ratio for elephants warrants future investigation as this would improve the accuracy of isotope-based assessments of body composition. For example, we believe that the negative body fat value (−1%) calculated for elephant 1770 in September 2016 reflects measurement error around a true body fat percentage that was very low, compounded by the use of a TBW : FFM value that was too low. Alternatively, this negative value could indicate that some portion of the dose was not ingested. Improved accuracy in the TBW : FFM value for elephants would enable us to assess these explanations. However, determining TBW : FFM ratio requires carcass lyophilisation, a method that is accurate but challenging for large animals.

Variation in gut fill, which can vary from 7% to 17% of body mass in elephants [35], must also be considered in interpreting measures of body composition for elephants in this study. Elephants in our sample probably increased their volume of food intake in response to the change from a high-calorie diet to a low-calorie diet [53]. Increased food intake will result in an increased gut fill [54] consisting of dry matter contents (which are mostly fat-free) and water (e.g. from saliva and other secretions). Previous studies have stated the relevance of considering gut fill when using isotope dilution methods in ruminants [55–57]. Water content of elephant gut contents [58] and faeces [59] ranges between 80–90% and thus is higher than the 73–75% hydration assumed in the calculation of FFM. Converting TBW (which includes the water in the gut) to FFM (including gut contents) using a hydration constant of 74%, as done in this study, will therefore lead to an overestimation of FFM (including gut contents). This overestimation will, in turn, lead to an underestimation of fat mass and body fat percentage. Such an underestimation of body fat has been reported in deer (*Odocoileus hemionus*) [55] and ponies, where body fat percentage determined via isotope dilution was on average 1.78% lower than carcass dissection-derived values [56]. Depending on the variation of gut water (80–90% in ruminants [56,58]) and variation in gut fill between 7 and 17% of body mass [35], it is likely that we underestimated body fat mass by 6.0–14.5% (via an over-estimation of FFM). Furthermore, within-subject variation body composition between measurement periods could be owing in part to variation in gut fill, and true FFM (excluding gut contents) could be even more stable than the weight measurements and isotope dilution analyses suggest.

Daily water turnover was high among African elephants, commensurate with their large body masses. Water requirements increased with higher environmental air temperatures, most likely owing to increased insensible water loss [12,13]. Permeability of elephant skin changes across seasons, and epidermal permeability is greater in summer compared to winter, allowing a higher rate of evaporative cooling when air temperatures are high [12]. Water turnovers for males in summer months were the highest ever measured for any animal, reaching 400 to over $500 \, l \, d^{-1}$ (electronic supplementary material, table S1). In the wild, where daily activity levels are greater, daytime temperatures exceed 40°C and animals experience a high heat load from solar radiation, water requirements are probably even greater and habitat and climate specific [12]. With TBW accounting for approximately 67% of body mass (table 1; and see [16]), and water loss in hot conditions in excess of 10% TBW, elephants would need to drink every 2–3 days to avoid critical water loss of approximately 10% body mass [60]. This result fits well with observations of wild elephants, which rarely drink less often than every 3–4 days in the dry season [61]. Elephants travel faster in the dry season than in the wet season, probably trying to minimize the time spent travelling between water and food sources [61].

Physiological water requirement provides additional context to the water-directed behaviours and ecology of elephants in the wild [9,13] and highlights the dependence of elephants on access to drinking water. Most parts of southern Africa are predicted to get drier and hotter in the future, and arid areas are expected to expand further [10]. Such climatic changes should place increased water stress on elephants inhabiting these areas which could affect migration routes and result in conflict over water with human populations (e.g. water used for agriculture). This would probably negatively

impact already vulnerable elephant populations. However, a large proportion of African elephants live in national parks and reserves that provide access to water at water holes, making conflict with humans over water less likely for these populations.

Limitations of this study include the small number of elephants measured and the relatively narrow age range of subjects. Uncertainty in gut content mass and the ratio of TBW : FFM limited the accuracy of isotope-based body composition measurements. Direct measures of tissue hydration and gut content volume would improve the application of isotope dilution methods in future studies.

Longitudinal physiological studies provide an important perspective on captive elephant health and well-being. The results of this study suggest that diet interventions can influence body composition, and that maintaining such changes remains a challenge. Long-term monitoring of body composition and other aspects of health, using stable isotopes and other minimally invasive techniques, will continue to improve the care of elephants in human settings as well as shed light on their physiological adaptations and ecology in the wild.

Ethics. The North Carolina Zoo is an accredited zoo with the Association of Zoos and Aquariums, and institutional approvals for this work were obtained prior to the study through the zoo's Research Review Committee.

Data accessibility. The dataset supporting this article has been uploaded as part of the electronic supplementary material, table S1.

Authors' contributions. H.P. and C.J.K. designed the study, H.P., C.J.K and R.R. analysed data, and wrote the manuscript. J.P. and E.L.I. assisted with data collection. J.P. and E.L.I. assisted in manuscript preparation and all authors gave final approval for publication.

Competing interests. We report no competing interests.

Funding. This study was supported by the North Carolina Zoo, Hunter College-City University of New York and Duke University.

Acknowledgements. We thank North Carolina Zoo staff for their assistance with data collection. We thank Marcus Clauss and one anonymous reviewer for their comments on earlier versions of the manuscript. Jb Minter provided comments on an earlier version of this manuscript.

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
