## [Reviewer comments · Royal Society Open Science]

Review History

RSOS-192105.R0 (Original submission)

Review form: Reviewer 1 (Marcus Clauss)

Is the manuscript scientifically sound in its present form?

No

Are the interpretations and conclusions justified by the results?

No

Is the language acceptable?

Yes

Do you have any ethical concerns with this paper?

No

Have you any concerns about statistical analyses in this paper?

No

Recommendation?

Major revision is needed (please make suggestions in comments)

Comments to the Author(s)

Please see the attached pdf file (Appendix A).

(pdf file in case the graphs do not look good in the word file and the endnote codes in word give problems)

Review form: Reviewer 2

Is the manuscript scientifically sound in its present form?

No

Are the interpretations and conclusions justified by the results?

No

Is the language acceptable?

Yes

Do you have any ethical concerns with this paper?

Yes

Have you any concerns about statistical analyses in this paper?

No

Recommendation?

Reject

Comments to the Author(s)

The authors aimed to (1) investigate how changes in diet altered body mass and body fat in five zoo African elephants, with measurements made once or twice a year over a three-year period, and (2) to investigate the effects of body size and ambient temperature on elephant water budgets. While I understand the difficulties of working with elephants, the study is limited by the small sample size and weak methodology, particularly for the change in diet (no details of exact diet composition or intake by the elephants is given - see my comments below). Despite aiming to give the elephants "an improved diet", their body mass and body fat were the same at the end of the study, changing only transiently. There is no information on exactly how dietary energy intake changed over time. There is no information on other factors that could have influenced the elephants' welfare and body mass, for example, activity level, exposure to heat or cold, disease, social interactions. The effect of temperature on water balance is assessed using data from a weather website, not taking into account the actual microclimate that the elephants would have experienced, for example, by being exposed to solar radiation (or the effect of other factors like activity level). Overall, the study provides few new findings and only tentative data from the small sample. Indeed, the authors recognise that and temper almost every conclusion they draw in the Discussion with a limitation and suggestion for future study. It is well known that elephants have a high water turnover and that increasing environmental heat load will increase that water turnover, and that body size influences water turnover. Unfortunately, I do not think this study provides a significant advance to the literature.

Specific comments:

General: I think that the Introduction is long and its focus could be improved. It jumps from studies of zoo elephant welfare to the physiology and activity of wild elephants, to ideas about evolved ecology, then back to zoo management strategies. I would suggest a brief introduction to wild elephant physiological ecology (including diet, water needs and activity), then moving the

focus to the zoo and how best to manage elephants in that setting, before identifying the gap for the study and the study approach. It can also be made more concise by removing unnecessary detail, for example, “Morfeld and colleagues (10) developed a 5-point body condition score (BCS) to visually assess whether elephants are under- or overweight. Higher scores indicate greater body fat and were correlated with a biological marker of adiposity (serum triglycerides; 10,23). More than 70% of the 240 elephants in their study had a BCS of 4 or 5, and were considered overweight or obese”, could be reduced to “elephants with a higher body condition score had higher serum triglycerides, a marker of adiposity, and more than 70% of elephants were considered overweight or obese”.

Comments per line:

8: This comment would apply to all animals.

13: Data on reduced dietary density is not available in the paper, as far as I can tell.

44-46: Elephants do not only achieve evaporative water loss by wetting their skin. There also is substantial transfer of water across a permeable integument, as described by Dunkin et al.

56: Throughout the paper (including the title), please specify what temperature you are referring to. The reader could think it is body temperature. Environmental temperature also can be measured in various ways (e.g. dry-bulb, wet-bulb, soil, black globe temperatures). Note also that dry-bulb air temperature alone does not describe the heat load experienced by animals outdoors.

79: I don't think the animals in this study were not monitored daily?

82: What dietary changes? A diet with lower energy content, different nutritional content?

87-89: Is the prediction based on different energy content for the two food types or is the energy content matched?

98: How did you account for one elephant being younger and still growing?

99: Please explain here or elsewhere in the Methods how the elephants were weighed, and the accuracy of the measurement. Please give the initial mass of all elephants here in the Methods, particularly since relationships with body mass are investigated in this study.

100-103: Given that the effect of environmental temperature is a key aim of the study, much more detail is required to describe exactly how the elephants were housed. For example, did the animals have access to shade, were they exposed to open skies at night, were the animals able to choose their own microclimate or were they herded in and out of the barn?

103-105: Exactly how were the five study animals housed? All together, or in two groups?

117: Please explain how elephants were restrained for blood sampling and where blood samples were taken from.

119: Were the plasma samples kept at a particular temperature?

145: Humidity (in the subtitle) is not measured/reported.

146: Presumably mean daily temperature refers to mean daily air temperature? What evidence is there that mean daily air temperature is suitable to quantify the heat load on elephants in the study?

147: How far is this weather station from the study site?

148-149: It is not clear what is meant here? Did you use mean, minimum and maximum daily air temperatures, or did you use the mean daily temperature you obtained and calculate the maximum of the mean, etc? Is the measurement period the 10 days over which blood samples were taken?

151: Was the actual feeding by the elephants observed? Even if the provided food was changed, how do you know how the energy intake and dietary content of each elephant changed? How much natural vegetation was provided, what was it exactly, and what was the nutritional content of this vegetation? As a main aim of the study was to investigate how dietary changes affect body mass, I would expect to see much greater detail on the nutritional content of the food and detail on actual food (and energy) intake by study elephants.

165: Please provide a brief description of how the behavioural observations were undertaken.

177: Dewpoint is not a component of ambient temperature.

193: The diet change occurred in February 2016, so why is the "mass over the first two and a half years of diet change" calculated as the "average increase Aug 15 - Feb 17"? That average includes both diets.

250-251: What data do you have to support the claim of an improved diet? Improved in what way?

250-252: This is speculation, especially for these study animals. Also, if animals were restrained for blood sampling, why were other measures of physiological health not obtained?

259: What information is available on reproductive cycling in the study animals at the times of measurements?

286: Dunkin et al. have shown how skin permeability changes seasonally resulting in much greater evaporative water loss across the skin in summer.

288-289: Daytime air temperatures can frequently exceed 40 deg C in the African elephant's natural habitat. Importantly, the animals also experience a high heat load from radiation.

291: Why is 10% body mass considered critical? Many large mammals can cope with larger percentage losses. Please provide a reference for 10% loss being critical for elephants.

299: Many elephants do not have migration routes available to them and in reserves would not be in direct conflict with humans for water.

303-304: I don't think this conclusion is valid given that the animals returned to initial values.

Figure 1: It would be helpful in the elephant ID key to know which elephants were male and female.

Decision letter (RSOS-192105.R0)

17-Jan-2020

Dear Dr Rimbach:

Manuscript ID RSOS-192105 entitled "Temperature and diet influence body composition and water throughput in zoo-living African elephants (*Loxodonta africana*)" which you submitted to Royal Society Open Science, has been reviewed. The comments from reviewers are included at the bottom of this letter.

In view of the criticisms of the reviewers, the manuscript has been rejected in its current form. However, a new manuscript may be submitted which takes into consideration these comments.

Please note that resubmitting your manuscript does not guarantee eventual acceptance, and that your resubmission will be subject to peer review before a decision is made.

Your resubmitted manuscript should be submitted by 16-Jul-2020. If you are unable to submit by this date please contact the Editorial Office.

on behalf of Professor Emily Standen (Associate Editor) and Kevin Padian (Subject Editor)
openscience@royalsociety.org

Associate Editor Comments to Author (Professor Emily Standen):

Associate Editor: 1

Comments to the Author:

Dear Dr. Rebecca Rimbach,

We have now received two comprehensive reviews of your manuscript (RSOS-192105) titled Temperature and diet influence body composition and water throughput in zoo-living African elephants. Both reviews have fairly major reservations regarding the methods and direction of presentation made in your paper. I do think there are several very useful comments and ideas put forth by the reviews and I can see that these ideas if addressed may significantly improve the impact of your paper.

For this reason we must reject the paper in its current form but would be willing to reconsider a new submission that sufficiently addresses the concerns and suggestions of the reviewers.

Sincerely,
Emily Standen

Associate Editor: 2

Comments to the Author:

Dear Dr. Rimbach,

I find your paper an interesting contribution to our understanding of elephant physiology and dietary requirements. From both a husbandry perspective and a larger physiological viewpoint, I find this to be important information on a little known mammal.

As this paper is outside of my normal expertise, I am looking forward to the reviews from scientists more versed in the fields of heavy water labeling and elephant husbandry and ecology.

Emily Standen

Editor comments:

Thanks for your submission. As you will see the reviewers had some substantial concerns. I am not sure how you will get around the criticism of small sample size without assessing some new subjects, but please address this and other comments if you plan to resubmit. Best wishes.

Reviewers' Comments to Author:

Reviewer: 1

Comments to the Author(s)

please see the attached word and pdf file

(pdf file in case the graphs do not look good in the word file and the endnote codes in word give problems)

Reviewer: 2

Comments to the Author(s)

The authors aimed to (1) investigate how changes in diet altered body mass and body fat in five zoo African elephants, with measurements made once or twice a year over a three-year period, and (2) to investigate the effects of body size and ambient temperature on elephant water budgets. While I understand the difficulties of working with elephants, the study is limited by the small sample size and weak methodology, particularly for the change in diet (no details of exact diet composition or intake by the elephants is given - see my comments below). Despite aiming to give the elephants "an improved diet", their body mass and body fat were the same at the end of the study, changing only transiently. There is no information on exactly how dietary energy intake changed over time. There is no information on other factors that could have influenced the elephants' welfare and body mass, for example, activity level, exposure to heat or cold, disease, social interactions. The effect of temperature on water balance is assessed using data from a weather website, not taking into account the actual microclimate that the elephants would have experienced, for example, by being exposed to solar radiation (or the effect of other factors like activity level). Overall, the study provides few new findings and only tentative data from the small sample. Indeed, the authors recognise that and temper almost every conclusion they draw in the Discussion with a limitation and suggestion for future study. It is well known that elephants have a high water turnover and that increasing environmental heat load will increase that water turnover, and that body size influences water turnover. Unfortunately, I do not think this study provides a significant advance to the literature.

Specific comments:

General: I think that the Introduction is long and its focus could be improved. It jumps from studies of zoo elephant welfare to the physiology and activity of wild elephants, to ideas about evolved ecology, then back to zoo management strategies. I would suggest a brief introduction to wild elephant physiological ecology (including diet, water needs and activity), then moving the focus to the zoo and how best to manage elephants in that setting, before identifying the gap for the study and the study approach. It can also be made more concise by removing unnecessary

detail, for example, “Morfeld and colleagues (10) developed a 5-point body condition score (BCS) to visually assess whether elephants are under- or overweight. Higher scores indicate greater body fat and were correlated with a biological marker of adiposity (serum triglycerides; 10,23). More than 70% of the 240 elephants in their study had a BCS of 4 or 5, and were considered overweight or obese”, could be reduced to “elephants with a higher body condition score had higher serum triglycerides, a marker of adiposity, and more than 70% of elephants were considered overweight or obese”.

Comments per line:

8: This comment would apply to all animals.

13. Data on reduced dietary density is not available in the paper, as far as I can tell.

44-46: Elephants do not only achieve evaporative water loss by wetting their skin. There also is substantial transfer of water across a permeable integument, as described by Dunkin et al.

56: Throughout the paper (including the title), please specify what temperature you are referring to. The reader could think it is body temperature. Environmental temperature also can be measured in various ways (e.g. dry-bulb, wet-bulb, soil, black globe temperatures). Note also that dry-bulb air temperature alone does not describe the heat load experienced by animals outdoors.

79: I don't think the animals in this study were not monitored daily?

82: What dietary changes? A diet with lower energy content, different nutritional content?

87-89: Is the prediction based on different energy content for the two food types or is the energy content matched?

98: How did you account for one elephant being younger and still growing?

99: Please explain here or elsewhere in the Methods how the elephants were weighed, and the accuracy of the measurement. Please give the initial mass of all elephants here in the Methods, particularly since relationships with body mass are investigated in this study.

100-103: Given that the effect of environmental temperature is a key aim of the study, much more detail is required to describe exactly how the elephants were housed. For example, did the animals have access to shade, were they exposed to open skies at night, were the animals able to choose their own microclimate or were they herded in and out of the barn?

103-105: Exactly how were the five study animals housed? All together, or in two groups?

117: Please explain how elephants were restrained for blood sampling and where blood samples were taken from.

119: Were the plasma samples kept at a particular temperature?

145: Humidity (in the subtitle) is not measured/reported.

146: Presumably mean daily temperature refers to mean daily air temperature? What evidence is there that mean daily air temperature is suitable to quantify the heat load on elephants in the study?

147: How far is this weather station from the study site?

148-149: It is not clear what is meant here? Did you use mean, minimum and maximum daily air

temperatures, or did you use the mean daily temperature you obtained and calculate the maximum of the mean, etc? Is the measurement period the 10 days over which blood samples were taken?

151: Was the actual feeding by the elephants observed? Even if the provided food was changed, how do you know how the energy intake and dietary content of each elephant changed? How much natural vegetation was provided, what was it exactly, and what was the nutritional content of this vegetation? As a main aim of the study was to investigate how dietary changes affect body mass, I would expect to see much greater detail on the nutritional content of the food and detail on actual food (and energy) intake by study elephants.

165: Please provide a brief description of how the behavioural observations were undertaken.

177: Dewpoint is not a component of ambient temperature.

193: The diet change occurred in February 2016, so why is the "mass over the first two and a half years of diet change" calculated as the "average increase Aug 15 - Feb 17"? That average includes both diets.

250-251: What data do you have to support the claim of an improved diet? Improved in what way?

250-252: This is speculation, especially for these study animals. Also, if animals were restrained for blood sampling, why were other measures of physiological health not obtained?

259: What information is available on reproductive cycling in the study animals at the times of measurements?

286: Dunkin et al. have shown how skin permeability changes seasonally resulting in much greater evaporative water loss across the skin in summer.

288-289: Daytime air temperatures can frequently exceed 40 deg C in the African elephant's natural habitat. Importantly, the animals also experience a high heat load from radiation.

291: Why is 10% body mass considered critical? Many large mammals can cope with larger percentage losses. Please provide a reference for 10% loss being critical for elephants.

299: Many elephants do not have migration routes available to them and in reserves would not be in direct conflict with humans for water.

303-304: I don't think this conclusion is valid given that the animals returned to initial values.

Figure 1: It would be helpful in the elephant ID key to know which elephants were male and female.

Author's Response to Decision Letter for (RSOS-192105.R0)

See Appendix B.

RSOS-201155.R0

Review form: Reviewer 1 (Marcus Clauss)

Is the manuscript scientifically sound in its present form?

No

Are the interpretations and conclusions justified by the results?

No

Is the language acceptable?

Yes

Do you have any ethical concerns with this paper?

No

Have you any concerns about statistical analyses in this paper?

No

Recommendation?

Accept with minor revision (please list in comments)

Comments to the Author(s)

Dear authors, I made the comments in a word file to keep some of the formatting and screenshots - please see attached (Appendix C). I also made comments in the pdf (Appendix D).

By the way, I apologize for the GLM result I put into the review last time - I tried to replicate it but came to the same p-value as you did.

sincerely m clauss

Decision letter (RSOS-201155.R0)

Dear Dr Rimbach

On behalf of the Editors, we are pleased to inform you that your Manuscript RSOS-201155 "Air temperature and diet influence body composition and water throughput in zoo-living African elephants (*Loxodonta africana*)" has been accepted for publication in Royal Society Open Science subject to minor revision in accordance with the referees' reports. Please find the referees' comments along with any feedback from the Editors below my signature.

Please submit your revised manuscript and required files (see below) no later than 7 days from today's (ie 02-Oct-2020) date. Note: the ScholarOne system will 'lock' if submission of the revision

is attempted 7 or more days after the deadline. If you do not think you will be able to meet this deadline please contact the editorial office immediately.

on behalf of Professor Emily Standen (Associate Editor) and Kevin Padian (Subject Editor)
openscience@royalsociety.org

Subject Editor Comments to Author (Professor Kevin Padian):

Thanks for your revision. There is one small thing that needs to be cleaned up before we can actually agree to publish, so please address this in your final revision. We feel it is important to address reviewers' comments. Thanks.

Associate Editor Comments to Author (Professor Emily Standen):

Dear Rebecca Rimbach,

As you can see the reviewer is grateful for your careful changes to the manuscript and feels you have done an excellent job of addressing most of his comments. Unfortunately, there still remains one rather important point that the reviewer has taken great care to explain more thoroughly. Because, according to the reviewer, this point affects the soundness of the scientific conclusions of the paper, I feel it is important to address the reviewers final concern.

Many suggestions have been included in the review to help address the uncertainty of ignoring gut contents when using doubly-labelled water to measure body composition. Note that the reviewer feels this is a critical yet 'minor' change to address.

We look forward to seeing a revised manuscript.

Emily Standen

Reviewer comments to Author:

Reviewer: 1

Comments to the Author(s)

Dear authors, I made the comments in a word file to keep some of the formatting and screenshots - please see attached. I also made comments in the pdf.

By the way, I apologize for the GLM result I put into the review last time - I tried to replicate it but came to the same p-value as you did.

sincerely m clauss

===PREPARING YOUR MANUSCRIPT===

- one version identifying all the changes that have been made (for instance, in coloured highlight, in bold text, or tracked changes);
- a 'clean' version of the new manuscript that incorporates the changes made, but does not highlight them.

 This version will be used for typesetting.

===PREPARING YOUR REVISION IN SCHOLARONE===

- Any electronic supplementary material (ESM).
- If you are requesting a discretionary waiver for the article processing charge, the waiver form must be included at this step.
- If you are providing image files for potential cover images, please upload these at this step, and inform the editorial office you have done so. You must hold the copyright to any image provided.
- A copy of your point-by-point response to referees and Editors. This will expedite the preparation of your proof.

- Ensure that your data access statement meets the requirements at <https://royalsociety.org/journals/authors/author-guidelines/#data>. You should ensure that you cite the dataset in your reference list. If you have deposited data etc in the Dryad repository, please only include the 'For publication' link at this stage. You should remove the 'For review' link.
- If you are requesting an article processing charge waiver, you must select the relevant waiver option (if requesting a discretionary waiver, the form should have been uploaded at Step 3 'File upload' above).
- If you have uploaded ESM files, please ensure you follow the guidance at <https://royalsociety.org/journals/authors/author-guidelines/#supplementary-material> to include a suitable title and informative caption. An example of appropriate titling and captioning may be found at https://figshare.com/articles/Table_S2_from_Is_there_a_trade-off_between_peak_performance_and_performance_breadth_across_temperatures_for_aerobic_scope_in_teleost_fishes_/3843624.

Author's Response to Decision Letter for (RSOS-201155.R0)

See Appendix E.

RSOS-201155.R1 (Revision)

Review form: Reviewer 1 (Marcus Clauss)

Is the manuscript scientifically sound in its present form?

Yes

Are the interpretations and conclusions justified by the results?

Yes

Is the language acceptable?

Yes

Do you have any ethical concerns with this paper?

No

Have you any concerns about statistical analyses in this paper?

No

Recommendation?

Accept as is

Comments to the Author(s)

Dear authors, thank you for all the changes. Just one word on the side that came to me in the recent past, when I had a lot of discussions about this method (not only in relation to this manuscript). In Wang et al. 1999, there is actually a whole paragraph on p. 836, right column, dedicated to the problem of which constant to use and how the constant is derived. So, while folks are citing this paper to emphasize how "constant" the hydration constant is, this paper itself discusses in detail the problems in deriving that constant depending on the state of the carcasses used and the feeding of the animals prior to turning them into carcasses. Thank you for having me review such a nice manuscript. m claus

Decision letter (RSOS-201155.R1)

Dear Dr Rimbach,

It is a pleasure to accept your manuscript entitled "Air temperature and diet influence body composition and water turnover in zoo-living African elephants (*Loxodonta africana*)" in its current form for publication in Royal Society Open Science. The comments of the reviewer(s) who reviewed your manuscript are included at the foot of this letter.

Kind regards,
Royal Society Open Science Editorial Office

on behalf of Professor Emily Standen (Associate Editor) and Kevin Padian (Subject Editor)
openscience@royalsociety.org

Associate Editor Final Comments to Author (Professor Emily Standen):

Dear Dr. Rimbach,

Thank you for addressing the final round of edits. We are pleased to accept this manuscript for publication. One final consideration for the manuscript would be to report the original data as the deuterium values in the original blood samples. We recognize in the past this is not commonly done, but perhaps is an important thing to consider in the era of access to data. If at all possible please include those values.

Thank you for your attention to detail throughout this process.

Sincerely,
Emily Standen

Associate Editor (First comments to the Author):

Dear Dr. Rimbach,

Thank you for your thorough edits to address the most recent reviewer comments. I feel you have done a good job of clarifying the reviewers concerns. I have sent it back to the reviewer one final time for approval and so they can see your kind gratitude and comments regarding his thorough review. These positive exchanges are wonderful to see as an assistant editor. Thank you for your efforts there!

I look forward to seeing this paper pushed through!

EMS

Reviewer comments to Author:

Reviewer: 1

Comments to the Author(s)

Dear authors, thank you for all the changes. Just one word on the side that came to me in the recent past, when I had a lot of discussions about this method (not only in relation to this manuscript). In Wang et al. 1999, there is actually a whole paragraph on p. 836, right column, dedicated to the problem of which constant to use and how the constant is derived. So, while folks are citing this paper to emphasize how "constant" the hydration constant is, this paper itself discusses in detail the problems in deriving that constant depending on the state of the carcasses used and the feeding of the animals prior to turning them into carcasses. Thank you for having me review such a nice manuscript. m clauss

Appendix A

RSOS-192105

Temperature and diet influence body composition and water throughput in zoo-living African elephants (*Loxodonta africana*)

Pontzer et al.

reviewed by Marcus Clauss, Zurich (does not do anonymous reviews)

This study presents serial data on the body composition of 5 African elephants as measured by the isotope dilution method. This is put into context with a diet change and environmental temperatures.

This represents an impressive setup and logistical achievement. There are two major areas of comments that I have – the internal logic of the data and implications of the method in general, and some scholarship with respect to more recent literature.

My overall recommendation is to make a completely new narrative focussing on the methodological aspect – what do isotope dilution values tell us if we do not control for intake in animals, like elephants, where a large variation in gut fill has been documented and is surely possible, especially when assessing animals on different diets and at different seasons. Because the main group that has used the method so far (Chusyd et al. 2018; Chusyd et al. 2019) has never mentioned this aspect (one is tempted to say: used the method uncritically), such a contribution would be an enormous step forward. See below under chapter 2. for details.

1. Literature (some of this is so new that it is not to be considered a criticism, just a hint; this goes a bit into the direction of not only considering the American populations)
drinking water intake in elephant (Benedict 1936)
elephants in zoos are over-conditioned as compared to free-ranging populations: BCS (Schiffmann et al. 2018), body mass (Schiffmann et al. 2019b)
longitudinal studies in elephant body condition: BCS on an individual and a population basis (Schiffmann et al. 2019a); BCS over age plotted in (Schiffmann et al. 2019b); body mass development over age indicating a putatively natural source of body mass cyclicity (Schiffmann et al. 2019c)
factors relating to foot health (Wendler et al. 2019)

2. physiological logic of data and links to method?

The main finding in the data, as they are, is against the intuitive logic of rapid body mass changes: in the data, variation in body mass is basically only due to changes in presumed fat-free mass, and not at all due to the presumed fat:

(all plots made quick-and-dirty from the supplemental information)

This is counter-intuitive: short-term changes in body mass, after a diet change (that also, as stated in the text l. 168, the “Lasky et al. (in review)”, did not lead to a change in activity, i.e. exercise), are expected to be due to fat, not fat-free mass (i.e., muscle mass).

Your data even appears to suggest that body mass is INDEPENDENT of the magnitude of the fat mass within individuals! The males – variation in fat mass appears to be COMPENSATED by fat-free mass so that overall body mass stays the same.

I am not aware of a good explanation for what you appear to observe, but I think that either you need to dig into the methodology and potential problems, or you have to state clearly that you do not know an explanation, either. Writing, in the conclusion of the paper (l. 303) “The results of this study suggest that diet interventions can improve body composition by reducing adiposity and increasing FFM.” is like proactively ignoring the physiological puzzle your data represents and gives the impression that you either do not care about the physiology, or that you proactively try to “talk away” reasonable doubts by assertive language. This is an impression you want to avoid I think.

The one measure that could fluctuate a lot (but not in the magnitude of the fat mass variation you have) is gut contents. Elephants may differ in the amount of food they actually eat (and this is something that your diet change most likely caused: an increase in total dry matter intake), and elephants are among those herbivores where increased food intake does not lead to a dramatic reduction in digesta retention time, i.e. increased food intake mostly leads to increased gut fill (increased actual gut volume) (Clauss et al. 2007). This is where I am not familiar enough with the literature on the use of isotope dilution method in large herbivores, how changes in gut fill, i.e. also in the amount of water “bound” in the gastrointestinal tract, affects FFM estimates. My guess is that gut contents, which contain water, are just part of the overall FFM that is calculated (and the factor used for calculation, here 0.746, is derived from measures that include gut content?). I just checked the paper you used as a source (Wang et al. 1999), and from reading that it becomes evident that the factor is derived in a way that makes it susceptible to changes in actual gut fill. Even if animals (for the derivation of the factor) had been investigated using the whole body (including gut contents, and not just the eviscerated carcass), the question is how they were fed before the analysis, i.e. was ad libitum intake variation on a variety of diets allowed, or not? I am not really into this literature, but the example of (Torbit et al. 1985) is nice, where this problem is specifically stated and the study design is explained in that first different amounts of food were given to the study animals (to create differences in body fat) but that

then, when the actual isotope study was done to assess body composition, intake was made constant for all animals in order to make the isotope results comparable between the individuals. This is a factor of uncertainty in studies with live animals that needs to be assessed, and with a putative variation in gut fill in elephants of a magnitude of 7-17 % of total body mass (Clauss et al. 2005), I personally believe that variation in gut fill, if not controlled for, will have a large effect on estimates of FFM and FM in elephants. By incidence, the range is similar to the range in body fat you mention in I. 256 (5-17%), but I am not sure whether, after accounting for gut content in the TBW conversion, % translates into % in this comparison.

Yet, basically, this seems to tell me that if you wanted to really assess the effect of different dietary regimes in the same animal over time, the only way to get reliable (comparable) longitudinal data is to ensure that food intake is constant AT THE MOMENT OF APPLYING ISOTOPE BODY COMPOSITION. Otherwise, differences in food intake within the same animal between the time points, e.g. due to different temperatures (less intake in hot summer? higher intake in cold winter?), different feeding regimes (higher intake on less-energy-dense diet), will affect the signal you are interpreting due to the variation in intake (and hence gut fill) at the point of measurement.

Your method do not indicate any consistency as to the amount of food available a day prior to, and during, isotope application. It does not even indicate consistency with respect to the moment of dosing in relation to feeding events or feeding behaviour.

I would suggest that you dig into that and assess whether it might relate to your data, and at least mention this option.

It would also help if you had actual intake data of your animals – your paper on the composition of the new diet (Wood et al. 2019) at least suggests that some of this information should be available for the new diet in the method section: “Daily diets were recorded by NC Zoo keeper staff for the study duration. This included number of timothy hay (*Phleum pratense*) bales, enrichment food items, and browse species”, and from these data the averages given in Table 1, and the percentages of the ingested diet in Table 3, of that paper were calculated (but the actually ingested absolute amounts are not given in that paper per season). It would be ideal, answering the methodological issues, to link that information with the isotope data!

Another factor of uncertainty might or might not be the variation in the blood sampling time points after isotope dosing (the methods seem to imply this was done a bit opportunistically, and not consistently across all assessments). Can this have an effect, or not? This should be mentioned. I also wonder whether a measure of fit for the slope-intercept method of back-calculating the initial isotope enrichment (or dilution) should be given routinely to give a measure about the “confidence” of that estimate (on which all other results hinge). Personally, I would recommend to do that.

Patterns in the data that are not mentioned, and may or may not be helpful in identifying physiological explanations:

Within individuals, there seems to be a negative correlation – is this something one expects? Does it equate to something like: better hydration status (more body water) = higher throughput because more is drunk? But that would sound like an instantaneous reaction, not something evident on an integrated measure of what happens on several days? Note that in my stats program (SPSS), the within-individual correlations were mostly not significant, though, in spite of the visual pattern. The same pattern is seen in your Fig. 2A

Why does higher water throughput seem to be correlated with less body fat? This effect tended towards significance in a GLM (random factor individual significant, water throughput $p=0.073$). The outlier would require explanation.

In particular, I would, as a reader, like to know how a decrease in gut fill would affect the results and an increase in water intake, and how the combination of these factors (if we claim that in summer, animals eat less yet drink more, both in absolute terms) would influence any estimates of FFM and FM.

3. details

I. 17 whichever way you word these findings, please ensure you do not use the words “during the first two years” because three spot samples need not represent the full time span.

I. 19 I always recommend to refrain from superlatives or “we are the first to”. If you want to keep your superlative, you cannot give the averages because (Chusyd et al. 2018) had an animals with 489 L/d, higher than the average you mention with the superlative. You need to mention your one individual value of 553 L/d that was the only one in your dataset that was higher than the maximum in the other study.

I. 159 please state whether this is gross energy, digestible energy, or what other kind of energy you are indicating, and whether the unit is kg as fed, or kg dry matter.

I. 171 I think the group level visualisations should only include those datapoints for which all data of the group are available; otherwise, a trend is implied that is not based on all individuals

Fig. 2A the intra-individual trend should be mentioned in the text (and possibly explained). x-axis label should be kg

l. 247-248 with the few measurements you have, I think you should refrain from qualitative language like “transient” and “returned” because the stability of the initial state is not ascertained in your data, and continuous measurements might indicate a similar fluctuation just as well as “stability” as implied by the word “return”

l. 263 ff the factor would have to be even higher if more gut contents were assumed, right? But see major discussion above. l. 271 points into the same direction. If the animal had a very high intake at the time, this could explain why the conversion value was too low - I think (I hope you can come up with a comprehensive explanation of the effect of intake).

I think in the acknowledgements it should be “Jeb” or similar, not “Jb”?

Sincerely marcus clauss

Literature cited:

- Benedict FG (1936) The physiology of the elephant. Carnegie Institution of Washington, Washington DC, USA
- Chusyd DE, Brown JL, Hambly C, Johnson MS, Morfeld KA, Patki A, Speakman JR, Allison DB, Nagy TR (2018) Adiposity and reproductive cycling status in zoo African elephants. *Obesity* 26:103-110
- Chusyd DE, Brown JL, Golzarri-Arroyo L, Dickinson SL, Johnson MS, Allison DB, Nagy TR (2019) Fat mass compared to four body condition scoring systems in the Asian elephant (*Elephas maximus*). *Zoo Biology* 38:424-433
- Clauss M, Robert N, Walzer C, Vitaud C, Hummel J (2005) Testing predictions on body mass and gut contents: dissection of an African elephant (*Loxodonta africana*). *European Journal of Wildlife Research* 51:291-294
- Clauss M, Streich WJ, Schwarm A, Ortmann S, Hummel J (2007) The relationship of food intake and ingesta passage predicts feeding ecology in two different megaherbivore groups. *Oikos* 116:209-216
- Schiffmann C, Clauss M, Fernando P, Pastorini J, Wendler P, Ertl N, Hoby S, Hatt J-M (2018) Body condition scores in European zoo elephants (*Elephas maximus* and *Loxodonta africana*) - status quo and influencing factors. *Journal of Zoo and Aquarium Research* 6:91-103
- Schiffmann C, Clauss M, Hoby S, Codron D, Hatt J-M (2019a) Body Condition Scores (BCS) in European zoo elephants (*Loxodonta africana* and *Elephas maximus*) lifetimes – a longitudinal analysis. *Journal of Zoo and Aquarium Research* 7:74-86
- Schiffmann C, Clauss M, Hoby S, Hatt J-M (2019b) Weigh and see – body mass recordings versus body condition scoring (BCS) in zoo elephants (*Loxodonta africana* and *Elephas maximus*). *Zoo Biology* doi 10.1002/zoo.21525
- Schiffmann C, Hatt J-M, Hoby S, Codron D, Clauss M (2019c) Elephant body mass cyclicality suggests effect of molar progression on chewing efficiency. *Mammalian Biology* 96:81-86
- Torbit SC, Carpenter LH, Alldredge AW, Swift DM (1985) Mule deer body composition: a comparison of methods. *Journal of Wildlife Management* 49:86-91

- Wang Z, Deurenberg P, Wang W, Pietrobelli A, Baumgartner RN, Heymsfield SB (1999) Hydration of fat-free body mass: review and critique of a classic body-composition constant. *American Journal of Clinical Nutrition* 69:833-841
- Wendler P, Ertl N, Flügger M, Sós E, Torgerson P, Heym PP, Schiffmann C, Clauss M, Hatt J-M (2019) Influencing factors on the foot health of captive Asian elephants (*Elephas maximus*) in European zoos. *Zoo Biology* (online) doi 10.1002/zoo.21528
- Wood J, Koutsos E, Kendall CJ, Minter LJ, Tollefson TN, Ange-van Heugten K (2019) Analyses of African elephant (*Loxodonta africana*) diet with various browse and pellet inclusion levels. *Zoo Biology* online

Appendix B

Reviewers' Comments to Author:

Reviewer: 1

Comments to the Author(s)

RSOS-192105

Temperature and diet influence body composition and water throughput in zoo-living African elephants (*Loxodonta africana*)

Pontzer et al.

This study presents serial data on the body composition of 5 African elephants as measured by the isotope dilution method. This is put into context with a diet change and environmental temperatures.

This represents an impressive setup and logistical achievement. There are two major areas of comments that I have – the internal logic of the data and implications of the method in general, and some scholarship with respect to more recent literature.

My overall recommendation is to make a completely new narrative focussing on the methodological aspect – what do isotope dilution values tell us if we do not control for intake in animals, like elephants, where a large variation in gut fill has been documented and is surely possible, especially when assessing animals on different diets and at different seasons. Because the main group that has used the method so far (Chusyd et al. 2018; Chusyd et al. 2019) has never mentioned this aspect (one is tempted to say: used the method uncritically), such a contribution would be an enormous step forward. See below under chapter 2. for details.

Response: We do not have data on individual intake and thus variation in gut fill and therefore cannot directly assess the effect of methodological differences in this paper. However, we agree this is an important issue and we have revised the paper and analyses to examine, to the extent possible given the data we have, the influence of feeding and gut fill on our results. We have included an analysis of time of day, and other factors that could influence gut fill, on our results (we find no evidence for a substantial effect in our sample). We also examine the issue of gut fill specifically in the Discussion and model the degree to which it could affect our results. The paragraph states:

“Another limitation is the fact that we did not control for variation in food intake and the resulting variation in gut fill, which can vary from 7% to 17% of body mass in elephants (Clauss et al. 2005). Dosing with DLW always occurred in the morning (with the exception of a single event when dosing occurred at 13:40). Elephants were on habitat overnight and would come to the barn in the morning where they ate one to two flakes of hay prior (roughly 3 kg) to being dosed with DLW. Due to water bound in gut content, we may have overestimated TBW by 1.4 - 3.4%, assuming that gut fill varies between 7 - 17% body mass and that water content of hay is typically around 20%. TBW of the study elephants ranged from 1640 – 4430 L and thus, an overestimation of 1.4% would equal 43 L (range: 23 – 62 L) on average and an overestimation of 3.4% would equal 104 L (range: 56 – 151 L), on average. Overestimating TBW by 3.4% will result in an overestimation of LBM by ~3% and an underestimation of percent body fat by ~3%. Thus, the measured changes in body fat percentage within ~3% of one another must be interpreted with caution, as they are within the range that changes in gut fill could affect. However, we did not find that the time of day when dosing with DLW occurred or the interaction between time of day and ID had a significant influence on the measurements of LBM, TBW, body fat percentage and water turnover (ANOVAs: $P > 0.47$), which we would expect if elephants dosed later in the day had more time to feed and thus a higher gut fill. Moreover, if the amount of water in the body was artificially high (due to water in the digestive tract), it would in turn make the dilution space N artificially high, which would result in an inflated $N*k$ (k = isotope depletion rate) value, which is how water throughput is calculated. Therefore, if there would be no variation in k , and all the apparent variation in water turnover would be due to noise in N , then N and water turnover would be positively correlated. And, since N is used to

calculate TBW, there would be a positive relationship between TBW and water turnover within individuals, which is not the case in our data. Thus, water turnover is not simply a function of noise or error in N due to variation in water in the digestive tract. Further, we note that measured changes in fat percentage exceeded 4% in most subjects, and that these changes are likely too large to be explained by variation in gut fill.” (lines 322-347)

1. Literature (some of this is so new that it is not to be considered a criticism, just a hint; this goes a bit into the direction of not only considering the American populations)
drinking water intake in elephant (Benedict 1936)
elephants in zoos are over-conditioned as compared to free-ranging populations: BCS (Schiffmann et al. 2018), body mass (Schiffmann et al. 2019b)
longitudinal studies in elephant body condition: BCS on an individual and a population basis (Schiffmann et al. 2019a); BCS over age plotted in (Schiffmann et al. 2019b); body mass development over age indicating a putatively natural source of body mass cyclicity (Schiffmann et al. 2019c)
factors relating to foot health (Wendler et al. 2019)

Response: Thank you for highlighting literature that was missing from the manuscript. We read these papers and cited them where appropriate.

2. physiological logic of data and links to method?

The main finding in the data, as they are, is against the intuitive logic of rapid body mass changes: in the data, variation in body mass is basically only due to changes in presumed fat-free mass, and not at all due to the presumed fat: (all plots made quick-and-dirty from the supplemental information)

This is counter-intuitive: short-term changes in body mass, after a diet change (that also, as stated in the text l. 168, the “Lasky et al. (in review)”, did not lead to a change in activity, i.e. exercise), are expected to be due to fat, not fat-free mass (i.e., muscle mass).

Your data even appears to suggest that body mass is INDEPENDENT of the magnitude of the fat mass within individuals! The males – variation in fat mass appears to be COMPENSATED by fat-free mass so that overall body mass stays the same.

I am not aware of a good explanation for what you appear to observe, but I think that either you need to dig into the methodology and potential problems, or you have to state clearly that you do not know an explanation, either. Writing, in the conclusion of the paper (l. 303) “The results of this study suggest that diet interventions can improve body composition by reducing adiposity and increasing FFM.” is like proactively ignoring the physiological puzzle your data represents and gives the impression that you either do not care about the physiology, or that you proactively try to “talk away” reasonable doubts by assertive language. This is an impression you want to avoid I think.

Response: We included more analyses and a new (Figure 2) regarding the relationship of body mass, FFM and body fat percentage, both for all individuals together and within individuals.

We included the following in the results section: “There was a strong positive relationship between FFM and body mass (Pearson's product-moment correlation: $t = 26.10$, $df = 22$, $P < 0.0001$, $R^2 = 0.96$; Fig. 2A), and a weaker and negative relationship between % body fat and both FFM ($t = -3.17$, $df = 22$, $P = 0.004$, $R^2 = 0.31$) and body mass ($t = -2.09$, $df = 22$, $P = 0.048$, $R^2 = 0.16$). There was no relationship between body mass and FFM within individuals M27 and F1611 (all $P > 0.39$; Fig. 2A), although there was a positive trend for M1770, F1771 and F1772 ($P = 0.042 - 0.089$; Fig.2A). There was no relationship between body mass and % body fat within individuals (all $P > 0.21$). There was a negative relationship between FFM and % body fat in males (all $P < 0.01$; Fig. 2B), but not in females (all $P > 0.15$; Fig. 2B).” (lines 235-243)

We included the following in the discussion: “When analyzing all individuals together, FFM increased when body mass increased. Within individuals we did not find this relationship, likely due to the restricted number of repeated measurements. When analyzing all individuals together, body fat percentage decreased when body mass increased but within individuals this relationship was

not evident. In males, but not females, there was a negative relationship between FFM and body fat percentage, where males with larger FFM had a lower body fat percentage. These results suggest that body mass of males is independent of body fat percentage, and variation in body fat percentage appears to be compensated by changes in FFM so that body mass of male elephants is kept relatively stable. At this point it is unclear if this is a general characteristic of male African elephants.” (lines 287-295)

The one measure that could fluctuate a lot (but not in the magnitude of the fat mass variation you have) is gut contents. Elephants may differ in the amount of food they actually eat (and this is something that your diet change most likely caused: an increase in total dry matter intake), and elephants are among those herbivores where increased food intake does not lead to a dramatic reduction in digesta retention time, i.e. increased food intake mostly leads to increased gut fill (increased actual gut volume) (Clauss et al. 2007). This is where I am not familiar enough with the literature on the use of isotope dilution method in large herbivores, how changes in gut fill, i.e. also in the amount of water “bound” in the gastrointestinal tract, affects FFM estimates. My guess is that gut contents, which contain water, are just part of the overall FFM that is calculated (and the factor used for calculation, here 0.746, is derived from measures that include gut content?). I just checked the paper you used as a source (Wang et al. 1999), and from reading that it becomes evident that the factor is derived in a way that makes it susceptible to changes in actual gut fill. Even if animals (for the derivation of the factor) had been investigated using the whole body (including gut contents, and not just the eviscerated carcass), the question is how they were fed before the analysis, i.e. was ad libitum intake variation on a variety of diets allowed, or not? I am not really into this literature, but the example of (Torbit et al. 1985) is nice, where this problem is specifically stated and the study design is explained in that first different amounts of food were given to the study animals (to create differences in body fat) but that then, when the actual isotope study was done to assess body composition, intake was made constant for all animals in order to make the isotope results comparable between the individuals. This is a factor of uncertainty in studies with live animals that needs to be assessed, and with a putative variation in gut fill in elephants of a magnitude of 7-17 % of total body mass (Clauss et al. 2005), I personally believe that variation in gut fill, if not controlled for, will have a large effect on estimates of FFM and FM in elephants. By incidence, the range is similar to the range in body fat you mention in l. 256 (5-17%), but I am not sure whether, after accounting for gut content in the TBW conversion, % translates into % in this comparison.

Yet, basically, this seems to tell me that if you wanted to really assess the effect of different dietary regimes in the same animal over time, the only way to get reliable (comparable) longitudinal data is to ensure that food intake is constant AT THE MOMENT OF APPLYING ISOTOPE BODY COMPOSITION. Otherwise, differences in food intake within the same animal between the time points, e.g. due to different temperatures (less intake in hot summer? higher intake in cold winter?), different feeding regimes (higher intake on less-energy-dense diet), will affect the signal you are interpreting due to the variation in intake (and hence gut fill) at the point of measurement.

Response: We do not have data on individual intake and thus variation in gut fill. However, we have added additional analyses and discussion to address this important point. See response to point 1, above.

Your method do not indicate any consistency as to the amount of food available a day prior to, and during, isotope application. It does not even indicate consistency with respect to the moment of dosing in relation to feeding events or feeding behaviour. I would suggest that you dig into that and assess whether it might relate to your data, and at least mention this option.

Response: We acknowledge that variation in gut fill is a factor that can influence measurements of body composition. We do not have detailed information on ingested food or amount of browse or other food items ingested on a daily basis. However, dosing of individuals always occurred in the morning (with the exception of a single event when dosing occurred at 13:40). The elephants were on habitat overnight and would come to the barn in the morning and have about two flakes of hay

(~3 kg) in the morning prior to the training session when dosing occurred. We calculated that we may have overestimated TBW by 1.4-3.4% assuming that gut fill varies between 7-17% body mass and that water content of hay is usually around 20% (if hay would have 25% water we would have overestimated TBW by 1.75-4.25%). Overestimating TBW will result in an overestimation of LBM and an underestimation of body fat percentage.

Assuming that elephants dosed later in the morning had more time to feed and a higher gut fill, we would expect that the time of dosing will influence our measurements of LBM, TBW, % body fat and water turnover. But we did not find any evidence for this in the data. Neither time of day nor the interaction between time of day and ID had a significant influence on any of the measurements.

It would also help if you had actual intake data of your animals – your paper on the composition of the new diet (Wood et al. 2019) at least suggests that some of this information should be available for the new diet in the method section: “Daily diets were recorded by NC Zoo keeper staff for the study duration. This included number of timothy hay (*Phleum pratense*) bales, enrichment food items, and browse species”, and from these data the averages given in Table 1, and the percentages of the ingested diet in Table 3, of that paper were calculated (but the actually ingested absolute amounts are not given in that paper per season). It would be ideal, answering the methodological issues, to link that information with the isotope data!

Response: We do not have detailed information on amount of browse or other food items (produce) given and ingested on a daily basis as this was not the aim of the study by Wood and colleagues.

Another factor of uncertainty might or might not be the variation in the blood sampling time points after isotope dosing (the methods seem to imply this was done a bit opportunistically, and not consistently across all assessments). Can this have an effect, or not? This should be mentioned. I also wonder whether a measure of fit for the slope intercept method of back-calculating the initial isotope enrichment (or dilution) should be given routinely to give a measure about the “confidence” of that estimate (on which all other results hinge). Personally, I would recommend to do that.

Response: The timing of sampling does not affect the rates of depletion or the intercept calculation unless there are substantial changes in physical activity. If there were, we would see this in the fit between the time vs ln-isotope enrichment regression. Instead, the median r^2 value for the regression between time and ln-enrichment was 0.997; the minimum r^2 was 0.990. The error resulting from variation in the timing of samples and thus the slope of the depletion rate is therefore negligible.

We did, however, endeavor to collect samples as close to 24h intervals as possible. We included this information and details regarding variation in deviation from multiples of 24h in the following way: “Blood samples (5ml) were drawn via venipuncture once prior to deuterium administration and then 3 or 4 times over the subsequent 10 days, at as close as feasible to multiples of 24 h. The range of absolute deviation from multiples of 24 h was 1–327 min (25th percentile=22.5 min; median=40 min, 75th percentile=80.75 min).” (lines 122-125)

We included two additional references showing that the slope intercept method is the preferred method to calculate dilution spaces for deuterium and oxygen-18 (Pontzer 2018a, Berman et al 2020). We also provide information regarding the fit for the slope intercept method: “Time of sampling and ln-isotope enrichment regression were strongly correlated (median $R^2 = 0.997$; range 0.990 – 1.000).” (lines 134-135)

Patterns in the data that are not mentioned, and may or may not be helpful in identifying physiological explanations:

Within individuals, there seems to be a negative correlation – is this something one expects? Does it equate to something like: better hydration status (more body water) = higher throughput because more is drunk? But that would sound like an instantaneous reaction, not something evident on an integrated measure of what happens on several days? Note that in my stats program (SPSS), the within-individual correlations were mostly not significant, though, in spite of the visual pattern. The same pattern is seen in your Fig. 2A

Response: The relationship between TBW and water throughput within species is not very strong. With only a few data points per subject, and mainly non-significant within-individual correlations, it is difficult to interpret this pattern.

However, this suggests that the variation in water throughput (which we discuss in terms of air temperature) is not just an artifact of the water in the digestive tract. If the amount of water in the body was artificially high (due to water in the digestive tract), it would in turn make the dilution space N artificially high. If this was the case, the result would be an inflated $N*k$ (k = isotope depletion rate) value, which is how water throughput is calculated. Therefore, if there would be no variation in k , and all the apparent variation in water turnover would be due to noise in N , then N and water turnover would be positively correlated. And, since N is used to calculate TBW, there would be a positive relationship between TBW and water turnover within individuals, which is not the case in our data. Thus, water turnover is not simply a function of noise or error in N due to variation in water in the digestive tract.

Why does higher water throughput seem to be correlated with less body fat? This effect tended towards significance in a GLM (random factor individual significant, water throughput $p=0.073$). The outlier would require explanation.

Response: When running a LMM (% body fat is normally distributed: Shapiro-Wilk normality test: $W = 0.95409$, p -value = 0.3315) that included ID as random factor, water throughput was not significantly related to body fat percentage in our data ($P= 0.34$).

In particular, I would, as a reader, like to know how a decrease in gut fill would affect the results and an increase in water intake, and how the combination of these factors (if we claim that in summer, animals eat less yet drink more, both in absolute terms) would influence any estimates of FFM and FM.

Response: Water was available ad libitum and we cannot assess the variation in how much water was drunk and thus, we do not want to speculate on its effect on measurements on body composition. We included a long section in the discussion elaborating how variation in gut fill can

affect measurements of body composition: “Another limitation is the fact that we did not control for variation in food intake and the resulting variation in gut fill, which can vary from 7% to 17% of body mass in elephants (Clauss et al. 2005). Dosing with DLW always occurred in the morning (with the exception of a single event when dosing occurred at 13:40). Elephants were on habitat overnight and would come to the barn in the morning where they ate one to two flakes of hay prior (roughly 3 kg) to being dosed with DLW. Due to water bound in gut content, we may have overestimated TBW by 1.4 - 3.4%, assuming that gut fill varies between 7 - 17% body mass and that water content of hay is typically around 20%. TBW of the study elephants ranged from 1640 – 4430 L and thus, an overestimation of 1.4% would equal 43 L (range: 23 – 62 L) on average and an overestimation of 3.4% would equal 104 L (range: 56 – 151 L), on average. Overestimating TBW by 3.4% will result in an overestimation of LBM by ~3% and an underestimation of percent body fat by ~3%. Thus, the measured changes in body fat percentage within ~3% of one another must be interpreted with caution, as they are within the range that changes in gut fill could affect. However, we did not find that the time of day when dosing with DLW occurred or the interaction between time of day and ID had a significant influence on the measurements of LBM, TBW, body fat percentage and water turnover (ANOVAs: $P > 0.47$), which we would expect if elephants dosed later in the day had more time to feed and thus a higher gut fill. Moreover, if the amount of water in the body was artificially high (due to water in the digestive tract), it would in turn make the dilution space N artificially high, which would result in an inflated $N \cdot k$ (k = isotope depletion rate) value, which is how water throughput is calculated. Therefore, if there would be no variation in k , and all the apparent variation in water turnover would be due to noise in N , then N and water turnover would be positively correlated. And, since N is used to calculate TBW, there would be a positive relationship between TBW and water turnover within individuals, which is not the case in our data. Thus, water turnover is not simply a function of noise or error in N due to variation in water in the digestive tract. Further, we note that measured changes in fat percentage exceeded 4% in most subjects, and that these changes are likely too large to be explained by variation in gut fill.” (lines 322-347)

3. details

I. 17 whichever way you word these findings, please ensure you do not use the words “during the first two years” because three spot samples need not represent the full time span.

Response: We changed the sentence to better reflect the results: “Within the first year after the dietary intervention, there was a reduction in body fat percentage and an increase in FFM. However, final values of both body fat percentage and FFM were similar to initial values.” (lines 14-15)

I. 19 I always recommend to refrain from superlatives or “we are the first to”. If you want to keep your superlative, you cannot give the averages because (Chusyd et al. 2018) had an animals with 489 L/d, higher than the average you mention with the superlative. You need to mention your one individual value of 553 L/d that was the only one in your dataset that was higher than the maximum in the other study.

Response: We deleted the superlative and changed the sentence in the following way: “Water throughput (males: 359 ± 9 L/d; females: 241 ± 28 L/d) was consistent with the allometric scaling of water use in other terrestrial mammals.” (lines 16-17)

I. 159 please state whether this is gross energy, digestible energy, or what other kind of energy you are indicating, and whether the unit is kg as fed, or kg dry matter.

Response: We included this information and changed the sentence in the following way: “The grain-free pellet has higher crude fiber (30% versus 22%) and lower digestible energy content (2,595 kcal/kg versus 3,030 kcal/kg as fed).” (lines 178-179)

I. 171 I think the group level visualisations should only include those datapoints for which all data of the group are available; otherwise, a trend is implied that is not based on all individuals

Response: We agree that this would be ideal. However, following this suggestion would result in the inclusion of only 2 study periods. However, we changed the figure to highlight differences in sample size per study period so the reader always knows how many datapoint were used to depict the group-level trends.

Fig. 2A the intra-individual trend should be mentioned in the text (and possibly explained).

x-axis label should be kg

Response: We mention the intra-individual trend and corrected the axis title: “In one individual (F1611), water throughput was negatively correlated with body mass ($t = -26.07$, $df = 1$, $P = 0.02$, $R^2 = 0.98$), but this relationship was not found in the other individuals (all $P > 0.05$).” (lines 251-253)

I. 247-248 with the few measurements you have, I think you should refrain from qualitative language like “transient” and “returned” because the stability of the initial state is not ascertained in your data, and continuous measurements might indicate a similar fluctuation just as well as “stability” as implied by the word “return”

Response: We removed qualitative language and change this sentence to: “However, there was no difference between initial and final measurements of body mass and body composition.”

I. 263 ff the factor would have to be even higher if more gut contents were assumed, right? But see major discussion above. I. 271 points into the same direction. If the animal had a very high intake at the time, this could explain why the conversion value was too low – I think (I hope you can come up with a comprehensive explanation of the effect of intake).

Response: We have included a paragraph to the discussion to elaborate how variation in gut fill can influence measurements of body composition. Please see responses above.

I think in the acknowledgements it should be “Jeb” or similar, not “Jb”?

Response: It is an unusual name but his middle name which he goes by is actually just the two letter Jb.

Reviewer: 2

Comments to the Author(s)

The authors aimed to (1) investigate how changes in diet altered body mass and body fat in five zoo African elephants, with measurements made once or twice a year over a three-year period, and (2) to investigate the effects of body size and ambient temperature on elephant water budgets. While I understand the difficulties of working with elephants, the study is limited by the small sample size and weak methodology, particularly for the change in diet (no details of exact diet composition or intake by the elephants is given - see my comments below). Despite aiming to give the elephants “an improved diet”, their body mass and body fat were the same at the end of the study, changing only transiently. There is no information on exactly how dietary energy intake changed over time. There is no information on other factors that could have influenced the elephants’ welfare and body mass, for example, activity level, exposure to heat or cold, disease, social interactions. The effect of temperature on water balance is assessed using data from a weather website, not taking into account the actual microclimate that the elephants would have experienced, for example, by being exposed to solar radiation (or the effect of other factors like activity level). Overall, the study provides few new findings and only tentative data from the small sample. Indeed, the authors recognise that and temper almost every conclusion they draw in the Discussion with a limitation and suggestion for future study. It is well known that elephants have a high water turnover and that increasing environmental heat load will increase that water turnover, and that body size influences water turnover. Unfortunately, I do not think this study provides a significant advance to the literature.

Specific comments:

General: I think that the Introduction is long and its focus could be improved. It jumps from studies of zoo elephant welfare to the physiology and activity of wild elephants, to ideas about evolved ecology, then back to zoo management strategies. I would suggest a brief introduction to wild elephant physiological ecology (including diet, water needs and activity), then moving the focus to the zoo and how best to manage elephants in that setting, before identifying the gap for the study and the study approach. It can also be made more concise by removing unnecessary detail, for example, “Morfeld and colleagues (10) developed a 5-point body condition score (BCS) to visually assess whether elephants are under- or overweight. Higher scores indicate greater body fat and were correlated with a biological marker of adiposity (serum triglycerides; 10,23). More than 70% of the 240 elephants in their study had a BCS of 4 or 5, and were considered overweight or obese”, could be reduced to “elephants with a higher body condition score had higher serum triglycerides, a marker of adiposity, and more than 70% of elephants were considered overweight or obese”.

Response: We re-structured the introduction following these suggestions.

Comments per line:

8: This comment would apply to all animals.

Response: We changed “African elephants” to “Animals”

13. Data on reduced dietary density is not available in the paper, as far as I can tell.

Response: The nutritional and energetic specifications of the different diets have been published prior to this study (Wood, J. et al. 2019. Analyses of African elephant (*Loxodonta africana*) diet with various browse and pellet inclusion levels. - Zoo Biol.: 1–14), and therefore these data are not presented here again. However, we cite the study by Wood and colleagues and state that the change in diet “increased browse and reduced the pellet component and energy content” (line 80)

44-46: Elephants do not only achieve evaporative water loss by wetting their skin. There also is substantial transfer of water across a permeable integument, as described by Dunkin et al.

Response: We changed the sentence in the following way: “In addition to the daily demands of water for normal physiological function, elephants use water for evaporative cooling by wetting their skin and via transfer across their skin (Dunkin et al. 2013, Mole et al. 2016).” (lines 33-34)

56: Throughout the paper (including the title), please specify what temperature you are referring to. The reader could think it is body temperature. Environmental temperature also can be measured in various ways (e.g. dry-bulb, wet-bulb, soil, black globe temperatures). Note also that dry-bulb air temperature alone does not describe the heat load experienced by animals outdoors.

Response: We clarified throughout the manuscript that we are referring to air temperature.

79: I don't think the animals in this study were not monitored daily?

Response: This is correct. We exchanged “daily” with “regularly”.

82: What dietary changes? A diet with lower energy content, different nutritional content?

Response: We clarified that energy content was reduced and which dietary changes occurred: “dietary changes that decreased energy content” (lines 74-75) and “We predicted that changes in diet that increased browse and reduced the pellet component and energy content of the diet” (lines 79-81)

87-89: Is the prediction based on different energy content for the two food types or is the energy content matched?

Response: We clarified that energy content was reduced and which dietary changes occurred: “dietary changes that decreased energy content” (lines 74-75) and “We predicted that changes in diet that increased browse and reduced the pellet component and energy content of the diet” (lines 79-81)

98: How did you account for one elephant being younger and still growing?

Response: To assess if including an individual in its growth phase influenced our results, we conducted all analyses again excluding this female. There were almost no differences and thus, we report results including this individual and mention results that changed after excluding this individual. (lines 217-220)

99: Please explain here or elsewhere in the Methods how the elephants were weighed, and the accuracy of the measurement. Please give the initial mass of all elephants here in the Methods, particularly since relationships with body mass are investigated in this study.

Response: We included this information in the text: “Elephants were weighed monthly via a digital scale (Avery Weigh Tronix Model 640; \pm 2.3 kg) which is covered by a platform. Elephants were asked to step on the scale and the weight was taken by an experienced keeper that ensures that all four feet are on the platform. The scale is tested and serviced regularly. At the beginning of the study elephants weighed 2556 – 5785kg.” (lines 93-97)

100-103: Given that the effect of environmental temperature is a key aim of the study, much more detail is required to describe exactly how the elephants were housed. For example, did the animals have access to shade, were they exposed to open skies at night, were the animals able to choose their own microclimate or were they herded in and out of the barn?

Response: We included more detailed information regarding housing conditions: “These elephants are primarily housed on two 1.4 ha habitats and are generally given access to the habitats throughout the day and the night. On the habitat, they have access to tree shade, cool pools, and mud wallows, and the habitat includes areas where trees block the wind. When air temperatures are below 4 °C and occasionally overnight, elephants are housed in or given access to a 537m² barn with four attached outdoor paddocks (237 m² each). At the barn, they can move between paddocks and stalls, unless air temperatures dropped below -1 °C, in which case they were secured in the barn for warmth.” (lines 97-103)

103-105: Exactly how were the five study animals housed? All together, or in two groups?

Response: We included the following additional information: “Social structure of the group varies, and animals are housed by themselves (for males), in pairs (for females) or in groups of 3 (two females, one male), 4 (all females), or 5 (4 females, one male).” (lines 104-106)

117: Please explain how elephants were restrained for blood sampling and where blood samples were taken from.

Response: We included additional information concerning blood draws: “Blood was collected from a vein on the back of the ear using a 21- or 19-gauge needle. Animals were not restraint during this procedure. They were asked to present their side parallel to the bollards and to stick their ear through an ear portal. The ear was cleaned with warm water and then the area of the blood draw was disinfected (using Novasan or alcohol) prior to the blood draw. Animals are able to decide if they want to participate in the blood draw or not.” (lines 125-129)

119: Were the plasma samples kept at a particular temperature?

Response: We included this information: “Samples were centrifuged immediately after collection to separate the plasma and plasma samples were stored at -80 °C.” (lines 129-131)

145: Humidity (in the subtitle) is not measured/reported.

Response: We corrected the subtitle and deleted “humidity”.

146: Presumably mean daily temperature refers to mean daily air temperature? What evidence is there that mean daily air temperature is suitable to quantify the heat load on elephants in the study?

Response: We included more information regarding the use of air temperature and how it relates to black globe temperatures: “We used air temperature as a crude estimate of ambient temperature that elephants experience in shaded locations. Studies conducted on African elephants in southern Africa found that air temperature measurements track black globe temperatures, which provide an integrated measure of air temperature, radiant temperature, and the cooling effect of wind (Hidden 2009, Mole et al. 2018).” (lines 162-166)

147: How far is this weather station from the study site?

Response: We included the distance (27km) between the two sites. (line 162)

148-149: It is not clear what is meant here? Did you use mean, minimum and maximum daily air temperatures, or did you use the mean daily temperature you obtained and calculate the maximum of the mean, etc? Is the measurement period the 10 days over which blood samples were taken?

Response: We provide additional information to clarify what we calculated: “We calculated average maximum, minimum, and mean air temperature for each 10-day measurement period using reading of daily maximum, minimum and mean air temperature. We also calculated mean dewpoint and cumulative precipitation over each 10-day measurement period.” (lines 166-169)

151: Was the actual feeding by the elephants observed? Even if the provided food was changed, how do you know how the energy intake and dietary content of each elephant changed? How much natural vegetation was provided, what was it exactly, and what was the nutritional content of this vegetation? As a main aim of the study was to investigate how dietary changes affect body mass, I would expect to see much greater detail on the nutritional content of the food and detail on actual food (and energy) intake by study elephants.

Response: Elephants were not observed closely for dietary consumption other than the dry feed. We noted this in the methods section: “However, data on dietary consumption per individual is not available.”

We also discuss how variation in intake, and thus gut fill, can affect our measurements; see comments to Reviewer 1 above.

Elephants have access to grass and trees on the habitat.

165: Please provide a brief description of how the behavioural observations were undertaken.

Response: We did scan sampling sporadically on the elephants during this study period. The full methods are laid out in the Lasky study. We included more information here: “Behavioral observations, performed as sporadic scan sampling between 9:00 – 17:00 during January 2015 – November 2016,” (lines 166-190)

177: Dewpoint is not a component of ambient temperature.

Response: We change the text to: “To examine how air temperature (average, maximum, and minimum) and dew point...” (line 200)

193: The diet change occurred in February 2016, so why is the “mass over the first two and a half years of diet change” calculated as the “average increase Aug 15 – Feb 17”? That average includes both diets.

Response: We corrected this sentence and the following to reflect changes in mass and body composition after the diet change: “Elephants in this study gained body mass over the first year of diet change (average increase Mar 16 – Feb 17: 324 ± 165 kg). This increase in body mass was largely the result of increased FFM (average increase Mar 16 – Feb 17: 593 ± 385 kg; Fig. 1). Body fat percentage generally decreased during this period (-6.0 ± 5.4 %), but considerable variation was evident among subjects (Fig. 1).” (lines 224-228)

250-251: What data do you have to support the claim of an improved diet? Improved in what way?

Response: We included additional information explaining how diet was improved: “Nonetheless, benefits of an improved diet, in this case a reduced amount of grain and caloric intake (Wood et al. 2019), can include a wide range of physiological measures, including improved cardiovascular health, even in the absence of permanent changes in body composition (Pontzer 2018b), ” (lines 282-285)

250-252: This is speculation, especially for these study animals. Also, if animals were restrained for blood sampling, why were other measures of physiological health not obtained?

Response: Elephants were not restrained during blood sample collection. We clarified this: “Animals were not restrained during this procedure. They were asked to present their side parallel to the bollards and to stick their ear through an ear portal. The ear was cleaned with warm water and then the area of the blood draw was disinfected prior to the blood draw. Animals are able decide if they want to participate in the blood draw or not.” (lines 125-129)

We did not collect additional blood volume to measure physiological measures and thus cannot assess changes in these measures here. We changed the sentence to reflect that changes in diet can improve physiological measures, even in the absence of permanent changes in body composition: “Nonetheless, benefits of an improved diet, in this case a reduced amount of grain and caloric intake (Wood et al. 2019), can include a wide range of physiological measures, including improved cardiovascular health, even in the absence of permanent changes in body composition (Pontzer 2018b).” (lines 282-285)

259: What information is available on reproductive cycling in the study animals at the times of measurements?

Response: We included information regarding the reproductive status of females: “None of the females was pregnant or lactating during the study.” (line 93)

286: Dunkin et al. have shown how skin permeability changes seasonally resulting in much greater evaporative water loss across the skin in summer.

Response: We included this information in the discussion: “Water requirements increased with higher environmental air temperatures, most likely due to increased insensible water loss (Dunkin et al. 2013, Mole et al. 2016). Permeability of elephant skin changes across seasons, and epidermal permeability is greater in summer compared to winter, allowing a higher rate of evaporative cooling when air temperatures are high (Dunkin et al. 2013).” (lines 349-352)

288-289: Daytime air temperatures can frequently exceed 40 deg C in the African elephant’s natural habitat. Importantly, the animals also experience a high heat load from radiation.

Response: We changed the sentence in the following way: “In the wild, where daily activity levels are greater, daytime temperatures exceed 40°C and animals experience a high heat load from solar radiation, water requirements are likely even greater and habitat and climate specific (Dunkin et al. 2013).” (lines 354-356)

291: Why is 10% body mass considered critical? Many large mammals can cope with larger percentage losses. Please provide a reference for 10% loss being critical for elephants.

Response: “We provide a reference (Feldhamer et al 2007) for the statement that 10% body water loss is critical. In the reference (p/ 180) it is stated that “most species die when they lose 10-20% of their body water”. (line 358)

299: Many elephants do not have migration routes available to them and in reserves would not be in direct conflict with humans for water.

Response: We included an additional sentence to incorporate this argument: “However, a large proportion of African elephants live in national parks and reserves that provide access to water at water holes, making conflict with humans over water less likely for these populations.” (lines 368-370)

303-304: I don't think this conclusion is valid given that the animals returned to initial values.

Response: We changed the sentence in the following way: “The results of this study suggest that diet interventions can influence body composition, and that maintaining such changes remains a challenge.” (lines 372-373)

Figure 1: It would be helpful in the elephant ID key to know which elephants were male and female.

Response: We included individual sex into the ID key.

Appendix C

RSOS-201155

Air temperature and diet influence body composition and water throughput in zoo-living African elephants (*Loxodonta africana*)

Pontzer et al.

reviewed by Marcus Clauss, Zurich (does not do anonymous reviews)

The revised version has addressed the majority of issues I had with the original version, except one.

I always hate reviewers being stubborn myself, which is what I am being here, but I try to word my points in a different way now that I hope will make clear why some of the results should be interpreted in a different way (and if you would do so, they would make physiological sense). My concern is not with the water turnover estimates (even if the discussion part on that might need re-calculation, I do not think it will change substantially), but the body composition part.

The major physiological issue was not addressed in the revised manuscript, and I first re-iterate here my previous comment:

Physiological logic of data and links to method?

The main finding in the data, as they are, is against the intuitive logic of rapid body mass changes: in the data, variation in body mass is basically only due to changes in presumed fat-free mass, and not at all due to the presumed fat:

(all plots made quick-and-dirty from the supplemental information)

This is counter-intuitive: short-term changes in body mass, after a diet change (that also, as stated in the text l. 168, the “Lasky et al. (in review)”, did not lead to a change in activity, i.e. exercise), are expected to be due to fat, not fat-free mass (i.e., muscle mass).

Your data even appears to suggest that body mass is INDEPENDENT of the magnitude of the fat mass within individuals! The males – variation in fat mass appears to be COMPENSATED by fat-free mass so that overall body mass stays the same.

This finding remains dramatically counter-intuitive, and in my view, it is not presented clearly as such in the revised manuscript, but stays hidden in the text, with an inconclusive statement l. 296: “variation in body fat percentage appears to be compensated by changes in FFM so that body mass of male elephants is kept relatively stable. At this point it is unclear if this is a general characteristic of male African elephants”.

I think this is partly my fault because I was not clear enough last time.

I need to put it in other words: yes, body mass changes mainly due to FFM can be expected in elephants, if FFM includes gut contents (as it does, see below).

When would we expect an increase in gut contents in elephants?

Especially when they are changed from a high-calory-diet to a low-calory-diet. This is also reflected in the increase in roughage and the increase in browse after the diet change.

So, what would I predict after such a diet change?

- in growing animals: a reduction in body fat % (of total body mass, and of fat-free body mass incl. gut contents, and of fat-free tissue mass excluding gut contents) as well as a continuous increase in fat-free tissue mass (muscles, bones etc. that grow)
- in animals not growing at all: a reduction in body fat % (of total body mass, and of fat-free body mass incl. gut contents, and of fat-free tissue mass excluding gut contents) but no change in fat-free tissue mass
- in ALL animals: an increase in gut fill, because they will eat more of the lower-quality diet after an adaptation period (where they may rely more on adipose reserves); this means both an increase in dry matter contents of the gut (and this dry matter content is more or less fat-free), and an increase in water bound in the gut (and this is fat-free). Total FFM (including gut contents) should therefore increase in all animals until a new equilibrium with the new diet is reached.

How do you estimate the amount of water bound in gut contents? Not by the water concentration of the food (the logic laid out in l. 331) – water concentration in gut contents is usually much higher than what is found in the food. Animals eating hay will not have gut contents with the water concentration of hay (or there would be no saliva, no secretions) ... The water concentration of gut contents is typically between 80-90 % depending on the section of the gut one is looking at – for free-ranging elephants: (Clemens and Maloiy 1982). But imagine that during the process of digestion, at the end of the gut, water is re-absorbed in the process of faecal bolus formation – and the faeces of hay-fed zoo elephants still contain 80% water (Clauss et al. 2003), i.e. it is reasonable to assume that gut contents are higher, as faeces are typically drier, not moister, than overall gut contents!

I think this information must inform the calculations about potential estimates of total body water in section l. 321-350.

If one checks out the data on water concentration of gut contents (e.g., (Clemens and Maloiy 1982)) and faeces (e.g., (Clauss et al. 2003)), one can see that this is typically not in the area of 73 or 75 %, but more. I might be wrong here, but doesn't this mean that, assuming FFM is composed of many things incl. gut content, that it is over-estimated if there is a relevant portion of total body mass (in the area of 7-17%) whose water contents are not 0.74, but something like 0.85?

And if FFM is over-estimated in that way, would that make fat estimates too low – incl. the one negative value?

So following the logic in l. 330, the over-estimation of FFM, and the under-estimation of body fat, is not 1.4-3.4%, but 6.0-14.5%. This is basically the range of body fat measured in the study, meaning the fat values might in theory, assuming our maximum gut fill range, be double of what is estimated, right?

My point is, in my view the whole narrative of body composition changes is flawed if one disregards gut contents. And given the change in diet regime, we would automatically expect an increased intake which – especially in elephants – does not translate into a quicker passage (due to a fixed gut fill volume) but on the contrary passage is kept stable (which means gut fill volume must increase) (Clauss et al. 2007). (don't cite all this my stuff, this is just for explanation).

So, writing as if FFM represented body tissue and not the combination of body tissue and gut contents remains, to me, the major flaw in the manuscript, and I apologize for not pointing

this out more specifically in the first review. I actually think that once one would define FFM in this particular manuscript as “including gut contents”, then the whole narrative would make sense.

By the way, you added this statement l. 142-148 about the Wang paper and the low coefficient of variation you find in the data. But look at that data in the Wang paper – of the 15 species, 4 are clearly indicated as “carcass”, i.e. without gut contents. Among the remaining 11, you have several that are carnivores, so one can assume – e.g. in seals dissected for disease or after a period of non-feeding – that their guts are empty anyhow, even if they are labelled “whole body”. For the cattle, a paper from 1920 is cited by Wang et al., and if you check it out, it becomes clear that this is not even the carcass but either only “flesh”, or “flesh”, liver and blood, again excluding gut contents that were analysed there. For the sheep and goats, a paper is cited whose abstract says that the animals were fasted – no food, no water! – for 48 hours! – so again, the gut contents would not be representative of live herbivores. For the baboon, a paper is cited in which the “total body weight” is actually defined as gut-contents-free, so the label “whole body” is not appropriate ... so resting the case of disregarding gut contents on the Wang et al. paper is not valid.

On the other hand, there are papers that state the relevance of considering gut fill in double-labelled water measurements: (Torbit et al. 1985).

The prediction that body fat is underestimated if gut contents are not taken into consideration is supported in the literature, e.g. a study with horses (Dugdale et al. 2011). By the way, in that study, without commenting on the feeding regime prior to deuterium application (except that most animals received the same hay), the equilibration time was without access to food and water.

“The variability in the gastrointestinal (GI) contents of ruminants can result in significant errors in prediction of body composition from estimates of total body water (TBW). Byers (7) developed a two-compartment model for steers to differentiate EBW and water associated with the GI contents. For the D2O technique to be applicable for ruminants, procedures are needed to differentiate GI water from EBW” (Andrew et al. 1995).

These literature findings cannot be explained away by citing Wang et al., and should be included in discussion about the interpretation of the findings.

Again, just calling it “FFM (including gut contents)” and keeping all else in estimation methods as it is, and then demonstrating in the discussion how, depending on 80-90% moisture in gut contents and gut contents of 7-17% of body mass, the results may change fat estimates (by overestimating FFM), and most particularly, linking this to the expected change in gut contents due to the diet change, would all make perfect sense to me.

The information about the amount of hay bales given would benefit from a more accurate description of the hay bales:

Rectangle Bales

Size	Dimension(L x H x W)	Weight
2 stringed bale	36 inches x 19 inches x 16 inches	40 to 75 lbs.
3 stringed bale	44 inches x 22 inches x 15 inches	100 to 140 lbs.
Half ton	6 ft x 4 ft x 3 ft	1000 lbs.
1 Ton	8 ft x 4 ft x 4 ft	2000 lbs.

Round Bales

Size	Dimension(Width x Height)	Weight
Small	4 ft x 4 ft	400 to 600 lbs.
Medium	5 ft x 4.5 ft	720 to 950 lbs.
Large	5 ft x 6 ft	1270 to 1700 lbs.

1. 340 gut fill in herbivores is not only a function of a single meal, but of the overall intake level over the preceding week. The analysis is good to include, but one should not think that that excludes variation in gut fill due to what they ate the previous days, or the days between dosing and sampling. To keep gut fill constant, it is not so decisive what happens in the few hours prior to dosing, but during a whole 14-d-interval. This is why digestion studies with herbivores are so tedious – one cannot rely on 2-3 day saympling intervals to have a constant signal!

Sincerely marcus clauss

Literature cited:

- Andrew SM, Erdman RA, Waldo DR (1995) Prediction of body composition of dairy cows at three physiological stages from deuterium oxide and urea dilution. *Journal of Dairy Science* 78:1083-1095
- Clauss M, Löhlein W, Kienzle E, Wiesner H (2003) Studies on feed digestibilities in captive Asian elephants (*Elephas maximus*). *Journal of Animal Physiology and Animal Nutrition* 87:160-173
- Clauss M, Streich WJ, Schwarm A, Ortmann S, Hummel J (2007) The relationship of food intake and ingesta passage predicts feeding ecology in two different megaherbivore groups. *Oikos* 116:209-216
- Clemens ET, Maloiy GMO (1982) The digestive physiology of three East African herbivores - the elephant, rhinoceros and hippopotamus. *Journal of Zoology* 198:141-156

Dugdale AHA, Curtis GC, Milne E, Harris PA, Argo CM (2011) Assessment of body fat in the pony: Part II. Validation of the deuterium oxide dilution technique for the measurement of body fat. *Equine Veterinary Journal* 43:562-570

Torbit SC, Carpenter LH, Alldredge AW, Swift DM (1985) Mule deer body composition: a comparison of methods. *Journal of Wildlife Management* 49:86-91

Appendix D**ROYAL SOCIETY
OPEN SCIENCE****Air temperature and diet influence body composition and
water throughput in zoo-living African elephants
(*Loxodonta africana*)**

Journal:	Royal Society Open Science
Manuscript ID	RSOS-201155
Article Type:	Research
Date Submitted by the Author:	29-Jun-2020
Complete List of Authors:	Pontzer, Herman; Duke University Rimbach, Rebecca; Duke University, ; University of the Witwatersrand, Paltan, Jenny; Hunter College Ivory, Erin; North Carolina Zoo Kendall, Corinne; North Carolina Zoo, ; North Carolina State University, Applied Ecology
Subject:	physiology < BIOLOGY
Keywords:	body fat, dietary intervention, stable isotopes, water turnover
Subject Category:	Organismal and Evolutionary Biology

Author-supplied statements

Relevant information will appear here if provided.

Ethics

Does your article include research that required ethical approval or permits?:

Yes

Statement (if applicable):

The North Carolina Zoo is an accredited zoo with the Association of Zoos and Aquariums, and institutional approvals for this work were obtained prior to the study through the zoo's Research Review Committee.

Data

It is a condition of publication that data, code and materials supporting your paper are made publicly available. Does your paper present new data?:

Yes

Statement (if applicable):

The dataset supporting this article has been uploaded as part of the supplementary material (Table S1).

Conflict of interest

I/We declare we have no competing interests

Statement (if applicable):

CUST_STATE_CONFLICT :No data available.

Authors' contributions

This paper has multiple authors and our individual contributions were as below

Statement (if applicable):

HP and CJK designed the study, analyzed data, and wrote the manuscript. JP and ELI assisted with data collection. JP, ELI, RR assisted in manuscript preparation and all authors gave final approval for publication.

Air temperature and diet influence body composition and water throughput in zoo-living African elephants (*Loxodonta africana*)

Herman Pontzer^{1,2}, Rebecca Rimbach^{1,3*}, Jenny Paltan⁴, Erin L. Ivory⁵, Corinne J. Kendall⁵

¹ Evolutionary Anthropology; Duke University, Durham NC USA

² Duke Global Health Institute; Duke University, Durham NC USA

³ School of Animal, Plant & Environmental Sciences, University of the Witwatersrand, Johannesburg, South Africa

⁴ Department of Anthropology; Hunter College, New York NY USA

⁵ North Carolina Zoo; Asheboro NC, USA

Keywords: body fat, dietary intervention, stable isotopes, water turnover

1. Summary

Animals face particular physiological challenges in captivity and the wild. Captive elephants can become over- or under-conditioned with inadequate exercise and diet management. Few studies have quantified body composition or water throughput in elephants, and none to date have examined longitudinal responses to changes in diet or air temperature. Using the stable isotope deuterium oxide ($^2\text{H}_2\text{O}$), we investigated changes in body mass, fat free mass (FFM), and body fat in response to a multi-year intervention that reduced dietary energy density for adult African elephants housed at the North Carolina Zoo. We also examined the relationship between air temperature and water throughput. Deuterium dilution and depletion rates were assayed via blood samples and used to calculate body composition and water throughput in 2 male and 3 female African elephants at 6 intervals over a 3-year period. Within the first year after the dietary intervention, there was a reduction in body fat percentage and an increase in FFM. However, final values of both body fat percentage and FFM were similar to initial values. Water throughput (males: 359 ± 9 L/d; females: 241 ± 28 L/d) was consistent with the allometric scaling of water use in other terrestrial mammals. Water throughput increased with outdoor air temperature. Our study highlights the physiological water dependence of elephants and shows that individuals have to drink every 2 to 3 days to avoid critical water loss of $\sim 10\%$ body mass in hot conditions.

*Author for correspondence (rimbach@gmail.com).

†Present address: Evolutionary Anthropology; Duke University, Durham NC USA

2. Introduction

[revised manuscript text omitted]

To date, most studies of elephant physiology and health have been cross-sectional in design
[but see 19]. An underutilized, but potentially powerful, complementary approach to improving

our understanding of elephant physiology and health is to measure longitudinal responses to
changes in diet, activity, or other environmental variables (such as precipitation and ambient air
temperature). Longitudinal studies require greater time investment, but allow comparisons within
subjects and can help isolate the effects of specific variables (i.e. environmental or intervention
strategies). In zoo settings, manipulations can be controlled and elephants monitored regularly to
assess effects. Prospective studies could be particularly useful for resolving the effects of diet and
activity on body mass and adiposity.

The aims of this study were to (i) examine the effects of dietary changes that decreased
energy content on body mass and body fat in zoo-living, adult African elephants, over a three-year
period and (ii) to assess the effects of body size and air temperature on elephant water budgets.
We measured adiposity and water throughput using deuterium dilution and depletion rates,
respectively. This minimally invasive approach provides precise measures of total body water
(TBW), fat free mass (FFM), fat mass (FM) and daily water throughput (L/d). We predicted that
changes in diet that increased browse and reduced the pellet component and energy content of the
diet would reduce body fat percentage among elephants. Further, we predicted that water
throughput among elephants would follow the body size scaling evident in other mammals, and
that warmer air temperatures would increase daily water throughput. In order to assess some of the
physiological challenges associated with their large body size, we compared daily water
throughput among elephants to that of other mammals and measured the effects of seasonal
changes in air temperature on daily water throughput.

**3. Material and Methods**

*Sample & Measurement Periods*

North Carolina Zoo (Asheboro, NC, USA) houses six African elephants (4F, 2M), all are adult
animals (aged 32 to 43) except for one female who was 13 years of age at the beginning of this
study. One adult female was not included in the study as it was not possible to weigh her during
the study period. None of the females was pregnant or lactating during the study. Elephants were
weighed monthly via a digital scale (Avery Weigh Tronix Model 640; ± 2.3 kg) which is covered
by a platform. Elephants were asked to step on the scale and the weight was taken by an
experienced keeper that ensures that all four feet are on the platform. The scale is tested and
serviced regularly. At the beginning of the study elephants weighed 2556 – 5785kg. These
elephants are primarily housed on two 1.4 ha habitats and are generally given access to the habitats
throughout the day and the night. On the habitat, they have access to tree shade, cool pools, and
mud wallows, and the habitat includes areas where trees block the wind. When air temperatures
are below 4 °C and occasionally overnight, elephants are housed in or given access to a 537m²
barn with four attached outdoor paddocks (237 m² each). At the barn, they can move between
paddocks and stalls, unless air temperatures dropped below -1 °C, in which case they were secured
in the barn for warmth. Elephants were housed in mixed social groups including male and female
animals. Social structure of the group varies, and animals are housed by themselves (for males),
in pairs (for females) or in groups of 3 (two females, one male), 4 (all females), or 5 (4 females,
one male).

Body mass, total body water (TBW), fat free mass (FFM), and daily water throughput (
[revised manuscript text omitted]

201
\pm SD.

To examine how air temperature (average, maximum, and minimum) and dew point and
body mass influence water throughput, we used linear mixed models (LMMs) with Gaussian error
structure using the package 'lme4' [42]. We log-transformed water throughput to reduce
heteroscedasticity when fitting the models, and we used individual ID as a random factor because
individuals were sampled repeatedly. We did not include sex as a predictor in the models because

there was collinearity between sex and mass. We included the interaction term between air
temperature and body mass, and removed this interaction due to non-significance. We standardized
(z-transformed) all numeric predictors for more accurate model fitting and to facilitate
comparisons of model estimates [43]. We verified the model by inspecting Q–Q plots and by
plotting model residuals against fitted values. We checked for multi-collinearity by calculating
variance inflation factors [44] for the predictor variables using the ‘vif’ function in the car package
[45], which did not indicate collinearity (all vifs < 2).

We used ANOVAs that included time of day, individual ID and the interaction between
time of day and ID to assess whether measurements of LBM, TBW, % body fat and water turnover
were influenced by the time of the day when dosing with DLW occurred. We used Pearson's
product-moment correlations to examine overall and within-individual relationships between body
mass, FFM and % body fat. One female was 13 years old at the onset of the study, and thus, was
still growing throughout the study [46]. To assess if including an individual in its growth phase
influenced our results, we conducted all analyses again excluding this female. There were almost
no differences and thus, we report results including this individual and mention results that
changed after excluding this individual.

42 225 **3. Results**

Individuals varied in body mass, body composition, and water throughput, with males
weighing considerably more and using more water (Table 1, Fig. 1). Elephants in this study gained
body mass over the first year of diet change (average increase Mar 16 – Feb 17: 324 ± 165 kg).
This increase in body mass was largely the result of increased FFM (average increase Mar 16 –
Feb 17: 593 ± 385 kg; Fig. 1). Body fat percentage generally decreased during this period ($-6.0 \pm$

5.4 %), but considerable variation was evident among subjects (Fig. 1). Final measurements of
 body mass, FFM and body fat percentage were similar to the initial values (paired t tests: body
 mass: $t = -0.058$, $df = 3$, $P = 0.957$; FFM: $t = -0.086$, $df = 3$, $P = 0.936$; % body fat: $t = 0.182$, $df =$
 3 , $P = 0.867$). Body fat percentage was significantly lower at the end of the study compared to
 initial values when excluding the young female ($t = 6.608$, $df = 2$, $P = 0.022$). Time of dosing with
 DLW and the interaction between time of day and ID did not influence measurements of LBM,
 TBW, % body fat and water turnover (ANOVAs: all $P > 0.47$).

There was a strong positive relationship between FFM and body mass (Pearson's product-
 moment correlation: $t = 26.10$, $df = 22$, $P < 0.0001$, $R^2 = 0.96$; Fig. 2A), and a weaker and negative
 relationship between % body fat and both FFM ($t = -3.17$, $df = 22$, $P = 0.004$, $R^2 = 0.31$) and body
 mass ($t = -2.09$, $df = 22$, $P = 0.048$, $R^2 = 0.16$). There was no relationship between body mass and
 FFM within individuals M27 and F1611 (all $P > 0.39$; Fig. 2A), although there was a positive trend
 for M1770, F1771 and F1772 ($P = 0.042 - 0.089$; Fig.2A). There was no relationship between
 body mass and % body fat within individuals (all $P > 0.21$). There was a negative relationship
 between FFM and % body fat in males (all $P < 0.01$; Fig. 2B), but not in females (all $P > 0.15$; Fig.
 2B).

**Table 1.** Key characteristics for the five elephants in this study. Means (\pm SD) are shown. Data
 249 per sampling event presented in Table S1.

Subject	Sex	Age	Body Mass (kg)	Total Body Water (L)	Fat Free Mass (g)	Body Fat %	Water Throughput (L/d)
1770	M	32-34	5815 \pm 190	4020 \pm 349	5389 \pm 467	7 \pm 6	353 \pm 43
27	M	41-43	5553 \pm 107	3698 \pm 177	4957 \pm 238	11 \pm 4	365 \pm 97
1772	F	13-15	2654 \pm 96	1731 \pm 79	2320 \pm 106	13 \pm 2	227 \pm 40
1611	F	33-35	4401 \pm 140	2817 \pm 131	3776 \pm 175	14 \pm 2	223 \pm 42
1771	F	37-39	3905 \pm 150	2587 \pm 142	3467 \pm 190	11 \pm 2	273 \pm 47

Daily water throughput was related to both body mass (Estimate \pm SD = 0.186 ± 0.055 , $t =$
3.345 , $P = 0.0008$; Fig. 3A) and mean daily air temperature (Estimate \pm SD = 0.141 ± 0.028 , $t =$
5.045 , $P < 0.0001$). Mean water throughput increased with mean body mass at a rate of
approximately 4.5 L/d for every 100 kg increase in body mass. In one individual (F1611), water
throughput was negatively correlated with body mass ($t = -26.07$, $df = 1$, $P = 0.02$, $R^2 = 0.98$), but
this relationship was not found in the other individuals (all $P > 0.05$).

Water throughput fluctuated seasonally, with lower throughput in the cooler months (Fig.
1). When mean daily air temperature ranged from 6 – 14 °C, water throughput was 325 ± 30 L/d
for males and 218 ± 38 L/d for females. During warmer measurement periods (mean daily air
temperature 23 – 24 °C) water throughput was 427 ± 89 L/d for males and 278 ± 18 L/d for females.
Measured as a percentage of TBW, water throughput increased at a rate of approximately 1% TBW
for every 4 °C increase in mean daily air temperature (Fig. 3B). Daily water throughput was also
related to maximum (Estimate \pm SD = 0.527 ± 0.229 , $t = 0.229$, $P = 0.032$) and minimum air
temperature (Estimate \pm SD = 0.715 ± 0.201 , $t = 3.548$, $P = 0.002$), and dew point (Estimate \pm SD
= 0.641 ± 0.212 , $t = 3.014$, $P = 0.007$). Males had a significantly larger body mass than females
(Estimate \pm SD = 2021.869 ± 675.103 , $t = 2.994$, $P = 0.002$), and body mass was weakly influenced
by mean daily air temperature (Estimate \pm SD = -8.474 ± 4.281 , $t = -1.979$, $P = 0.047$). There was
no relationship between body mass mean daily air temperature when excluding the young female
(Estimate \pm SD = -8.790 ± 5.190 , $t = -1.693$, $P = 0.09$).

Water throughput among adult African elephants was consistent with that of other
mammals. Mean water throughput for males (body mass: 5684 ± 200 kg), averaged across all
periods (359 L/d), was 17% above the regression for other mammals (Fig. 4). Mean water

[revised manuscript text omitted]

consuming and the killing of animals makes it ethically undesirable and likely explains why
TBW:FFM has not been assessed for elephants so far.

Limitations of this study include the small number of elephants measured and the relatively
narrow age range of subjects. Uncertainty in the ratio of TBW:FFM limited the accuracy of
isotope-based body composition measurements. This uncertainty might result in less accurate
absolute numbers, but since we used the same ratio for all individuals, longitudinal changes should
be accurately reflected. Another limitation is the fact that we did not control for variation in food
intake and the resulting variation in gut fill, which can vary from 7% to 17% of body mass in
elephants [51]. Dosing with DLW always occurred in the morning (with the exception of a single
event when dosing occurred at 13:40). Elephants were on habitat overnight and would come to the
barn in the morning where they ate one to two flakes of hay prior (roughly 3 kg) to being dosed
with DLW. Due to water bound in gut content, we may have overestimated TBW by 1.4 - 3.4%
assuming that gut fill varies between 7 - 17% body mass and that water content of hay is typically
around 20%. TBW of the study elephants ranged from 1640 – 4430 L and thus, an overestimation
of 1.4% would equal 43 L (range: 23 – 62 L) on average and an overestimation of 3.4% would
equal 104 L (range: 56 – 151 L), on average. Overestimating TBW by 3.4% will result in an
overestimation of LBM by ~3% and an underestimation of percent body fat by ~3%. Thus, the
measured changes in body fat percentage within ~3% of one another must be interpreted with
caution, as they are within the range that changes in gut fill could affect. However, we did not find
that the time of day when dosing with DLW occurred or the interaction between time of day and
ID had a significant influence on the measurements of LBM, TBW, body fat percentage and water
turnover (ANOVAs: $P > 0.47$), which we would expect if elephants dosed later in the day had
more time to feed and thus a higher gut fill. Moreover, if the amount of water in the body was

artificially high (due to water in the digestive tract), it would in turn make the dilution space N
artificially high, which would result in an inflated $N*k$ (k = isotope depletion rate) value, which is
how water throughput is calculated. Therefore, if there would be no variation in k , and all the
apparent variation in water turnover would be due to noise in N , then N and water turnover would
be positively correlated. And, since N is used to calculate TBW, there would be a positive
relationship between TBW and water turnover within individuals, which is not the case in our data.
Thus, water turnover is not simply a function of noise or error in N due to variation in water in the
digestive tract. Further, we note that measured changes in fat percentage exceeded 4% in most
subjects, and that these changes are likely too large to be explained by variation in gut fill.

Daily water throughput was high among African elephants, commensurate with their large
body masses. Water requirements increased with higher environmental air temperatures, most
likely due to increased insensible water loss [12,13]. Permeability of elephant skin changes across
seasons, and epidermal permeability is greater in summer compared to winter, allowing a higher
rate of evaporative cooling when air temperatures are high [12]. Water throughputs for males in
summer months were the highest ever measured for any animal, reaching 400 to over 500 L/d
(Table S1). In the wild, where daily activity levels are greater, daytime temperatures exceed 40°C
and animals experience a high heat load from solar radiation, water requirements are likely even
greater and habitat and climate specific [12]. With TBW accounting for ~67% of body mass (Table
1; and see [16]), and water loss in hot conditions in excess of 10% TBW, elephants would need to
drink every 2 to 3 days to avoid critical water loss of ~10% body mass [52]. This result fits well
with observations of wild elephants, which rarely drink less often than every 3-4 days in the dry
season [53]. Elephants travel faster in the dry season than in the wet season, likely trying to
minimize the time spent traveling between water and food sources [53].

Physiological water requirement provides additional context to the water-directed
behaviors and ecology of elephants in the wild [9,13] and highlights the dependence of elephants
on access to drinking water. Most parts of southern Africa are predicted to get drier and hotter in
the future, and arid areas are expected to expand further [10]. Such climatic changes should place
increased water stress on elephants inhabiting these areas which could affect migration routes and
result in conflict over water with human population (e.g. water used for agriculture). This would
likely negatively impact already vulnerable elephant populations. However, a large proportion of
African elephants live in national parks and reserves that provide access to water at water holes,
making conflict with humans over water less likely for these populations.

Longitudinal physiological studies provide an important perspective on captive elephant
health and well-being. The results of this study suggest that diet interventions can influence body
composition, and that maintaining such changes remains a challenge. Long-term monitoring of
body composition and other aspects of health, using stable isotopes and other minimally invasive
techniques, will continue to improve the care of elephants in human settings as well as shed light
on their physiological adaptations and ecology in the wild.

**Acknowledgements**

We thank North Carolina Zoo staff for their assistance with data collection. We thank Marcus Clauss and
one anonymous reviewer for their comments on earlier versions of the manuscript. Jb Minter provided
comments on an earlier version of this manuscript.

**Ethical Statement**

The North Carolina Zoo is an accredited zoo with the Association of Zoos and Aquariums, and institutional
approvals for this work were obtained prior to the study through the zoo's Research Review Committee.

**Author' Contributions**

1
2
3 391 HP and CJK designed the study, analyzed data, and wrote the manuscript. JP and ELI assisted with data

[revised manuscript text omitted]
_{10}\text{Water Throughput} = 0.924 \pm 0.023 \log_{10}\text{Mass} - 0.971 \pm 0.039$ (df = 40, adj. $r^2 = 0.97$, $P < 0.0001$).

172x109mm (96 x 96 DPI)

1
2
3 Reviewers' Comments to Author:

4 Reviewer: 1
5

6 Comments to the Author(s)

7 RSOS-192105

8 Temperature and diet influence body composition and water throughput in zoo-living African
9 elephants (*Loxodonta africana*)

10 Pontzer et al.
11

12
13 This study presents serial data on the body composition of 5 African elephants as measured by the
14 isotope dilution method. This is put into context with a diet change and environmental temperatures.
15

16 This represents an impressive setup and logistical achievement. There are two major areas of
17 comments that I have – the internal logic of the data and implications of the method in general, and
18 some scholarship with respect to more recent literature.
19

20
21 My overall recommendation is to make a completely new narrative focussing on the methodological
22 aspect – what do isotope dilution values tell us if we do not control for intake in animals, like
23 elephants, where a large variation in gut fill has been documented and is surely possible, especially
24 when assessing animals on different diets and at different seasons. Because the main group that has
25 used the method so far (Chusyd et al. 2018; Chusyd et al. 2019) has never mentioned this aspect (one
26 is tempted to say: used the method uncritically), such a contribution would be an enormous step
27 forward. See below under chapter 2. for details.
28

**Response: We do not have data on individual intake and thus variation in gut fill and therefore**
**cannot directly assess the effect of methodological differences in this paper. However, we agree this**
**is an important issue and we have revised the paper and analyses to examine, to the extent possible**
**given the data we have, the influence of feeding and gut fill on our results. We have included an**
**analysis of time of day, and other factors that could influence gut fill, on our results (we find no**
**evidence for a substantial effect in our sample). We also examine the issue of gut fill specifically in**
**the Discussion and model the degree to which it could affect our results. The paragraph states:**

**“Another limitation is the fact that we did not control for variation in food intake and the resulting**
**variation in gut fill, which can vary from 7% to 17% of body mass in elephants (Clauss et al. 2005).**
**Dosing with DLW always occurred in the morning (with the exception of a single event when dosing**
**occurred at 13:40). Elephants were on habitat overnight and would come to the barn in the morning**
**where they ate one to two flakes of hay prior (roughly 3 kg) to being dosed with DLW. Due to water**
**bound in gut content, we may have overestimated TBW by 1.4 - 3.4%, assuming that gut fill varies**
**between 7 - 17% body mass and that water content of hay is typically around 20%. TBW of the study**
**elephants ranged from 1640 – 4430 L and thus, an overestimation of 1.4% would equal 43 L (range:**
**23 – 62 L) on average and an overestimation of 3.4% would equal 104 L (range: 56 – 151 L), on**
**average. Overestimating TBW by 3.4% will result in an overestimation of LBM by ~3% and an**
**underestimation of percent body fat by ~3%. Thus, the measured changes in body fat percentage**
**within ~3% of one another must be interpreted with caution, as they are within the range that**
**changes in gut fill could affect. However, we did not find that the time of day when dosing with DLW**
**occurred or the interaction between time of day and ID had a significant influence on the**
**measurements of LBM, TBW, body fat percentage and water turnover (ANOVAs: $P > 0.47$), which**
**we would expect if elephants dosed later in the day had more time to feed and thus a higher gut**
**fill. Moreover, if the amount of water in the body was artificially high (due to water in the digestive**
**tract), it would in turn make the dilution space N artificially high, which would result in an inflated**
**$N \cdot k$ (k = isotope depletion rate) value, which is how water throughput is calculated. Therefore, if**
**there would be no variation in k , and all the apparent variation in water turnover would be due to**
**noise in N , then N and water turnover would be positively correlated. And, since N is used to**

calculate TBW, there would be a positive relationship between TBW and water turnover within individuals, which is not the case in our data. Thus, water turnover is not simply a function of noise or error in N due to variation in water in the digestive tract. Further, we note that measured changes in fat percentage exceeded 4% in most subjects, and that these changes are likely too large to be explained by variation in gut fill.” (lines 322-347)

1. Literature (some of this is so new that it is not to be considered a criticism, just a hint; this goes a bit into the direction of not only considering the American populations) drinking water intake in elephant (Benedict 1936) elephants in zoos are over-conditioned as compared to free-ranging populations: BCS (Schiffmann et al. 2018), body mass (Schiffmann et al. 2019b) longitudinal studies in elephant body condition: BCS on an individual and a population basis (Schiffmann et al. 2019a); BCS over age plotted in (Schiffmann et al. 2019b); body mass development over age indicating a putatively natural source of body mass cyclicity (Schiffmann et al. 2019c) factors relating to foot health (Wendler et al. 2019)

Response: Thank you for highlighting literature that was missing from the manuscript. We read these papers and cited them where appropriate.

2. physiological logic of data and links to method?

The main finding in the data, as they are, is against the intuitive logic of rapid body mass changes: in the data, variation in body mass is basically only due to changes in presumed fat-free mass, and not at all due to the presumed fat: (all plots made quick-and-dirty from the supplemental information)

This is counter-intuitive: short-term changes in body mass, after a diet change (that also, as stated in the text l. 168, the “Lasky et al. (in review)”, did not lead to a change in activity, i.e. exercise), are expected to be due to fat, not fat-free mass (i.e., muscle mass).

Your data even appears to suggest that body mass is INDEPENDENT of the magnitude of the fat mass within individuals! The males – variation in fat mass appears to be COMPENSATED by fat-free mass so that overall body mass stays the same.

I am not aware of a good explanation for what you appear to observe, but I think that either you need to dig into the methodology and potential problems, or you have to state clearly that you do not know an explanation, either. Writing, in the conclusion of the paper (l. 303) “The results of this study suggest that diet interventions can improve body composition by reducing adiposity and increasing FFM.” is like proactively ignoring the physiological puzzle your data represents and gives the impression that you either do not care about the physiology, or that you proactively try to “talk away” reasonable doubts by assertive language. This is an impression you want to avoid I think.

Response: We included more analyses and a new (Figure 2) regarding the relationship of body mass, FFM and body fat percentage, both for all individuals together and within individuals.

We included the following in the results section: “There was a strong positive relationship between FFM and body mass (Pearson's product-moment correlation: $t = 26.10$, $df = 22$, $P < 0.0001$, $R^2 = 0.96$; Fig. 2A), and a weaker and negative relationship between % body fat and both FFM ($t = -3.17$, $df = 22$, $P = 0.004$, $R^2 = 0.31$) and body mass ($t = -2.09$, $df = 22$, $P = 0.048$, $R^2 = 0.16$). There was no relationship between body mass and FFM within individuals M27 and F1611 (all $P > 0.39$; Fig. 2A), although there was a positive trend for M1770, F1771 and F1772 ($P = 0.042 - 0.089$; Fig.2A). There was no relationship between body mass and % body fat within individuals (all $P > 0.21$). There was a negative relationship between FFM and % body fat in males (all $P < 0.01$; Fig. 2B), but not in females (all $P > 0.15$; Fig. 2B).” (lines 235-243)

We included the following in the discussion: “When analyzing all individuals together, FFM increased when body mass increased. Within individuals we did not find this relationship, likely due to the restricted number of repeated measurements. When analyzing all individuals together, body fat percentage decreased when body mass increased but within individuals this relationship was

**not evident. In males, but not females, there was a negative relationship between FFM and body**
**fat percentage, where males with larger FFM had a lower body fat percentage. These results suggest**
**that body mass of males is independent of body fat percentage, and variation in body fat percentage**
**appears to be compensated by changes in FFM so that body mass of male elephants is kept relatively**
**stable. At this point it is unclear if this is a general characteristic of male African elephants.” (lines**
**287-295)**

The one measure that could fluctuate a lot (but not in the magnitude of the fat mass variation you
have) is gut contents. Elephants may differ in the amount of food they actually eat (and this is
something that your diet change most likely caused: an increase in total dry matter intake), and
elephants are among those herbivores where increased food intake does not lead to a dramatic
reduction in digesta retention time, i.e. increased food intake mostly leads to increased gut fill
(increased actual gut volume) (Clauss et al. 2007). This is where I am not familiar enough with the
literature on the use of isotope dilution method in large herbivores, how changes in gut fill, i.e. also in
the amount of water “bound” in the gastrointestinal tract, affects FFM estimates. My guess is that gut
contents, which contain water, are just part of the overall FFM that is calculated (and the factor used
for calculation, here 0.746, is derived from measures that include gut content?). I just checked the
paper you used as a source (Wang et al. 1999), and from reading that it becomes evident that the
factor is derived in a way that makes it susceptible to changes in actual gut fill. Even if animals (for the
derivation of the factor) had been investigated using the whole body (including gut contents, and not
just the eviscerated carcass), the question is how they were fed before the analysis, i.e. was ad libitum
intake variation on a variety of diets allowed, or not? I am not really into this literature, but the
example of (Torbit et al. 1985) is nice, where this problem is specifically stated and the study design is
explained in that first different amounts of food were given to the study animals (to create differences
in body fat) but that then, when the actual isotope study was done to assess body composition, intake
was made constant for all animals in order to make the isotope results comparable between the
individuals. This is a factor of uncertainty in studies with live animals that needs to be assessed, and
with a putative variation in gut fill in elephants of a magnitude of 7-17 % of total body mass (Clauss et
al. 2005), I personally believe that variation in gut fill, if not controlled for, will have a large effect on
estimates of FFM and FM in elephants. By incidence, the range is similar to the range in body fat you
mention in l. 256 (5-17%), but I am not sure whether, after accounting for gut content in the TBW
conversion, % translates into % in this comparison.

Yet, basically, this seems to tell me that if you wanted to really assess the effect of different dietary
regimes in the same animal over time, the only way to get reliable (comparable) longitudinal data is
to ensure that food intake is constant AT THE MOMENT OF APPLYING ISOTOPE BODY COMPOSITION.
Otherwise, differences in food intake within the same animal between the time points, e.g. due to
different temperatures (less intake in hot summer? higher intake in cold winter?), different feeding
regimes (higher intake on less-energy-dense diet), will affect the signal you are interpreting due to the
variation in intake (and hence gut fill) at the point of measurement.

**Response: We do not have data on individual intake and thus variation in gut fill. However, we have**
**added additional analyses and discussion to address this important point. See response to point 1,**
**above.**

Your method do not indicate any consistency as to the amount of food available a day prior to, and
during, isotope application. It does not even indicate consistency with respect to the moment of
dosing in relation to feeding events or feeding behaviour. I would suggest that you dig into that and
assess whether it might relate to your data, and at least mention this option.

**Response: We acknowledge that variation in gut fill is a factor that can influence measurements of**
**body composition. We do not have detailed information on ingested food or amount of browse or**
**other food items ingested on a daily basis. However, dosing of individuals always occurred in the**
**morning (with the exception of a single event when dosing occurred at 13:40). The elephants were**
**on habitat overnight and would come to the barn in the morning and have about two flakes of hay**

**(~3 kg) in the morning prior to the training session when dosing occurred. We calculated that we**
**may have overestimated TBW by 1.4-3.4% assuming that gut fill varies between 7-17% body mass**
**and that water content of hay is usually around 20% (if hay would have 25% water we would have**
**overestimated TWB by 1.75-4.25%). Overestimating TBW will result in an overestimation of LBM**
**and an underestimation of body fat percentage.**

**Assuming that elephants dosed later in the morning had more time to feed and a higher gut**
**fill, we would expect that the time of dosing will influence our measurements of LBM, TBW, % body**
**fat and water turnover. But we did not find any evidence for this in the data. Neither time of day**
**nor the interaction between time of day and ID had a significant influence on any of the**
**measurements.**

It would also help if you had actual intake data of your animals – your paper on the composition of
the new diet (Wood et al. 2019) at least suggests that some of this information should be available for
the new diet in the method section: “Daily diets were recorded by NC Zoo keeper staff for the study
duration. This included number of timothy hay (*Phleum pratense*) bales, enrichment food items, and
browse species”, and from these data the averages given in Table 1, and the percentages of the
ingested diet in Table 3, of that paper were calculated (but the actually ingested absolute amounts
are not given in that paper per season). It would be ideal, answering the methodological issues, to link
that information with the isotope data!

**Response: We do not have detailed information on amount of browse or other food items (produce)**
**given and ingested on a daily basis as this was not the aim of the study by Wood and colleagues.**

Another factor of uncertainty might or might not be the variation in the blood sampling time points
after isotope dosing (the methods seem to imply this was done a bit opportunistically, and not
consistently across all assessments). Can this have an effect, or not? This should be mentioned. I also
wonder whether a measure of fit for the slope intercept method of back-calculating the initial isotope
enrichment (or dilution) should be given routinely to give a measure about the “confidence” of that
estimate (on which all other results hinge). Personally, I would recommend to do that.

**Response: The timing of sampling does not affect the rates of depletion or the intercept calculation**
**unless there are substantial changes in physical activity. If there were, we would see this in the fit**
**between the time vs ln-isotope enrichment regression. Instead, the median r^2 value for the**
**regression between time and ln-enrichment was 0.997; the minimum r^2 was 0.990. The error**
**resulting from variation in the timing of samples and thus the slope of the depletion rate is therefore**
**negligible.**

**We did, however, endeavor to collect samples as close to 24h intervals as possible. We included this**
**information and details regarding variation in deviation from multiples of 24h in the following way:**
**“Blood samples (5ml) were drawn via venipuncture once prior to deuterium administration and**
**then 3 or 4 times over the subsequent 10 days, at as close as feasible to multiples of 24 h. The range**
**of absolute deviation from multiples of 24 h was 1–327 min (25th percentile=22.5 min; median=40**
**min, 75th percentile=80.75 min).” (lines 122-125)**

**We included two additional references showing that the slope intercept method is the preferred**
**method to calculate dilution spaces for deuterium and oxygen-18 (Pontzer 2018a, Berman et al**
**2020). We also provide information regarding the fit for the slope intercept method: “Time of**
**sampling and ln-isotope enrichment regression were strongly correlated (median $R^2 = 0.997$; range**
**0.990 – 1.000).” (lines 134-135)**

Patterns in the data that are not mentioned, and may or may not be helpful in identifying physiological
explanations:

Within individuals, there seems to be a negative correlation – is this something one expects? Does it
equate to something like: better hydration status (more body water) = higher throughput because
more is drunk? But that would sound like an instantaneous reaction, not something evident on an
integrated measure of what happens on several days? Note that in my stats program (SPSS), the
within-individual correlations were mostly not significant, though, in spite of the visual pattern. The
same pattern is seen in your Fig. 2A

**Response: The relationship between TBW and water throughput within species is not very strong.**
**With only a few data points per subject, and mainly non-significant within-individual correlations,**
**it is difficult to interpret this pattern.**

**However, this suggests that the variation in water throughput (which we discuss in terms of air**
**temperature) is not just an artifact of the water in the digestive tract. If the amount of water in the**
**body was artificially high (due to water in the digestive tract), it would in turn make the dilution**
**space N artificially high. If this was the case, the result would be an inflated $N \cdot k$ (k = isotope**
**depletion rate) value, which is how water throughput is calculated. Therefore, if there would be no**
**variation in k , and all the apparent variation in water turnover would be due to noise in N , then N**
**and water turnover would be positively correlated. And, since N is used to calculate TBW, there**
**would be a positive relationship between TBW and water turnover within individuals, which is not**
**the case in our data. Thus, water turnover is not simply a function of noise or error in N due to**
**variation in water in the digestive tract.**

Why does higher water throughput seem to be correlated with less body fat? This effect tended
towards significance in a GLM (random factor individual significant, water throughput $p=0.073$). The
outlier would require explanation.

**Response: When running a LMM (% body fat is normally distributed: Shapiro-Wilk normality test:**
**$W = 0.95409$, p -value = 0.3315) that included ID as random factor, water throughput was not**
**significantly related to body fat percentage in our data ($P= 0.34$).**

In particular, I would, as a reader, like to know how a decrease in gut fill would affect the results and
an increase in water intake, and how the combination of these factors (if we claim that in summer,
animals eat less yet drink more, both in absolute terms) would influence any estimates of FFM and
FM.

**Response: Water was available ad libitum and we cannot assess the variation in how much water**
**was drunk and thus, we do not want to speculate on its effect on measurements on body**
**composition. We included a long section in the discussion elaborating how variation in gut fill can**

**affect measurements of body composition: “Another limitation is the fact that we did not control**
**for variation in food intake and the resulting variation in gut fill, which can vary from 7% to 17% of**
**body mass in elephants (Clauss et al. 2005). Dosing with DLW always occurred in the morning (with**
**the exception of a single event when dosing occurred at 13:40). Elephants were on habitat overnight**
**and would come to the barn in the morning where they ate one to two flakes of hay prior (roughly**
**3 kg) to being dosed with DLW. Due to water bound in gut content, we may have overestimated**
**TBW by 1.4 - 3.4%, assuming that gut fill varies between 7 - 17% body mass and that water content**
**of hay is typically around 20%. TBW of the study elephants ranged from 1640 – 4430 L and thus, an**
**overestimation of 1.4% would equal 43 L (range: 23 – 62 L) on average and an overestimation of**
**3.4% would equal 104 L (range: 56 – 151 L), on average. Overestimating TBW by 3.4% will result in**
**an overestimation of LBM by ~3% and an underestimation of percent body fat by ~3%. Thus, the**
**measured changes in body fat percentage within ~3% of one another must be interpreted with**
**caution, as they are within the range that changes in gut fill could affect. However, we did not find**
**that the time of day when dosing with DLW occurred or the interaction between time of day and ID**
**had a significant influence on the measurements of LBM, TBW, body fat percentage and water**
**turnover (ANOVAs: $P > 0.47$), which we would expect if elephants dosed later in the day had more**
**time to feed and thus a higher gut fill. Moreover, if the amount of water in the body was artificially**
**high (due to water in the digestive tract), it would in turn make the dilution space N artificially high,**
**which would result in an inflated $N \cdot k$ (k = isotope depletion rate) value, which is how water**
**throughput is calculated. Therefore, if there would be no variation in k , and all the apparent**
**variation in water turnover would be due to noise in N , then N and water turnover would be**
**positively correlated. And, since N is used to calculate TBW, there would be a positive relationship**
**between TBW and water turnover within individuals, which is not the case in our data. Thus, water**
**turnover is not simply a function of noise or error in N due to variation in water in the digestive**
**tract. Further, we note that measured changes in fat percentage exceeded 4% in most subjects, and**
**that these changes are likely too large to be explained by variation in gut fill.” (lines 322-347)**

3. details

I. 17 whichever way you word these findings, please ensure you do not use the words “during the first
two years” because three spot samples need not represent the full time span.

**Response: We changed the sentence to better reflect the results: “Within the first year after the**
**dietary intervention, there was a reduction in body fat percentage and an increase in FFM. However,**
**final values of both body fat percentage and FFM were similar to initial values.” (lines 14-15)**

I. 19 I always recommend to refrain from superlatives or “we are the first to”. If you want to keep your
superlative, you cannot give the averages because (Chusyd et al. 2018) had an animals with 489 L/d,
higher than the average you mention with the superlative. You need to mention your one individual
value of 553 L/d that was the only one in your dataset that was higher than the maximum in the other
study.

**Response: We deleted the superlative and changed the sentence in the following way: “Water**
**throughput (males: 359 ± 9 L/d; females: 241 ± 28 L/d) was consistent with the allometric scaling of**
**water use in other terrestrial mammals.” (lines 16-17)**

I. 159 please state whether this is gross energy, digestible energy, or what other kind of energy you
are indicating, and whether the unit is kg as fed, or kg dry matter.

**Response: We included this information and changed the sentence in the following way: “The grain-**
**free pellet has higher crude fiber (30% versus 22%) and lower digestible energy content (2,595**
**kcal/kg versus 3,030 kcal/kg as fed).” (lines 178-179)**

I. 171 I think the group level visualisations should only include those datapoints for which all data of
the group are available; otherwise, a trend is implied that is not based on all individuals

Response: We agree that this would be ideal. However, following this suggestion would result in the inclusion of only 2 study periods. However, we changed the figure to highlight differences in sample size per study period so the reader always knows how many datapoint were used to depict the group-level trends.

Fig. 2A the intra-individual trend should be mentioned in the text (and possibly explained).

x-axis label should be kg

Response: We mention the intra-individual trend and corrected the axis title: "In one individual (F1611), water throughput was negatively correlated with body mass ($t = -26.07$, $df = 1$, $P = 0.02$, $R^2 = 0.98$), but this relationship was not found in the other individuals (all $P > 0.05$)." (lines 251-253)

I. 247-248 with the few measurements you have, I think you should refrain from qualitative language like "transient" and "returned" because the stability of the initial state is not ascertained in your data, and continuous measurements might indicate a similar fluctuation just as well as "stability" as implied by the word "return"

Response: We removed qualitative language and change this sentence to: "However, there was no difference between initial and final measurements of body mass and body composition."

I. 263 ff the factor would have to be even higher if more gut contents were assumed, right? But see major discussion above. I. 271 points into the same direction. If the animal had a very high intake at the time, this could explain why the conversion value was too low – I think (I hope you can come up with a comprehensive explanation of the effect of intake).

Response: We have included a paragraph to the discussion to elaborate how variation in gut fill can influence measurements of body composition. Please see responses above.

I think in the acknowledgements it should be "Jeb" or similar, not "Jb"?

Response: It is an unusual name but his middle name which he goes by is actually just the two letter Jb.

Reviewer: 2

Comments to the Author(s)

The authors aimed to (1) investigate how changes in diet altered body mass and body fat in five zoo African elephants, with measurements made once or twice a year over a three-year period, and (2) to investigate the effects of body size and ambient temperature on elephant water budgets. While I understand the difficulties of working with elephants, the study is limited by the small sample size and weak methodology, particularly for the change in diet (no details of exact diet composition or intake by the elephants is given - see my comments below). Despite aiming to give the elephants "an improved diet", their body mass and body fat were the same at the end of the study, changing only transiently. There is no information on exactly how dietary energy intake changed over time. There is no information on other factors that could have influenced the elephants' welfare and body mass, for example, activity level, exposure to heat or cold, disease, social interactions. The effect of temperature on water balance is assessed using data from a weather website, not taking into account the actual microclimate that the elephants would have experienced, for example, by being exposed to solar radiation (or the effect of other factors like activity level). Overall, the study provides few new findings and only tentative data from the small sample. Indeed, the authors recognise that and temper almost every conclusion they draw in the Discussion with a limitation and suggestion for future study. It is well known that elephants have a high water turnover and that increasing environmental heat load will increase that water turnover, and that body size influences water turnover. Unfortunately, I do not think this study provides a significant advance to the literature.

Specific comments:

General: I think that the Introduction is long and its focus could be improved. It jumps from studies of
zoo elephant welfare to the physiology and activity of wild elephants, to ideas about evolved ecology,
then back to zoo management strategies. I would suggest a brief introduction to wild elephant
physiological ecology (including diet, water needs and activity), then moving the focus to the zoo and
how best to manage elephants in that setting, before identifying the gap for the study and the study
approach. It can also be made more concise by removing unnecessary detail, for example, “Morfeld
and colleagues (10) developed a 5-point body condition score (BCS) to visually assess whether
elephants are under- or overweight. Higher scores indicate greater body fat and were correlated with
a biological marker of adiposity (serum triglycerides; 10,23). More than 70% of the 240 elephants in
their study had a BCS of 4 or 5, and were considered overweight or obese”, could be reduced to
“elephants with a higher body condition score had higher serum triglycerides, a marker of adiposity,
and more than 70% of elephants were considered overweight or obese”.

**Response: We re-structured the introduction following these suggestions.**

Comments per line:

8: This comment would apply to all animals.

**Response: We changed “African elephants” to “Animals”**

13. Data on reduced dietary density is not available in the paper, as far as I can tell.

**Response: The nutritional and energetic specifications of the different diets have been published**
**prior to this study (Wood, J. et al. 2019. Analyses of African elephant (*Loxodonta africana*) diet with**
**various browse and pellet inclusion levels. - Zoo Biol.: 1–14), and therefore these data are not**
**presented here again. However, we cite the study by Wood and colleagues and state that the change**
**in diet “increased browse and reduced the pellet component and energy content” (line 80)**

44-46: Elephants do not only achieve evaporative water loss by wetting their skin. There also is
substantial transfer of water across a permeable integument, as described by Dunkin et al.

**Response: We changed the sentence in the following way: “In addition to the daily demands of**
**water for normal physiological function, elephants use water for evaporative cooling by wetting**
**their skin and via transfer across their skin (Dunkin et al. 2013, Mole et al. 2016).” (lines 33-34)**

56: Throughout the paper (including the title), please specify what temperature you are referring to.
The reader could think it is body temperature. Environmental temperature also can be measured in
various ways (e.g. dry-bulb, wet-bulb, soil, black globe temperatures). Note also that dry-bulb air
temperature alone does not describe the heat load experienced by animals outdoors.

**Response: We clarified throughout the manuscript that we are referring to air temperature.**

79: I don't think the animals in this study were not monitored daily?

**Response: This is correct. We exchanged “daily” with “regularly”.**

82: What dietary changes? A diet with lower energy content, different nutritional content?

**Response: We clarified that energy content was reduced and which dietary changes occurred:**
**“dietary changes that decreased energy content” (lines 74-75) and “We predicted that changes in**
**diet that increased browse and reduced the pellet component and energy content of the diet” (lines**
**79-81)**

87-89: Is the prediction based on different energy content for the two food types or is the energy
content matched?

**Response: We clarified that energy content was reduced and which dietary changes occurred:**
**“dietary changes that decreased energy content” (lines 74-75) and “We predicted that changes in**
**diet that increased browse and reduced the pellet component and energy content of the diet” (lines**
**79-81)**

98: How did you account for one elephant being younger and still growing?

**Response: To assess if including an individual in its growth phase influenced our results, we**
**conducted all analyses again excluding this female. There were almost no differences and thus, we**
**report results including this individual and mention results that changed after excluding this**
**individual. (lines 217-220)**

99: Please explain here or elsewhere in the Methods how the elephants were weighed, and the
accuracy of the measurement. Please give the initial mass of all elephants here in the Methods,
particularly since relationships with body mass are investigated in this study.

**Response: We included this information in the text: “Elephants were weighed monthly via a digital**
**scale (Avery Weigh Tronix Model 640; \pm 2.3 kg) which is covered by a platform. Elephants were asked**
**to step on the scale and the weight was taken by an experienced keeper that ensures that all four**
**feet are on the platform. The scale is tested and serviced regularly. At the beginning of the study**
**elephants weighed 2556 – 5785kg.” (lines 93-97)**

100-103: Given that the effect of environmental temperature is a key aim of the study, much more
detail is required to describe exactly how the elephants were housed. For example, did the animals
have access to shade, were they exposed to open skies at night, were the animals able to choose their
own microclimate or were they herded in and out of the barn?

**Response: We included more detailed information regarding housing conditions: “These elephants**
**are primarily housed on two 1.4 ha habitats and are generally given access to the habitats**
**throughout the day and the night. On the habitat, they have access to tree shade, cool pools, and**
**mud wallows, and the habitat includes areas where trees block the wind. When air temperatures**
**are below 4 °C and occasionally overnight, elephants are housed in or given access to a 537m² barn**
**with four attached outdoor paddocks (237 m² each). At the barn, they can move between paddocks**
**and stalls, unless air temperatures dropped below -1 °C, in which case they were secured in the barn**
**for warmth.” (lines 97-103)**

103-105: Exactly how were the five study animals housed? All together, or in two groups?

**Response: We included the following additional information: “Social structure of the group varies,**
**and animals are housed by themselves (for males), in pairs (for females) or in groups of 3 (two**
**females, one male), 4 (all females), or 5 (4 females, one male).” (lines 104-106)**

117: Please explain how elephants were restrained for blood sampling and where blood samples were
taken from.

**Response: We included additional information concerning blood draws: “Blood was collected from**
**a vein on the back of the ear using a 21- or 19-gauge needle. Animals were not restraint during this**
**procedure. They were asked to present their side parallel to the bollards and to stick their ear**
**through an ear portal. The ear was cleaned with warm water and then the area of the blood draw**
**was disinfected (using Novasan or alcohol) prior to the blood draw. Animals are able to decide if**
**they want to participate in the blood draw or not.” (lines 125-129)**

119: Were the plasma samples kept at a particular temperature?

**Response: We included this information: “Samples were centrifuged immediately after collection to**
**separate the plasma and plasma samples were stored at -80 °C.” (lines 129-131)**

145: Humidity (in the subtitle) is not measured/reported.

**Response: We corrected the subtitle and deleted “humidity”.**

146: Presumably mean daily temperature refers to mean daily air temperature? What evidence is
there that mean daily air temperature is suitable to quantify the heat load on elephants in the study?

**Response: We included more information regarding the use of air temperature and how it relates**
**to black globe temperatures: “We used air temperature as a crude estimate of ambient temperature**
**that elephants experience in shaded locations. Studies conducted on African elephants in southern**
**Africa found that air temperature measurements track black globe temperatures, which provide an**
**integrated measure of air temperature, radiant temperature, and the cooling effect of wind (Hidden**
**2009, Mole et al. 2018).” (lines 162-166)**

147: How far is this weather station from the study site?

**Response: We included the distance (27km) between the two sites. (line 162)**

148-149: It is not clear what is meant here? Did you use mean, minimum and maximum daily air
temperatures, or did you use the mean daily temperature you obtained and calculate the maximum
of the mean, etc? Is the measurement period the 10 days over which blood samples were taken?

**Response: We provide additional information to clarify what we calculated: “We calculated average**
**maximum, minimum, and mean air temperature for each 10-day measurement period using reading**
**of daily maximum, minimum and mean air temperature. We also calculated mean dewpoint and**
**cumulative precipitation over each 10-day measurement period.” (lines 166-169)**

151: Was the actual feeding by the elephants observed? Even if the provided food was changed, how
do you know how the energy intake and dietary content of each elephant changed? How much natural
vegetation was provided, what was it exactly, and what was the nutritional content of this vegetation?
As a main aim of the study was to investigate how dietary changes affect body mass, I would expect
to see much greater detail on the nutritional content of the food and detail on actual food (and energy)
intake by study elephants.

**Response: Elephants were not observed closely for dietary consumption other than the dry feed.**
**We noted this in the methods section: “However, data on dietary consumption per individual is not**
**available.”**

**We also discuss how variation in intake, and thus gut fill, can affect our measurements; see**
**comments to Reviewer 1 above.**
**Elephants have access to grass and trees on the habitat.**

165: Please provide a brief description of how the behavioural observations were undertaken.

**Response: We did scan sampling sporadically on the elephants during this study period. The full**
**methods are laid out in the Lasky study. We included more information here: “Behavioral**
**observations, performed as sporadic scan sampling between 9:00 – 17:00 during January 2015 –**
**November 2016,” (lines 166-190)**

177: Dewpoint is not a component of ambient temperature.

**Response: We change the text to: “To examine how air temperature (average, maximum, and**
**minimum) and dew point...” (line 200)**

193: The diet change occurred in February 2016, so why is the “mass over the first two and a half years
of diet change” calculated as the “average increase Aug 15 – Feb 17”? That average includes both
diets.

**Response: We corrected this sentence and the following to reflect changes in mass and body**
**composition after the diet change: “Elephants in this study gained body mass over the first year of**
**diet change (average increase Mar 16 – Feb 17: 324 ± 165 kg). This increase in body mass was largely**
**the result of increased FFM (average increase Mar 16 – Feb 17: 593 ± 385 kg; Fig. 1). Body fat**
**percentage generally decreased during this period (-6.0 ± 5.4 %), but considerable variation was**
**evident among subjects (Fig. 1).” (lines 224-228)**

250-251: What data to you have to support the claim of an improved diet? Improved in what way?
**Response: We included additional information explaining how diet was improved: “Nonetheless,**
**benefits of an improved diet, in this case a reduced amount of grain and caloric intake (Wood et al.**
**2019), can include a wide range of physiological measures, including improved cardiovascular**
**health, even in the absence of permanent changes in body composition (Pontzer 2018b), ” (lines**
**282-285)**

250-252: This is speculation, especially for these study animals. Also, if animals were restrained for
blood sampling, why were other measures of physiological health not obtained?
**Response: Elephants were not restrained during blood sample collection. We clarified this: “Animals**
**were not restrained during this procedure. They were asked to present their side parallel to the**
**bollards and to stick their ear through an ear portal. The ear was cleaned with warm water and then**
**the area of the blood draw was disinfected prior to the blood draw. Animals are able decide if they**
**want to participate in the blood draw or not.” (lines 125-129)**

**We did not collect additional blood volume to measure physiological measures and thus cannot**
**assess changes in these measures here. We changed the sentence to reflect that changes in diet can**
**improve physiological measures, even in the absence of permanent changes in body composition:**
**“Nonetheless, benefits of an improved diet, in this case a reduced amount of grain and caloric intake**
**(Wood et al. 2019), can include a wide range of physiological measures, including improved**
**cardiovascular health, even in the absence of permanent changes in body composition (Pontzer**
**2018b).” (lines 282-285)**

259: What information is available on reproductive cycling in the study animals at the times of
measurements?
**Response: We included information regarding the reproductive status of females: “None of the**
**females was pregnant or lactating during the study.” (line 93)**

286: Dunkin et al. have shown how skin permeability changes seasonally resulting in much greater
evaporative water loss across the skin in summer.
**Response: We included this information in the discussion: “Water requirements increased with**
**higher environmental air temperatures, most likely due to increased insensible water loss (Dunkin**
**et al. 2013, Mole et al. 2016). Permeability of elephant skin changes across seasons, and epidermal**
**permeability is greater in summer compared to winter, allowing a higher rate of evaporative cooling**
**when air temperatures are high (Dunkin et al. 2013).” (lines 349-352)**

288-289: Daytime air temperatures can frequently exceed 40 deg C in the African elephant’s natural
habitat. Importantly, the animals also experience a high heat load from radiation.
**Response: We changed the sentence in the following way: “In the wild, where daily activity levels**
**are greater, daytime temperatures exceed 40°C and animals experience a high heat load from solar**
**radiation, water requirements are likely even greater and habitat and climate specific (Dunkin et al.**
**2013).” (lines 354-356)**

291: Why is 10% body mass considered critical? Many large mammals can cope with larger percentage
losses. Please provide a reference for 10% loss being critical for elephants.

**Response: “We provide a reference (Feldhamer et al 2007) for the statement that 10% body water**
**loss is critical. In the reference (p/ 180) it is stated that “most species die when they lose 10-20% of**
**their body water”. (line 358)**

299: Many elephants do not have migration routes available to them and in reserves would not be in
direct conflict with humans for water.

**Response: We included an additional sentence to incorporate this argument: “However, a large**
**proportion of African elephants live in national parks and reserves that provide access to water at**
**water holes, making conflict with humans over water less likely for these populations.” (lines 368-**
**370)**

303-304: I don't think this conclusion is valid given that the animals returned to initial values.

**Response: We changed the sentence in the following way: “The results of this study suggest that**
**diet interventions can influence body composition, and that maintaining such changes remains a**
**challenge.” (lines 372-373)**

Figure 1: It would be helpful in the elephant ID key to know which elephants were male and female.

**Response: We included individual sex into the ID key.**

Air temperature and diet influence body composition and water throughput in zoo-living African elephants (*Loxodonta africana*)

Herman Pontzer^{1,2}, Rebecca Rimbach^{1,3*}, Jenny Paltan⁴, Erin L. Ivory⁵, Corinne J. Kendall⁵

¹ Evolutionary Anthropology; Duke University, Durham NC USA

² Duke Global Health Institute; Duke University, Durham NC USA

³ School of Animal, Plant & Environmental Sciences, University of the Witwatersrand, Johannesburg, South Africa

⁴ Department of Anthropology; Hunter College, New York NY USA

⁵ North Carolina Zoo; Asheboro NC, USA

Keywords: body fat, dietary intervention, stable isotopes, water turnover

1. Summary

Animals face particular physiological challenges in captivity and the wild. Captive elephants can become over- or under-conditioned with inadequate exercise and diet management. Few studies have quantified body composition or water throughput in elephants, and none to date have examined longitudinal responses to changes in diet or **air** temperature. Using the stable isotope deuterium oxide ($^2\text{H}_2\text{O}$), we investigated changes in body mass, fat free mass (FFM), and body fat in response to a multi-year intervention that reduced dietary energy density for adult African elephants housed at the North Carolina Zoo. We also examined the relationship between **air** temperature and water throughput. Deuterium dilution and depletion rates were assayed via blood samples and used to calculate body composition and water throughput in 2 male and 3 female African elephants at 6 intervals over a 3-year period. **Within the first year after the dietary intervention, there was a reduction in body fat percentage and an increase in FFM. However, final values of both body fat percentage and FFM were similar to initial values in the first 2 years, but these changes were transient.** Water throughput (males: 359 ± 9 L/d; females: 241 ± 28 L/d) **was the highest ever measured in any terrestrial mammal but** was consistent with the allometric scaling of water use in other terrestrial mammals. Water throughput increased with outdoor **air** temperature. Our study highlights the physiological water dependence of elephants and shows that individuals have to drink every 2 to 3 days to avoid critical water loss of $\sim 10\%$ body mass in hot conditions.

*Author for correspondence (rimbach@gmail.com).

†Present address: Evolutionary Anthropology; Duke University, Durham NC USA

2. Introduction

[revised manuscript text omitted]

To date, most studies of elephant physiology and health have been cross-sectional in design
(but see Schiffmann et al. 2019). An underutilized, but potentially powerful, complementary

approach to improving our understanding of elephant physiology and health is to measure
longitudinal responses to changes in diet, activity, or other environmental variables (such as
precipitation and ambient air temperature). Longitudinal studies require greater time investment,
but allow comparisons within subjects and can help isolate the effects of specific variables (i.e.
environmental or intervention strategies). In zoo settings, manipulations can be controlled and
elephants monitored regularly to assess effects. Prospective studies could be particularly useful for
resolving the effects of diet and activity on body mass and adiposity.

The aims of this study were to (i) examine the effects of dietary changes that decreased
energy content on body mass and body fat in zoo-living, adult African elephants, over a three-year
period and (ii) to assess the effects of body size and air temperature on elephant water budgets.
We measured adiposity and water throughput using deuterium dilution and depletion rates,
respectively. This minimally invasive approach provides precise measures of total body water
(TBW), fat free mass (FFM), fat mass (FM) and daily water throughput (L/d). We predicted that
changes in diet that increased browse and reduced the pellet component and energy content of the
diet would reduce body fat percentage among elephants. Further, we predicted that water
throughput among elephants would follow the body size scaling evident in other mammals, and
that warmer air temperatures would increase daily water throughput. In order to assess some of the
physiological challenges associated with their large body size, we compared daily water
throughput among elephants to that of other mammals and measured the effects of seasonal
changes in air temperature on daily water throughput.

**3. Material and Methods**

*Sample & Measurement Periods*

North Carolina Zoo (Asheboro, NC, USA) houses six African elephants (4F, 2M), all are adult
animals (aged 32 to 43) except for one female who was 13 years of age at the beginning of this
study. One adult female was not included in the study as it was not possible to weigh her during
the study period. None of the females was pregnant or lactating during the study. Elephants were
weighed monthly via a digital scale (Avery Weigh Tronix Model 640; ± 2.3 kg) which is covered
by a platform. Elephants are asked to step on the scale and the weight is taken by an experienced
keeper that ensures that all four feet are on the platform. The scale is tested and serviced regularly.
At the beginning of the study elephants weighed 2556 – 5785kg. These elephants are primarily
housed on two 1.4 ha~~3.5 aere~~ habitats and are generally given access to the habitats throughout the
103 day and the night. On the habitat, they have access to tree shade, cool pools, and mud wallows,
and the habitat includes areas where trees block the wind. When air temperatures are below 4 °C
and occasionally overnight, elephants are housed in or given access to a 537,783-m²sq-ft barn with
four attached outdoor paddocks (237 m²2,550-sq-ft each). At the barn, they can move between
paddocks and stalls, unless air~~r~~ temperatures dropped below -1 °C, in which case they were secured
in the barn for warmth. Elephants weare housed in mixed social groups including male and female
animals. Social structure of the group varies, and animals are housed by themselves (for males),
in pairs (for females) or in groups of 3 (two females, one male), 4 (all females), or 5 (4 females,
one male).

Body mass, total body water (TBW), fat free mass (FFM), and daily water throughput (
[revised manuscript text omitted]

\pm SD.

To examine how airmbient temperature (average, maximum, and minimum) and dew
point) and body mass influence water throughput, we used linear mixed models (LMMs) with
Gaussian error structure using the package 'lme4' [42]. We log-transformed water throughput to
reduce heteroscedasticity when fitting the models, and we used individual ID as a random factor

because individuals were sampled repeatedly. We did not include sex as a predictor in the models
because there was collinearity between sex and mass. We included the interaction term between
air temperature and body mass, and removed this interaction due to non-significance. We
standardized (z-transformed) all numeric predictors for more accurate model fitting and to
facilitate comparisons of model estimates [43]. We verified the model by inspecting Q–Q plots
and by plotting model residuals against fitted values. We checked for multi-collinearity by
calculating variance inflation factors [44] for the predictor variables using the ‘vif’ function in the
car package [45], which did not indicate collinearity (all vifs < 2).

We used ANOVAs that included time of day, individual ID and the interaction between
time of day and ID to assess whether measurements of LBM, TBW, % body fat and water turnover
were influenced by the time of the day when dosing with DLW occurred. We used Pearson's
product-moment correlations to examine overall and within-individual relationships between body
mass, FFM and % body fat. One female was 13 years old at the onset of the study, and thus, was
still growing throughout the study [46]. To assess if including an individual in its growth phase
influenced our results, we conducted all analyses again excluding this female. There were almost
no differences and thus, we report results including this individual and mention results that
changed after excluding this individual.

**3. Results**

Individuals varied in body mass, body composition, and water throughput, with males
weighing considerably more and using more water (Table 1, Fig. 1). Elephants in this study gained
body mass over the first two and a half years of diet change (average increase Mar-Aug 156 – Feb
17: 32407 ± 165185 kg). This increase in body mass was largely the result of increased FFM

(average increase ~~Aug-15~~Mar 16 – Feb 17: 593 ± 385 kg; Fig. 1). Body fat percentage generally
decreased during this period (-6.0 ± 5.4 %), but considerable variation was evident among subjects
(Fig. 1). ~~In the final measurements of;~~ body mass, FFM and body fat percentage were similar
~~returned to values similar~~ to the initial values (paired t tests: body mass: $t = -0.058$, $df = 3$, $P =$
0.957 ; FFM: $t = -0.086$, $df = 3$, $P = 0.936$; % body fat: $t = 0.182$, $df = 3$, $P = 0.867$). Body fat
percentage was significantly lower at the end of the study compared to initial values when
excluding the young female ($t = 6.608$, $df = 2$, $P = 0.022$). Time of dosing with DLW and the
interaction between time of day and ID did not influence measurements of LBM, TBW, % body
fat and water turnover (ANOVAs: all $P > 0.47$).

There was a strong positive relationship between FFM and body mass (Pearson's product-
moment correlation: $t = 26.10$, $df = 22$, $P < 0.0001$, $R^2 = 0.96$; Fig. 2A), and a weaker and negative
relationship between % body fat and both FFM ($t = -3.17$, $df = 22$, $P = 0.004$, $R^2 = 0.31$) and body
mass ($t = -2.09$, $df = 22$, $P = 0.048$, $R^2 = 0.16$). There was no relationship between body mass and
FFM within individuals M27 and F1611 (all $P > 0.39$; Fig. 2A), although there was a positive trend
for M1770, F1771 and F1772 ($P = 0.042 - 0.089$; Fig. 2A). There was no relationship between
body mass and % body fat within individuals (all $P > 0.21$). There was a negative relationship
between FFM and % body fat in males (all $P < 0.01$; Fig. 2B), but not in females (all $P > 0.15$; Fig.
2B).

 **Figure 1.** Changes in key characteristics over the course of the study. Absolute values for each
 subject (A - D) and pooled values (E - H) with each subject's values normalized to their mean for
 the study. Month/year of measurement is indicated. Diet was modified in February 2016 (vertical
 dashed line). Sample sizes per sampling period are presented for pooled values (E - H). Data in
 Table S1.

Figure 2. Relationship between (A) body mass and FFM and (B) FFM and body fat percentage (lines indicate linear regression lines).

Table 1. Key characteristics for the five elephants in this study. Means (\pm SD) are shown. Data per sampling event presented in Table S1.

Subject	Sex	Age	Body Mass (kg)	Total Body Water (L)	Fat Free Mass (kg)	Body Fat %	Water Throughput (L/d)
1770	M	32-34	5815 \pm 190	4020 \pm 349	5389 \pm 467	7 \pm 6	353 \pm 43
27	M	41-43	5553 \pm 107	3698 \pm 177	4957 \pm 238	11 \pm 4	365 \pm 97
1772	F	13-15	2654 \pm 96	1731 \pm 79	2320 \pm 106	13 \pm 2	227 \pm 40
1611	F	33-35	4401 \pm 140	2817 \pm 131	3776 \pm 175	14 \pm 2	223 \pm 42
1771	F	37-39	3905 \pm 150	2587 \pm 142	3467 \pm 190	11 \pm 2	273 \pm 47

Daily water throughput was related to both body mass (Estimate \pm SD = 0.186 \pm 0.055, $t = 3.345$, $P = 0.0008$; Fig. 32A) and mean daily air temperature (Estimate \pm SD = 0.141 \pm 0.028, $t = 5.045$, $P < 0.0001$). Mean water throughput increased with mean body mass at a rate of approximately 4.5 L/d for every 100 kg increase in body mass. In one individual (F1611), water throughput was negatively correlated with body mass ($t = -26.07$, $df = 1$, $P = 0.02$, $R^2 = 0.98$), but this relationship was not found in the other individuals (all $P > 0.05$).

Water throughput fluctuated seasonally, with lower throughput in the cooler months (Fig.
 1). When mean daily air temperature ranged from 6 – 14 °C, water throughput was 325 ± 30 L/d
 for males and 218 ± 38 L/d for females. During warmer measurement periods (mean daily air
 temperature 23 – 24 °C) water throughput was 427 ± 89 L/d for males and 278 ± 18 L/d for females.
 Measured as a percentage of TBW, water throughput increased at a rate of approximately 1% TBW
 for every 4 °C increase in mean daily air temperature (Fig. 23B). Daily water throughput was also
 related to maximum (Estimate \pm SD = 0.527 ± 0.229 , $t = 0.229$, $P = 0.032$) and minimum air
 temperature (Estimate \pm SD = 0.715 ± 0.201 , $t = 3.548$, $P = 0.002$), and dew point (Estimate \pm SD
 = 0.641 ± 0.212 , $t = 3.014$, $P = 0.007$). Males had a significantly larger body mass than females
 (Estimate \pm SD = 2021.869 ± 675.103 , $t = 2.994$, $P = 0.002$), and body mass was weakly influenced
 by mean daily air temperature (Estimate \pm SD = -8.474 ± 4.281 , $t = -1.979$, $P = 0.047$). There was
 no relationship between body mass mean daily air temperature when excluding the young female
 (Estimate \pm SD = -8.790 ± 5.190 , $t = -1.693$, $P = 0.09$).

 **Figure 32.** Water turnover in adult African elephants (females = open circles, males = filled
 circles). A. Water throughput (L/d) increased with body mass. B. Water turnover as a percentage

of total body water (% TBW) increased with mean daily air temperature. Line indicate linear
regression line and shaded area shows 95% confidence interval.

Water throughput among adult African elephants was consistent with that of other
mammals. Mean water throughput for males (body mass: 5684 ± 200 kg), averaged across all
periods (359 L/d), was 17% above the regression for other mammals (Fig. 34). Mean water
throughput for female elephants (body mass: 3508 ± 787 kg) averaged across all periods (241 L/d)
was 18% above the regression for other mammals. These deviations from the regression were
modest relative to other species in the comparative dataset (Fig. 43). Indeed, the regression
equation for mammals excluding elephants ($y = 0.920x - 0.974$, $df = 38$, $adj. r^2 = 0.97$, $P < 0.0001$)
was nearly identical to the regression equation including elephants ($y = 0.924x - 0.971$, $df = 40$,
$adj. r^2 = 0.97$, $P < 0.0001$; Fig. 43).

**Figure 34.** Water throughput in adult female (filled circle) and male (filled square) elephants and
other terrestrial mammals (open circles). Line indicates least squares regression for all data,
including elephants: $\log_{10}\text{Water Throughput} = 0.924 \pm 0.023 \log_{10}\text{Mass} - 0.971 \pm 0.039$ (df = 40,
adj. $r^2 = 0.97$, $P < 0.0001$).

5. Discussion

Male and female African elephants in this study showed increased FFM and reduced body fat
percentage in the first year after response to a dietary interventions. However, there was no
difference between initial and final measurements of body mass and body composition ~~changes~~
~~were transient, with values returning to baseline within approximately two years.~~ The difficulty in
maintaining beneficial changes in body composition among these elephants mirrors the challenges
experienced in other species, including humans [47–50]. Nonetheless, ~~the~~ benefits of an improved
diet, in this case a reduced amount of grain and caloric intake [40], can are likely to include a wide
range of physiological measures, including improved cardiovascular health, even in the absence
of permanent changes in body composition [48]. Future intervention studies should investigate
factors that improve the duration of body composition changes and assay additional health impacts
and parameters.

When analyzing all individuals together, FFM increased when body mass increased.
Within individuals we did not find this relationship, likely due to the restricted number of repeated
measurements. When analyzing all individuals together, body fat percentage decreased when body
mass increased but within individuals this relationship was not evident. In males, but not females,
there was a negative relationship between FFM and body fat percentage, where males with larger
FFM had a lower body fat percentage. These results suggest that body mass of males is

325 independent of body fat percentage, and variation in body fat percentage appears to be
compensated by changes in FFM so that body mass of male elephants is kept relatively stable. At
this point it is unclear if this is a general characteristic of male African elephants.

Body fat percentages for elephants in this study were similar to those reported for a large
adult female sample ($9 \pm 4\%$, range: 5 – 17%, $n = 20$) [16]. Adult males and females appear to
have similar body fat percentages, but we are limited in this assessment to the small number of
individuals in this study. A lack of sexual dimorphism in body fat percentage would suggest that
the energetic burdens of pregnancy and lactation are not sufficiently large as to select for additional
energy buffering adaptations in females. However, a much larger sample size is needed to verify
this finding as a general characteristic of African elephants. Moreover, it will be pertinent to assess
variation of relative body fat with age and life stage.

It should be noted that we used a TBW:FFM ratio of 0.746 in this study, based on a
comparative mammalian dataset [34], whereas Chusyd and colleagues [16] used a value of 0.730.
If we were to use the 0.730 value, body fat percentages for elephants in this study would be reduced
~2% (i.e., 11% body fat would be 9%). These differences are not substantial, but the proper
TBW:FFM ratio for elephants warrants future investigation as this would improve the accuracy of
isotope-based assessments of body composition. For example, we believe that the negative body
fat value (-1%) calculated for elephant 1770 in September 2016 reflects measurement error around
a true body fat percentage that was very low, compounded by the use of a TBW:FFM value that
was too low. Alternatively, this negative value could indicate that some portion of the dose was
not ingested. Improved accuracy in the TBW:FFM value for elephants would enable us to assess
these explanations. However, determining TBW:FFM ratio for elephants requires carcass
lyophilisation, a direct and accurate method for determining body fat content. This method is time

consuming and the killing of animals makes it ethically undesirable and likely explains why
TBW:FFM has not been assessed for elephants so far.

Limitations of this study include the small number of elephants measured and the relatively
narrow age range of subjects. Uncertainty in the ratio of TBW:FFM limited the accuracy of
isotope-based body composition measurements. This uncertainty might result in less accurate
absolute numbers, but since we used the same ratio for all individuals, longitudinal changes should
be accurately reflected.

Another limitation is the fact that we did not control for variation in food intake and the
resulting variation in gut fill, which can vary from 7% to 17% of body mass in elephants [51].
Dosing with DLW always occurred in the morning (with the exception of a single event when
dosing occurred at 13:40). Elephants were on habitat overnight and would come to the barn in the
morning where they ate one to two flakes of hay prior (roughly 3 kg) to being dosed with DLW.
Due to water bound in gut content, we may have overestimated TBW by 1.4 - 3.4%, assuming that
gut fill varies between 7 - 17% body mass and that water content of hay is typically around 20%.
TBW of the study elephants ranged from 1640 – 4430 L and thus, an overestimation of 1.4% would
equal 43 L (range: 23 – 62 L) on average and an overestimation of 3.4% would equal 104 L (range:
56 – 151 L), on average. Overestimating TBW by 3.4% will result in an overestimation of LBM
by ~3% and an underestimation of percent body fat by ~3%. Thus, the measured changes in body
fat percentage within ~3% of one another must be interpreted with caution, as they are within the
range that changes in gut fill could affect. However, we did not find that the time of day when
dosing with DLW occurred or the interaction between time of day and ID had a significant
influence on the measurements of LBM, TBW, body fat percentage and water turnover (ANOVAs:
P > 0.47), which we would expect if elephants dosed later in the day had more time to feed and

371 thus a higher gut fill. Moreover, if the amount of water in the body was artificially high (due to
372 water in the digestive tract), it would in turn make the dilution space N artificially high, which
would result in an inflated $N*k$ (k = isotope depletion rate) value, which is how water throughput
is calculated. Therefore, if there would be no variation in k , and all the apparent variation in water
turnover would be due to noise in N , then N and water turnover would be positively correlated.
And, since N is used to calculate TBW, there would be a positive relationship between TBW and
water turnover within individuals, which is not the case in our data. Thus, water turnover is not
simply a function of noise or error in N due to variation in water in the digestive tract. Further, we
note that measured changes in fat percentage exceeded 4% in most subjects, and that these changes
are likely too large to be explained by variation in gut fill.

Daily water throughput was high among African elephants, commensurate with their large
body masses. Water requirements increased with higher environmental air temperatures, most
likely due to increased insensible water loss [12,13]. Permeability of elephant skin changes across
seasons, and epidermal permeability is greater in summer compared to winter, allowing a higher
rate of evaporative cooling when air temperatures are high [12]. Water throughputs for males in
summer months were the highest ever measured for any animal, reaching 400 to over 500 L/d
(Table S1). In the wild, where daily activity levels are greater, and daytime temperatures exceed
and approach ~40°C and animals experience a high heat load from solar radiation, water requirements
are likely even greater and habitat and climate specific [12]. With TBW accounting for ~67% of
body mass (Table 1; and see [16]), and water loss in hot conditions in excess of 10% TBW,
elephants would need to drink every 2 to 3 days to avoid critical water loss of ~10% body mass
[52]. This result fits well with observations of wild elephants, which rarely drink less often than

every 3-4 days in the dry season [53]. Elephants travel faster in the dry season than in the wet
season, likely trying to minimize the time spent traveling between water and food sources [53].

Physiological water requirement provides additional context to the water-directed
behaviors and ecology of elephants in the wild [9,13] and highlights the dependence of elephants
on access to drinking water. Most parts of southern Africa are predicted to get drier and hotter in
the future, and arid areas are expected to expand further [10]. Such climatic changes should place
increased water stress on elephants inhabiting these areas which could affect migration routes and
result in conflict over water with human population (e.g. water used for agriculture). This would
likely negatively impact already vulnerable elephant populations. However, a large proportion of
African elephants live in national parks and reserves that provide access to water at water holes,
making conflict with humans over water less likely for these populations.

Longitudinal physiological studies provide an important perspective on captive elephant
health and well-being. The results of this study suggest that diet interventions can influence
improve body composition, and that ~~by reducing adiposity and increasing FFM.~~ Mmaintaining
such these positive changes remains a challenge. Long-term monitoring of body composition and
other aspects of health, using stable isotopes and other minimally invasive techniques, will
continue to improve the care of elephants in human settings as well as shed light on their
physiological adaptations and ecology in the wild.

**Acknowledgements**

We thank North Carolina Zoo staff for their assistance with data collection. We thank Marcus Clauss and
one anonymous reviewer for their comments on earlier versions of the manuscript. Jb Minter provided
comments on an earlier version of this manuscript.

**Ethical Statement**

The North Carolina Zoo is an accredited zoo with the Association of Zoos and Aquariums, and institutional
approvals for this work were obtained prior to the study through the zoo's Research Review Committee.

**Author' Contributions**

HP and CJK designed the study, analyzed data, and wrote the manuscript. JP and ELI assisted with data

[revised manuscript text omitted]

580

Appendix E

I would call these "estimated fat free ..." to stress that these are estimates based on the method

Response: We followed this suggestion and changed the text accordingly. (line 9)

in my view, the best solution for the gut-contents-issue would be to write "fat free mass (FFM, including by definition fat-free gut contents)"

Response: We followed this suggestion and changed the text accordingly:

"Using the stable isotope deuterium oxide ($^2\text{H}_2\text{O}$), we investigated changes in body mass, estimated fat free mass (FFM, including fat-free gut content), and body fat in response to a multi-year intervention that reduced dietary energy density for adult African elephants housed at the North Carolina Zoo." (lines 8-11)

While this sentence is true, considering the results, it might well mis-represent the results: In my view, you need to also state that there was an increase in overall body mass. Otherwise, the sentence will be interpreted as a reduction in body mass with a loss of body fat and a (less-than-compensating) increase in FFM. Given the propensity of readers to think a diet intervention is designed to reduce obesity, it is important to state that actually, body mass increased after the intervention.

Response: We followed this suggestion and changed the text accordingly:

"Within the first year after the dietary intervention, there was an increase in overall body mass, a reduction in body fat percentage and an increase in FFM." (lines 14-16)

to re-inforce the comment in the abstract: these are the goals of the study, and therefore the development not only of %fat and absolute FFM, but also of total body mass, should be recorded in the abstract.

Response: We followed this suggestion and changed the abstract accordingly. Please see response to previous comment.

I recommend to change this to "estimates". In herbivores with no consistent food intake and water intake level, very evidently the measures cannot be "precise".

Response: We followed this suggestion and exchanged "precise measures" to "estimates". (line 79)

While the prediction is honest, it would be appropriate to also state your predictions on overall body mass, and on the development of FFM - as you give these results as well.

Response: We included prediction for body mass and FFM:

"We predicted that changes in diet that increased browse and reduced the pellet component and energy content of the diet would reduce body fat percentage and body mass, but not affect FFM, among elephants." (lines 81-83)

Normal elephant husbandry provides animals with access to hay throughout the night so that they are never completely without food. Withdrawing of food overnight would be unusual. I recommend to state the complete feeding regime.

Additionally, gut contents are not determined by a single feeding event, but by the general feeding regime in herbivores.

Response: We included information about food availability for elephants throughout the night.

"Animals were on habitat overnight, with access to hay, browse and grass and would come to the barn in the morning where they may have ingested 1-2 flakes of hay (roughly 3 kg) before dosing, but animals were fed the majority of their diet after dosing on their habitat." (lines 126-128)

I think the best way here would be to say "FFM (including, by definition, gut contents) was ..."

Response: We followed this suggestion and changed the text as suggested.

PDF: By the way, you added this statement l. 142-148 about the Wang paper and the low coefficient of variation you find in the data. But look at that data in the Wang paper – of the 15 species, 4 are clearly indicated as “carcass”, i.e. without gut contents. Among the remaining 11, you have several that are carnivores, so one can assume – e.g. in seals dissected for disease or after a period of non-feeding – that their guts are empty anyhow, even if they are labelled “whole body”. For the cattle, a paper from 1920 is cited by Wang et al., and if you check it out, it becomes clear that this is not even the carcass but either only “flesh”, or “flesh”, liver and blood, again excluding gut contents that were analysed there. For the sheep and goats, a paper is cited whose abstract says that the animals were fasted – no food, no water! – for 48 hours! – so again, the gut contents would not be representative of live herbivores. For the baboon, a paper is cited in which the “total body weight” is actually defined as gut-contents-free, so the label “whole body” is not appropriate ... so resting the case of disregarding gut contents on the Wang et al. paper is not valid.

On the other hand, there are papers that state the relevance of considering gut fill in doublelabelled water measurements: (Torbit et al. 1985).

The prediction that body fat is underestimated if gut contents are not taken into consideration is supported in the literature, e.g. a study with horses (Dugdale et al. 2011). By the way, in that study, without commenting on the feeding regime prior to deuterium application (except that most animals received the same hay), the equilibration time was without access to food and water.

“The variability in the gastrointestinal (GI) contents of ruminants can result in significant errors in prediction of body composition from estimates of total body water (TBW). Byers (7) developed a two-compartment model for steers to differentiate EBW and water associated with the GI contents. For the D2O technique to be applicable for ruminants, procedures are needed to differentiate GI water from EBW” (Andrew et al. 1995).

These literature findings cannot be explained away by citing Wang et al., and should be included in discussion about the interpretation of the findings.

MS: The 15 species of Wang comprise 11 species for which the data is based on total body composition (including gut contents) but also 4 species for which it is based on carcass composition (i.e., without GIT and its contents). These latter 4 should not be included in my view (even though that does not change the data in a relevant way).

But then, the 11 species include data for goats and sheep cited as "Panaretto". The corresponding paper, though giving data for the whole body, used animals "deprived of food and water for 48 h" (I only had access to the abstract, which did not speak about sheep, so I do not know how the paper was used by Wang to reference sheep).

For cattle, the Wang paper claims "whole body" from a paper by Moulton, but when one reads that paper, it is evident that this is not based on a whole-body analysis but either on muscle only, or on muscle, liver, and blood. So again, no whole body.

Checking the data source for baboon in Wang et al., the Lewis paper, shows that in that paper, total body weight was defined as wet body weight minus gut contents, so again, without considering gut contents even though "whole body" is stated in Wang et al.'s table.

For the rat data, the animals were "fasted overnight" and not given food during the measurement.

Citing Wang in the way done here, as if it represented data for animals including their gut contents, is in my view not correct.

Response: We included a sentence to highlight that animals included in that study had either an empty gut, where in fact carcasses or where studies that excluded gut content:

“It should be noted here that Wang et al. [34] reviewed studies that excluded gut contents when determining hydration, examined animals which were deprived of food and water, or studies that used carcasses.” (lines 148-150)

Further, we explain here already that estimates include gut contents:

“Elephants in this study were never deprived of food or water, and thus their body masses include gut contents, which can account for 7% - 17% of body mass in elephants [35]. Gut contents provide an additional reservoir for isotope dilution, and are therefore included in calculations of TBW and hence FFM. We discuss the potential effects of gut content variation and uncertainty in the TBW:FFM ratio for our analyses below. Fat mass was calculated from the difference in body mass and FFM.” (lines 151-156)

Again, just calling it “FFM (including gut contents)” and keeping all else in estimation methods as it is, and then demonstrating in the discussion how, depending on 80-90% moisture in gut contents and gut contents of 7-17% of body mass, the results may change fat estimates (by overestimating FFM), and most particularly, linking this to the expected change in gut contents due to the diet change, would all make perfect sense to me.

Response: We followed this suggestion. We state that FFM includes gut content in several instances throughout the manuscript and the figures. We deleted “At this point it is unclear if this is a general characteristic of male African elephants.”

In the discussion we state now that:

“Variation in gut fill, which can vary from 7% to 17% of body mass in elephants [35], must also be considered in interpreting measures of body composition for elephants in this study. Elephants in our sample likely increased their volume of food intake in response to the change from a high-calorie diet to a low-calorie diet [52]. Increased food intake will result in an increased gut fill [53] consisting of dry matter contents (which are mostly fat-free) and water (e.g., from saliva and other secretions). Previous studies have stated the relevance of considering gut fill when using isotope dilution methods in ruminants [54–56]. Water content of elephant gut contents [57] and feces [58] ranges between 80 – 90% and thus is higher than the 73 - 75% hydration assumed in the calculation of FFM. Converting TBW (which includes the water in the gut) to FFM (including gut contents) using a hydration constant of 74%, as done in this study, will therefore lead to an overestimation of FFM (including gut contents). This overestimation will, in turn, lead to an underestimation of fat mass and body fat percentage. Such an underestimation of body fat has been reported in deer (*Odocoileus hemionus*) [54] and ponies, where body fat percentage determined via isotope dilution was on average 1.78% lower than carcass dissection-derived values [55]. Depending on the variation of gut water (80 - 90% in ruminants [55,57]) and variation in gut fill between 7 - 17% of body mass [35], it is likely that we under-estimated body fat mass by 6.0 – 14.5% (via an over-estimation of FFM). Further, within-subject variation body composition between measurement periods could be due in part to variation in gut fill, and true FFM (excluding gut contents) could be even more stable than the weight measurements and isotope dilution analyses suggest.” (lines 324-343)

so the overall food intake should have increased, which would be a good reason to have a higher FFM (which includes gut contents).

Response: We discuss that overall food intake should have increased in the discussion. Please see response to the previous comment.

please describe the distribution of the food across the day, incl. potential periods when the animals did not have food available

Response: We included the following sentences to describe to describe the distribution of food across the day.

“Elephants had access to hay, browse and grass during the night. Most food items were consumed overnight. In the morning, elephants received 1 - 2 flakes of hay as "breakfast". The pellets were utilized in enrichment or for training and each elephant was given their appropriate quantity of grain individually.” (lines 196-200)

please define the dates you use for that. if I go by fig. 1, the beginning is the first data point, and the end the last - then it is n=

Response: We included the additional information:

“We used paired t tests to examine whether body mass, FFM and body fat percentage of the study animals had changed throughout the study. We used an elephant’s first (January 2015: n=1; March 2015: n=1, or August 2015: n=2) and last (February 2017: n=1 or October 2017: n=3) measurement for this analysis.” (lines 210-214)

again, I would add a footnote saying: including GIT contents

Response: We included a footnote in Table 1 and also in Table S1.

I would place all body-mass related descriptives up above to the first section on body mass.

Response: We followed this suggestion and moved this information accordingly. (lines 242-246)

I maintain that this is highly unusual in short-term studies: quick variation in body mass should be associated with fat gain and loss, not FFM when understanding FFM as fat-free tissue. When understanding FFM as fat-free tissue plus fat-free gut contents, then all makes sense. It would also correspond to physiology as we expect it (animals eating more of the lower-calory food), which has been specifically mentioned as a strategy in elephants

Meyer K, Hummel J, Clauss M (2010) The relationship between forage cell wall content and voluntary food intake in mammalian herbivores. *Mammal Review* 40:221-245

need to put it in other words: yes, body mass changes mainly due to FFM can be expected in elephants, if FFM includes gut contents (as it does, see below).

When would we expect an increase in gut contents in elephants?

Especially when they are changed from a high-calory-diet to a low-calory-diet. This is also reflected in the increase in roughage and the increase in browse after the diet change.

So, what would I predict after such a diet change?

- in growing animals: a reduction in body fat % (of total body mass, and of fat-free body mass incl. gut contents, and of fat-free tissue mass excluding gut contents) as well as a continuous increase in fat-free tissue mass (muscles, bones etc. that grow)

- in animals not growing at all: a reduction in body fat % (of total body mass, and of fat-free body mass incl. gut contents, and of fat-free tissue mass excluding gut contents) but no change in fat-free tissue mass

- in ALL animals: an increase in gut fill, because they will eat more of the lower-quality diet after an adaptation period (where they may rely more on adipose reserves); this means both an increase in dry matter contents of the gut (and this dry matter content is more or less fatfree), and an increase in water bound in the gut (and this is fat-free). Total FFM (including gut contents) should therefore increase in all animals until a new equilibrium with the new diet is reached.

How do you estimate the amount of water bound in gut contents? Not by the water concentration of the food (the logic laid out in l. 331) – water concentration in gut contents is usually much higher than what is found in the food. Animals eating hay will not have gut contents with the water concentration of hay (or there would be no saliva, no secretions) ... The water concentration of gut contents is typically between 80-90 % depending on the section of the gut one is looking at – for free-ranging elephants: (Clemens and Maloiy 1982). But imagine that during the process of digestion, at the end of the gut, water is re-absorbed in the process of faecal bolus formation – and the faeces of hay-fed zoo elephants still contain 80% water (Clauss et al. 2003), i.e. it is reasonable to assume that gut contents are higher, as faeces are typically drier, not moister, than overall gut contents!

I think this information must inform the calculations about potential estimates of total body water in section l. 321-350.

If one checks out the data on water concentration of gut contents (e.g., (Clemens and Maloiy 1982)) and faeces (e.g., (Clauss et al. 2003)), one can see that this is typically not in the area of 73 or 75 %, but more. I might be wrong here, but doesn't this mean that, assuming FFM is composed of many things incl. gut content, that it is over-estimated if there is a relevant portion of total body mass (in the area of 7-17%) whose water contents are not 0.74, but something like 0.85?

And if FFM is over-estimated in that way, would that make fat estimates too low – incl. the one negative value?

So following the logic in l. 330, the over-estimation of FFM, and the under-estimation of body fat, is not 1.4-3.4%, but 6.0-14.5%. This is basically the range of body fat measured in the study, meaning the fat values might in theory, assuming our maximum gut fill range, be double of what is estimated, right?

My point is, in my view the whole narrative of body composition changes is flawed if one disregards gut contents. And given the change in diet regime, we would automatically expect an increased intake which – especially in elephants – does not translate into a quicker passage (due to a fixed gut fill volume) but on the contrary passage is kept stable (which means gut fill volume must increase) (Clauss et al. 2007). (don't cite all this my stuff, this is just for explanation).

So, writing as if FFM represented body tissue and not the combination of body tissue and gut contents remains, to me, the major flaw in the manuscript, and I apologize for not pointing this out more specifically in the first review. I actually think that once one would define FFM in this particular manuscript as “including gut contents”, then the whole narrative would make sense.

Response: Thank you for your detailed explanations and thorough comments.

We changed this part of the discussion in the following way:

“Variation in gut fill, which can vary from 7% to 17% of body mass in elephants [35], must also be considered in interpreting measures of body composition for elephants in this study. Elephants in our sample likely increased their volume of food intake in response to the change from a high-calorie diet to a low-calorie diet [52]. Increased food intake will result in an increased gut fill [53] consisting of dry matter contents (which are mostly fat-free) and water (e.g., from saliva and other secretions). Previous studies have stated the relevance of considering gut fill when using isotope dilution methods in ruminants [54–56]. Water content of elephant gut contents [57] and feces [58] ranges between 80 – 90% and thus is higher than the 73 - 75% hydration assumed in the calculation of FFM. Converting TBW (which includes the water in the gut) to FFM (including gut contents) using a hydration constant of 74%, as done in this study, will therefore lead to an overestimation of FFM (including gut contents). This

overestimation will, in turn, lead to an underestimation of fat mass and body fat percentage. Such an underestimation of body fat has been reported in deer (*Odocoileus hemionus*) [54] and ponies, where body fat percentage determined via isotope dilution was on average 1.78% lower than carcass dissection-derived values [55]. Depending on the variation of gut water (80 - 90% in ruminants [55,57]) and variation in gut fill between 7 - 17% of body mass [35], it is likely that we under-estimated body fat mass by 6.0 – 14.5% (via an over-estimation of FFM). Further, within-subject variation body composition between measurement periods could be due in part to variation in gut fill, and true FFM (excluding gut contents) could be even more stable than the weight measurements and isotope dilution analyses suggest.” (lines 324-343)

with or without food?

Response: We clarified that elephants had access to hay, grass and browse during the night. Animals were on habitat overnight, with access to hay, browse and grass and would come to the barn in the morning where they may have ingested 1-2 flakes of hay (roughly 3 kg) before dosing, but animals were fed the majority of their diet after dosing on their habitat. (lines 126-128)

No, this is not an overestimation because gut contents are part of your body mass measurement you use. TBW stays the same if one considers gut contents. It is lean body mass or FFM excluding gut contents that is overestimated.

This is the reason why I think it would be really good to use definitions like "total body water (TBW, incl. gut contents)" and "fat free mass (FFM, incl. gut contents)" - then all is correct in terms of terminology.

Response: We state now that FFM includes gut content in several instances throughout the manuscript and the figures. We changed this part of the discussion in the following way:

“Variation in gut fill, which can vary from 7% to 17% of body mass in elephants [35], must also be considered in interpreting measures of body composition for elephants in this study. Elephants in our sample likely increased their volume of food intake in response to the change from a high-calorie diet to a low-calorie diet [52]. Increased food intake will result in an increased gut fill [53] consisting of dry matter contents (which are mostly fat-free) and water (e.g., from saliva and other secretions). Previous studies have stated the relevance of considering gut fill when using isotope dilution methods in ruminants [54–56]. Water content of elephant gut contents [57] and feces [58] ranges between 80 – 90% and thus is higher than the 73 - 75% hydration assumed in the calculation of FFM. Converting TBW (which includes the water in the gut) to FFM (including gut contents) using a hydration constant of 74%, as done in this study, will therefore lead to an overestimation of FFM (including gut contents). This overestimation will, in turn, lead to an underestimation of fat mass and body fat percentage. Such an underestimation of body fat has been reported in deer (*Odocoileus hemionus*) [54] and ponies, where body fat percentage determined via isotope dilution was on average 1.78% lower than carcass dissection-derived values [55]. Depending on the variation of gut water (80 - 90% in ruminants [55,57]) and variation in gut fill between 7 - 17% of body mass [35], it is likely that we under-estimated body fat mass by 6.0 – 14.5% (via an over-estimation of FFM). Further, within-subject variation body composition between measurement periods could be due in part to variation in gut fill, and true FFM (excluding gut contents) could be even more stable than the weight measurements and isotope dilution analyses suggest.” (lines 324-343)

? If a material has a water content of 20%, you cannot call it hay. Hay has a water content of about 10%. But what does that have to do with the water content of the gut contents? Gut contents are never as dry as having only 20% water. For elephants, it has been shown that gut contents in free-ranging animals have a water content of 80-90 %, and this is in line with other herbivores (where gut contents are never as dry as a dry diet ingested, incl. hay).

Response” This paragraph was completely re-written and this is not mentioned anymore.

I. 340 gut fill in herbivores is not only a function of a single meal, but of the overall intake level over the preceding week. The analysis is good to include, but one should not think that that excludes variation in gut fill due to what they ate the previous days, or the days between dosing and sampling. To keep gut fill constant, it is not so decisive what happens in the few hours prior to dosing, but during a whole 14-d-interval. This is why digestion studies with herbivores are so tedious – one cannot rely on 2-3 day saympling intervals to have a constant signal!

Response: Thank you for highlighting this aspect of ruminant digestion to us. We deleted the last part of the sentence:

“, which we would expect if elephants dosed later in the day had more time to feed and thus a higher gut fill.”

I agree with the calculations and explanations on water turnover. But not for the %fat estimates - because they will depend on the estimated FFM and if that is not constant with respect to gut contents, but variable (as one would expect when changing to a lower-calorie diet), then this will affect % estimates.

We changed this part of the discussion in the following way:

“Variation in gut fill, which can vary from 7% to 17% of body mass in elephants [35], must also be considered in interpreting measures of body composition for elephants in this study. Elephants in our sample likely increased their volume of food intake in response to the change from a high-calorie diet to a low-calorie diet [52]. Increased food intake will result in an increased gut fill [53] consisting of dry matter contents (which are mostly fat-free) and water (e.g., from saliva and other secretions). Previous studies have stated the relevance of considering gut fill when using isotope dilution methods in ruminants [54–56]. Water content of elephant gut contents [57] and feces [58] ranges between 80 – 90% and thus is higher than the 73 - 75% hydration assumed in the calculation of FFM. Converting TBW (which includes the water in the gut) to FFM (including gut contents) using a hydration constant of 74%, as done in this study, will therefore lead to an overestimation of FFM (including gut contents). This overestimation will, in turn, lead to an underestimation of fat mass and body fat percentage. Such an underestimation of body fat has been reported in deer (*Odocoileus hemionus*) [54] and ponies, where body fat percentage determined via isotope dilution was on average 1.78% lower than carcass dissection-derived values [55]. Depending on the variation of gut water (80 - 90% in ruminants [55,57]) and variation in gut fill between 7 - 17% of body mass [35], it is likely that we under-estimated body fat mass by 6.0 – 14.5% (via an over-estimation of FFM). Further, within-subject variation body composition between measurement periods could be due in part to variation in gut fill, and true FFM (excluding gut contents) could be even more stable than the weight measurements and isotope dilution analyses suggest.” (lines 324-343)

The information about the amount of hay bales given would benefit from a more accurate description of the hay bales:

Response: We clarified that rectangle bales were fed. (line 192)

Figures 1 and 2: We clarified that body fat percentage represents body fat percentage of body mass.